# ShuffleMTM: Learning Cross-channel Dependence in Multivariate Time Series from Shuffled Patches

## Abstract

Masked time-series modeling has widely gained attention as a self-supervised pre-training method for multivariate time series (MTS). Recent studies adopt a channel-independent (CI) strategy to enhance the temporal modeling capacity. Despite the effectiveness and performance of this strategy, the CI methods inherently overlook cross-channel dependence, which is inherent and crucial in MTS data in various domains. To fill this gap, we propose ShuffleMTM, a simple yet effective masked time-series modeling framework to learn cross-channel dependence from shuffled patches. Technically, ShuffleMTM proposes to shuffle the unmasked patches from masked series across different channels, positioned at the same index. Then, Siamese encoders learn two views of masked patch representations from original and shuffled masked series, simultaneously capturing the temporal dependence within a channel as well as spatial dependence across different channels. ShuffleMTM pre-trains the Siamese encoders to reconstruct the original series by incorporating cross-channel information with intra-channel cross-time information. Our proposed method consistently achieves superior performance in various experiments, compared to advanced CI pre-training methods and channel-dependent methods in both time series forecasting and classification tasks.

## 1 Introduction

Time series analysis plays a significant role in various domains, including energy, traffic and medicine. Significant amount of time series are collected from IoT sensors and wearable devices, with the majority being multivariate time series (MTS) that contains multiple channels (*a.k.a.*, variables). Due to the high cost of labeling for its indiscernible dependent structure, self-supervised pre-training has gained increasing popularity for identifying useful time series representations through pretext tasks on vast amounts of unlabeled data (Yue et al., 2022; Dong et al., 2024b). Notably, contrastive learning and masked modeling have demonstrated the superior performance in time series data, as well as in other fields like computer vision and natural language processing (Devlin et al., 2018; He et al., 2022; Chen et al., 2020).

Masked time-series modeling (MTM) focuses on learning temporal dependency through the reconstruction of masked segments based on the unmasked parts (Dong et al., 2024a). Meanwhile, recent MTMs have adopted the channel-independent (CI) strategy to strengthen the capability of modeling temporal relationships within a channel (Nie et al., 2023). CI MTMs have significantly improved the performance of various downstream tasks by concentrating on temporal patterns within each channel through univariate encoding (Nie et al., 2023; Lee et al., 2024). However, while CI methods separately learn cross-time dependency in each channel, their mechanism inherently overlooks the dependence among channels in MTS: Separate processing of univariate series cannot incorporate various interactions across channels, although these relationships might be implicitly reflected when multiple channels are simultaneously optimized in a single iteration. As patterns of each channel intricately influence each other (Zivot & Wang, 2006), neglecting the correlation among channels produces sub-optimal performance in downstream tasks (Zhang & Yan, 2023). These analyses raised an important question: how can we design a pre-training framework that effectively captures cross-channel dependence while maintaining the effectiveness of the channel-independent strategy?

(a) Channel-independent MTM          (b) ShuffleMTM

Figure 1: Comparison of channel-independent MTM and ShuffleMTM. (a) Channel-independent MTM reconstructs the masked series (a channel) using its own series. (b) ShuffleMTM reconstructs it integrating the representations of randomly shuffled masked series across channels. The gray cells indicate the masked patches of the channel, and the green and pink cells represent patches shuffled from other channels.

To fill the gap, we present ShuffleMTM, a simple yet effective self-supervised pre-training framework for multivariate time series. Unlike previous methods that recover masked patches from unmasked patches in the same channel (see Figure 1), ShuffleMTM proposes to randomly *shuffle* unmasked patches along the channel from masked series, termed as "*shuffled masked series*". Then, ShuffleMTM utilizes patch-based Transformers as encoder and Siamese networks to take the original and the shuffled masked series as inputs at each branch. With this simple shuffling mechanism and Siamese networks, ShuffleMTM learns two views of masked patch representations, each leveraging the temporal dependence between patches in the channel and spatial dependence between patches from different channels. Then, a decoder that takes the original and shuffled views of representations integrates these representations by utilizing cross-attention and self-attention mechanisms. Lastly, ShuffleMTM recovers each channel of raw time series by leveraging cross-channel information from shuffled patches.

Empowered by this design, ShuffleMTM extends the channel-independent reconstruction task to efficiently capture both temporal dependence within a channel and spatial dependence across channels. ShuffleMTM demonstrates state-of-the-art performance across various downstream tasks, including time series forecasting and classification. The main contributions of our work are summarized as follows:

- We identify and solve a problem in existing MTM methods: the channel-independent MTM overlooks cross-channel dependence inherent in MTS data. To address this issue, we propose ShuffleMTM, a simple yet effective pre-training framework for MTS data to capture complex spatial and temporal patterns.

- Specifically, ShuffleMTM shuffles unmasked patches along the channel to allow the model to attend to patches in other channels. In addition, ShuffleMTM leverages Siamese networks to encode both the original and shuffled masked series. This design extends the channel-independent reconstruction task to capture both spatial and temporal dependencies, representing the first technical contribution of MTM to learning cross-channel dependencies within the channel-independent strategy.

- Experimentally, ShuffleMTM consistently achieves state-of-the-art performance in time series forecasting and classification tasks. Ablation studies and further analyses show that both the shuffling mechanism and the Siamese network architecture are effective. The proposed ShuffleMTM is able to capture patch-level and channel-level dependencies, enhancing forecasting capacity and robustness compared to channel-independent MTM.

## 2 RELATED WORKS

**Channel Independence.** Channel-independent (CI) methods separately model each channel as a univariate time series, while channel-dependent methods jointly model multiple channels. Without explicitly considering interactions between channels, CI models focus on learning cross-time dependency within a channel. The concept of channel independence was first proposed in Nie et al. (2023), and subsequent research adopting this strategy has reported significant improvements in time series forecasting and classification tasks (Xu et al., 2024; Lee et al., 2024). The CI strategy has several advantages: it improves adaptability to various temporal patterns within the channel, increases training efficiency, and reduces the likelihood of overfitting (Nie et al., 2023; Liu et al., 2024b). Analysis from Han et al. (2024) reveal that CI forecasting models are more robust, whereas channel-

dependent models have higher forecasting capability, leading to consistently improved performance on MTS data with noise and distribution shifts. In this work, we propose a novel pre-training framework that utilizes cross-channel dependence in the CI strategy and demonstrate that our pre-training framework combines the advantages of both CI and channel-dependent models to achieve greater robustness and capacity in MTS forecasting.

**Cross-channel Dependence.** As the channels mutually influence one another in MTS, capturing cross-channel dependence is crucial in MTS modeling, allowing for richer representations of the underlying patterns (Zhang & Yan, 2023). In time series forecasting, some deep models explicitly capture the cross-channel dependence using convolutional neural networks or graph neural networks (Wu et al., 2020; Huang et al., 2023). For Transformer-based models, Crossformer (Zhang & Yan, 2023) proposes a two-stage attention layer in time and channel dimensions to utilize both cross-time and cross-channel dependencies. UniTST (Liu et al., 2024a) applies self-attention to the flattened time series and iTransformer (Liu et al., 2024b) inverts the embedding dimension to the channel perspective and perform self-attention on channels.

However, applying self-attention sequentially in horizontal and vertical manners, as well as in the inverted manner, is inefficient for learning dependencies between patches from other channels at lagged locations (Zhao & Shen, 2024). Similarly, applying self-attention to flattened patches from all channels allows access to unmasked patch embeddings with identical temporal information. This simplification, however, may lead the encoder to learn spurious information, hindering its training (Na et al., 2024). The proposed shuffling method dynamically imposes patches at lagged locations, capturing patch-wise dependencies across channels and integrates cross-channel information into the channel-independent reconstruction task without relying on identical temporal information.

**Masked Time-series Modeling.** While various MTS research addresses channel independence and cross-channel dependence, this paper focuses specifically on masked time-series modeling (MTM). As a principal paradigm in self-supervised pre-training, MTM optimizes deep models to capture temporal dependency by reconstructing masked parts from unmasked ones. Recent MTMs have adopted the CI strategy to enhance the capability of modeling temporal correlation. PatchTST (Nie et al., 2023) is the first CI MTM that divides each variable into multiple patches and reconstructs masked patches, thereby enhancing its temporal modeling capacity. SimMTM (Dong et al., 2024b) also utilizes the CI strategy to learn embedding manifold of variables from multiple masked variables. PITS (Lee et al., 2024) further advances the CI strategy to focus on temporal correlation within the patch through a patch-independent strategy. TimeSiam (Dong et al., 2024a) introduces a past-to-current reconstruction task in Siamese networks to accurately capture temporal correlations. Previous CI MTM methods focus exclusively on modeling temporal dependency. Despite their natural advantages in modeling temporal interactions, these methods inherently overlook dependencies among channels in MTS. Given the importance of cross-channel interactions in MTS modeling, we propose a novel MTM framework that captures these dependencies in the CI strategy.

## 3 SHUFFLEMTM

To capture dependence among channels in a channel-independent setting, we propose to generate shuffled masked series to utilize information from other channels in the reconstruction process (Section 3.1). ShuffleMTM performs cross-view representation learning using shuffled masked series and original masked series within Siamese networks (Section 3.2). ShuffleMTM then reconstructs each channel by integrating temporal dependency within the channel and spatial dependence from different channels (Section 3.3). During the fine-tuning stage, the weights of the ShuffleMTM encoder are transferred to downstream tasks without utilizing the shuffled view (Section 3.4).

### 3.1 MASKED SERIES SHUFFLING

We denote a multivariate time series sample $x = (x^{(1)}, \ldots, x^{(C)}) \in \mathbb{R}^{L \times C}$, where each $x^{(i)} \in \mathbb{R}^L$ contains $L$ timestamps and $C$ is the number of channels. First, the input series is decomposed into $C$ univariate series $x^{(i)}$, following the CI strategy. Then, each univariate series $x^{(i)}$ is divided into non-overlapping patches of length $P$, where $x_p^{(i)} = (x_p^{(i,1)}, \ldots, x_p^{(i,N)}) \in \mathbb{R}^{P \times N}$ is a sequence of patches and $N$ is the number of patches. Afterward, we randomly mask a portion of patches, where

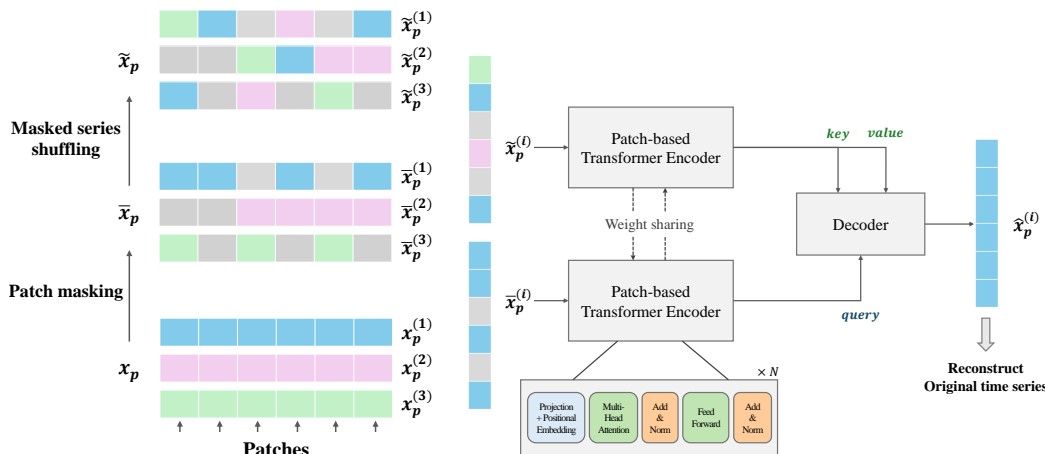

Figure 2: ShuffleMTM Architecture. Each colored cell represents a time series patch, with blue, pink, and green corresponding to three different channels. Gray cells denote masked patches. During pre-training, we randomly mask patches and shuffle unmasked patches along the channel dimension. The two views of each univariate time series channel are processed by Siamese encoders and integrated in a decoder with cross-attention layers to recover the raw time series. We illustrate the encoding process for the univariate time series of channel 1.

we denote a masked series $\bar{x}_p^{(i)} = mask_r(x_p^{(i)})$ and $r \in [0, 1]$ is the mask ratio, formalized by:

$$\bar{x}_p^{(i,j)} = \begin{cases} 0 & \text{if } j \in \mathbb{I}_m^{(i)} \\ x_p^{(i,j)} & \text{otherwise} \end{cases}$$

where $\mathbb{I}_m^{(i)}$ is the set of masked patch indices on channel $i$. We also define $\mathbb{J}_m^{(j)}$ as the set of masked patch indices at patch index $j$ across channels. The CI approach, by design, cannot learn relationships among patches from different channels, thereby causing the model to neglect cross-channel interactions. To incorporate information from different channels within independent channel encoding, we propose to randomly shuffle unmasked patches along the channel (Figure 2). Concretely, we generate a shuffled masked series $\tilde{x}_p^{(i)} = (\tilde{x}_p^{(i,1)}, \dots, \tilde{x}_p^{(i,N)})$ by rearranging unmasked patches along the channel axis while keeping the masked patches fixed, formalized as follows:

$$\tilde{x}_p = (\tilde{x}_p^{(1)}, \dots, \tilde{x}_p^{(C)}) = shuffle\left( (\bar{x}_p^{(1)}, \dots, \bar{x}_p^{(C)}) \right)$$

$$\tilde{x}_p^{(i,j)} = \begin{cases} 0 & \text{if } j \in \mathbb{I}_m^{(i)} \\ \bar{x}_p^{(i',j)} & \text{otherwise} \end{cases}$$

where $i' \in \{1, \dots, C\} \setminus \mathbb{J}_m^{(j)}$ is randomly selected without replacement among unmasked patches at patch index $j$ across channels. By obtaining a pair of $(\bar{x}_p^{(i)}, \tilde{x}_p^{(i)})$, we can construct two views of a channel univariate series: an original masked series that retains the true temporal patterns and a shuffled masked series that establishes the inter-channel dependencies.

## 3.2 CROSS-VIEW REPRESENTATION LEARNING

For processing patched masked series, ShuffleMTM utilizes patch-based Transformer encoder (Nie et al., 2023), where, the patches are mapped into the latent space of dimension $d_m$ through a learnable linear projection $\boldsymbol{W}_{patch} \in \mathbb{R}^{d_m \times P}$ and a learnable positional embedding $W_{pos} \in \mathbb{R}^{d_m \times N}$: $x_{emb}^{(i)} = \boldsymbol{W}_{patch} \cdot x_p^{(i)} + \boldsymbol{W}_{pos}$. Next, we apply a vanilla Transformer encoder (Vaswani, 2017) to sequence of patch embeddings.

Given a pair of original and shuffled masked series, ShuffleMTM leverages Siamese networks to learn two views of masked series representations. Siamese networks (Bromley et al., 1993) are two-branch neural network architectures sharing model parameters. After the Siamese encoders, we can obtain pairs of representations of original and shuffled masked series as:

$$\bar{z}_p^{(i)} = \text{Encoder}(\bar{x}_p^{(i)}), \ \tilde{z}_p^{(i)} = \text{Encoder}(\tilde{x}_p^{(i)})$$

Siamese encoders allow ShuffleMTM to leverage both temporal information from the original masked series representations $\bar{z}_p^{(i)}$ and spatial information from the shuffled masked series representations $\tilde{z}_p^{(i)}$ in reconstruction-based pre-training. In one branch, $\bar{z}_p^{(i)}$ models temporal dependencies within a channel, as done by CI MTM encoders, while the other branch, $\tilde{z}_p^{(i)}$, models cross-channel dependencies by attending to shuffled patches, similar to channel-dependent models. By processing a pair of masked series, we obtain two views of masked series representations: one focused on the temporal structure of a single channel and the other capturing spatial dependencies between patches across channels.

Note that any inductive bias associated with channel information is provided to the shuffled masked series. Some previous studies have constructed the channel embeddings to encode the relationships across channels (Zhang & Yan, 2023). However, there is no predetermined position for different channels in MTS (Xiao et al., 2024). It is challenging to train embeddings to build inductive bias to understand the channel-wise structure (Su et al., 2024). Thus, ShuffleMTM excludes any channel-related bias, ensuring it focuses on the dependencies hidden within cross-channel patterns.

## 3.3 SELF-SUPERVISED PRE-TRAINING

To incorporate learned cross-time and cross-channel dependencies into pre-training of masked modeling, the output representations of the Siamese encoders are fed into a decoder with cross-attention and self-attention mechanisms (Gupta et al., 2023). A decoder block consists of a cross-attention layer, a self-attention layer and a Feed-Forward Network (FFN). $\bar{z}_p^{(i)}$ acts as the query, and $\tilde{z}_p^{(i)}$ serves for the key and value in the cross-attention layer. Next, the representation attends to each other via the self-attention layer and is passed to the FFN. As the cross-attention layer is functionally similar to learning the similarity matrix between target and reference in self-supervised correspondence learning (Gupta et al., 2023; Vondrick et al., 2018), the decoder can integrate different views of patch representations in the original and shuffled masked series. For clarity, we formalize this process as $\bar{h}_p^{(i)} = \text{Decoder}(\bar{z}_p^{(i)}, \tilde{z}_p^{(i)})$. Finally, the integrated representation $\bar{h}_p^{(i)} \in \mathbb{R}^{d_m \times N}$ is used to reconstruct the original time series through a linear projection head on each patch: $\hat{x}_p^{(i)} = \text{projector}(\bar{h}_p^{(i)})$.

Leveraging this design, ShuffleMTM can reconstruct the original input series (a single channel) by referring to cross-channel relationships encoded in the shuffled series representations, thereby capturing both cross-time and cross-channel dependencies. The overall reconstruction loss is gathered across $C$ channels and averaged as: $L = \mathbb{E}_x \frac{1}{C} \sum_{i=1}^{C} ||\hat{x}_p^{(i)} - x_p^{(i)}||_2^2$.

## 3.4 FINE-TUNING TO DOWNSTREAM TASKS

Through the random shuffling mechanism in pre-training, ShuffleMTM learns both cross-time and cross-channel dependencies in multivariate time series. During the fine-tuning process, the shuffled view is not utilized and one branch of Siamese encoders for the shuffled view is removed. The weights of the encoder are transferred to project each channel (univariate time series) into a deep representation, which is then fine-tuned with a linear decoder layer to predict for the downstream tasks. The cross-channel dependence learned during the pre-training is effectively transferred to the downstream tasks without the shuffled view.

## 3.5 RELATIONS WITH PREVIOUS WORKS

It is notable that ShuffleMTM can be reduced to PatchTST (Nie et al., 2023) if the branch for the shuffled masked series is removed and its corresponding decoder is replaced with a linear head. However, ShuffleMTM is fundamentally different from PatchTST, as ShuffleMTM captures both cross-time and cross-channel dependencies through simple masked series shuffling and the use of Siamese encoders, while PatchTST focuses solely on cross-time dependencies within each channel. TimeSiam aims to enhance time-dependent representation learning, while ShuffleMTM expands CI's limited focus on temporal dependency—a limitation shared by TimeSiam—to learning cross-channel dependence. Additionally, TimeSiam employs complex time-difference embeddings, whereas ShuffleMTM has a simpler architecture that does not require any additional embedding.

| Models | | Self-supervised | | | | | | | | | Supervised | | | | | | | | | | |
|---|---|---|---|---|---|---|---|---|---|---|---|---|---|---|---|---|---|---|---|---|---|---|
| | | ShuffleMTM | | TimeSiam | | PITS | | PatchTST | | SimMTM | | iTransformer | | Crossformer | | CrossGNN | | MTGNN | | PatchTST (sup) | | Dlinear | |
| Metrics | | MSE | MAE | MSE | MAE | MSE | MAE | MSE | MAE | MSE | MAE | MSE | MAE | MSE | MAE | MSE | MAE | MSE | MAE | MSE | MAE | MSE | MAE |
| ETTh1 | 96 | 0.376 | 0.397 | 0.379 | 0.402 | 0.377 | 0.395 | 0.379 | 0.399 | **0.367** | **0.389** | 0.386 | 0.405 | 0.391 | 0.418 | 0.389 | 0.399 | 0.515 | 0.517 | 0.392 | 0.407 | 0.390 | 0.404 |
| | 192 | **0.420** | 0.425 | 0.425 | 0.431 | 0.425 | 0.425 | 0.425 | 0.427 | 0.424 | **0.423** | 0.443 | 0.437 | 0.450 | 0.453 | 0.441 | 0.430 | 0.553 | 0.522 | 0.445 | 0.434 | 0.451 | 0.446 |
| | 336 | **0.456** | **0.446** | 0.459 | 0.451 | 0.478 | 0.448 | 0.470 | 0.446 | 0.473 | 0.456 | 0.489 | 0.460 | 0.526 | 0.503 | 0.484 | 0.452 | 0.612 | 0.577 | 0.483 | 0.451 | 0.498 | 0.474 |
| | 720 | **0.474** | 0.471 | 0.475 | 0.478 | 0.499 | 0.475 | 0.482 | 0.466 | 0.494 | 0.493 | 0.508 | 0.494 | 0.643 | 0.593 | 0.483 | 0.472 | 0.609 | 0.597 | 0.477 | 0.469 | 0.511 | 0.505 |
| | Avg | **0.432** | **0.435** | 0.435 | 0.441 | 0.446 | 0.436 | 0.439 | 0.435 | 0.440 | 0.440 | 0.457 | 0.449 | 0.503 | 0.492 | 0.449 | 0.438 | 0.572 | 0.553 | 0.449 | 0.440 | 0.463 | 0.457 |
| ETTh2 | 96 | **0.288** | **0.338** | 0.291 | 0.343 | 0.289 | 0.338 | 0.299 | 0.346 | 0.299 | 0.347 | 0.301 | 0.351 | 0.683 | 0.562 | 0.295 | 0.345 | 0.354 | 0.454 | 0.299 | 0.346 | 0.378 | 0.413 |
| | 192 | **0.368** | **0.390** | 0.370 | 0.393 | 0.374 | 0.393 | 0.382 | 0.398 | 0.383 | 0.398 | 0.379 | 0.399 | 0.824 | 0.628 | 0.383 | 0.400 | 0.457 | 0.464 | 0.382 | 0.398 | 0.447 | 0.452 |
| | 336 | **0.412** | **0.426** | 0.414 | 0.428 | 0.415 | 0.426 | 0.427 | 0.433 | 0.420 | 0.430 | 0.422 | 0.432 | 0.966 | 0.701 | 0.427 | 0.440 | 0.515 | 0.540 | 0.427 | 0.433 | 0.515 | 0.497 |
| | 720 | **0.421** | **0.441** | 0.422 | 0.443 | 0.423 | 0.443 | 0.438 | 0.452 | 0.425 | 0.444 | 0.435 | 0.450 | 1.395 | 0.853 | 0.436 | 0.453 | 0.532 | 0.576 | 0.438 | 0.452 | 0.688 | 0.593 |
| | Avg | **0.372** | **0.399** | 0.374 | 0.402 | 0.375 | 0.400 | 0.387 | 0.407 | 0.382 | 0.405 | 0.384 | 0.408 | 0.967 | 0.686 | 0.385 | 0.410 | 0.465 | 0.509 | 0.387 | 0.407 | 0.507 | 0.489 |
| ETTm1 | 96 | **0.317** | 0.357 | **0.317** | 0.359 | 0.332 | 0.363 | 0.318 | 0.356 | 0.327 | 0.365 | 0.343 | 0.377 | 0.360 | 0.395 | 0.345 | 0.372 | 0.379 | 0.446 | 0.321 | 0.359 | 0.346 | 0.372 |
| | 192 | 0.361 | 0.382 | 0.357 | 0.382 | 0.366 | 0.384 | **0.356** | **0.380** | 0.368 | 0.389 | 0.381 | 0.394 | 0.390 | 0.410 | 0.379 | 0.388 | 0.470 | 0.428 | 0.362 | 0.382 | 0.383 | 0.393 |
| | 336 | 0.390 | **0.402** | 0.386 | 0.403 | 0.396 | 0.406 | **0.385** | 0.403 | 0.396 | 0.409 | 0.419 | 0.419 | 0.452 | 0.456 | 0.410 | 0.408 | 0.473 | 0.430 | 0.393 | 0.403 | 0.415 | 0.416 |
| | 720 | 0.446 | **0.435** | 0.444 | 0.438 | 0.457 | 0.441 | **0.444** | 0.439 | 0.452 | 0.440 | 0.490 | 0.458 | 0.542 | 0.516 | 0.469 | 0.441 | 0.553 | 0.479 | 0.453 | 0.437 | 0.475 | 0.454 |
| | Avg | 0.379 | **0.394** | 0.376 | 0.396 | 0.388 | 0.399 | **0.376** | 0.395 | 0.386 | 0.401 | 0.408 | 0.412 | 0.436 | 0.444 | 0.401 | 0.402 | 0.469 | 0.446 | 0.382 | 0.395 | 0.405 | 0.409 |
| ETTm2 | 96 | **0.175** | **0.259** | 0.176 | 0.261 | 0.177 | 0.261 | 0.176 | 0.262 | 0.186 | 0.276 | 0.184 | 0.269 | 0.269 | 0.353 | 0.179 | **0.259** | 0.203 | 0.299 | 0.178 | 0.260 | 0.188 | 0.282 |
| | 192 | **0.240** | 0.302 | 0.241 | 0.303 | 0.244 | 0.304 | 0.242 | 0.306 | 0.253 | 0.317 | 0.252 | 0.313 | 0.379 | 0.432 | 0.243 | **0.302** | 0.265 | 0.328 | 0.245 | 0.304 | 0.269 | 0.343 |
| | 336 | **0.299** | 0.340 | 0.302 | 0.342 | 0.302 | 0.342 | 0.304 | 0.346 | 0.317 | 0.356 | 0.315 | 0.352 | 0.520 | 0.535 | 0.304 | 0.342 | 0.365 | 0.374 | 0.307 | 0.343 | 0.351 | 0.400 |
| | 720 | 0.399 | 0.398 | 0.400 | 0.398 | 0.400 | **0.397** | 0.406 | 0.405 | 0.417 | 0.412 | 0.412 | 0.406 | 1.453 | 0.875 | 0.405 | 0.399 | 0.461 | 0.459 | 0.406 | 0.401 | 0.492 | 0.484 |
| | Avg | **0.278** | 0.325 | 0.280 | 0.326 | 0.280 | 0.326 | 0.282 | 0.330 | 0.293 | 0.340 | 0.291 | 0.335 | 0.655 | 0.549 | 0.283 | **0.326** | 0.324 | 0.365 | 0.284 | 0.327 | 0.325 | 0.377 |
| Exchange | 96 | **0.082** | **0.199** | 0.085 | 0.204 | 0.084 | 0.200 | 0.085 | 0.203 | 0.087 | 0.208 | 0.087 | 0.207 | 0.429 | 0.453 | 0.084 | **0.200** | 0.102 | 0.228 | 0.088 | 0.205 | 0.088 | 0.218 |
| | 192 | **0.173** | **0.296** | 0.182 | 0.304 | 0.176 | 0.297 | 0.182 | 0.302 | 0.180 | 0.303 | 0.179 | 0.302 | 0.531 | 0.554 | 0.177 | **0.296** | 0.267 | 0.335 | 0.176 | 0.297 | 0.176 | 0.315 |
| | 336 | **0.324** | **0.411** | 0.334 | 0.419 | 0.340 | 0.421 | 0.331 | 0.416 | 0.330 | 0.417 | 0.335 | 0.420 | 0.886 | 0.732 | 0.340 | 0.418 | 0.393 | 0.457 | 0.344 | 0.424 | **0.313** | 0.427 |
| | 720 | **0.833** | **0.687** | 0.866 | 0.702 | 0.855 | 0.696 | 0.869 | 0.700 | 0.852 | 0.696 | 0.853 | 0.697 | 1.571 | 1.016 | 1.017 | 0.761 | 1.090 | 0.811 | 0.904 | 0.716 | 0.839 | 0.695 |
| | Avg | **0.353** | **0.398** | 0.367 | 0.407 | 0.364 | 0.404 | 0.367 | 0.405 | 0.362 | 0.406 | 0.364 | 0.407 | 0.854 | 0.689 | 0.405 | 0.419 | 0.463 | 0.458 | 0.378 | 0.411 | 0.354 | 0.414 |
| Weather | 96 | 0.168 | **0.211** | 0.173 | 0.215 | 0.184 | 0.223 | 0.170 | 0.212 | 0.170 | 0.216 | 0.175 | 0.215 | **0.158** | 0.230 | 0.259 | 0.230 | 0.329 | 0.371 | 0.178 | 0.219 | 0.199 | 0.260 |
| | 192 | 0.215 | 0.253 | 0.222 | 0.257 | 0.229 | 0.262 | 0.215 | 0.253 | 0.216 | 0.255 | 0.225 | 0.258 | **0.206** | 0.298 | 0.227 | 0.302 | 0.263 | 0.322 | 0.225 | 0.259 | 0.237 | 0.294 |
| | 336 | 0.271 | 0.293 | 0.273 | 0.294 | 0.283 | 0.300 | 0.273 | 0.294 | **0.271** | 0.294 | 0.281 | 0.299 | 0.272 | 0.335 | 0.282 | 0.342 | 0.354 | 0.396 | 0.278 | 0.298 | 0.283 | 0.332 |
| | 720 | 0.348 | 0.343 | 0.353 | 0.346 | 0.357 | 0.349 | 0.348 | 0.345 | 0.346 | 0.343 | 0.359 | 0.350 | 0.398 | 0.386 | 0.360 | 0.399 | 0.409 | 0.371 | 0.351 | 0.347 | 0.383 | |
| | Avg | 0.250 | 0.275 | 0.255 | 0.278 | 0.263 | 0.284 | 0.252 | 0.276 | **0.251** | 0.277 | 0.262 | 0.281 | 0.398 | 0.386 | 0.262 | 0.326 | 0.314 | 0.355 | 0.258 | 0.281 | 0.267 | 0.317 |
| Electricity | 96 | 0.161 | 0.248 | 0.161 | 0.247 | 0.187 | 0.269 | 0.174 | 0.258 | 0.198 | 0.291 | **0.148** | **0.240** | 0.226 | 0.308 | 0.199 | 0.279 | 0.217 | 0.318 | 0.166 | 0.252 | 0.195 | 0.278 |
| | 192 | 0.170 | 0.257 | 0.171 | 0.257 | 0.190 | 0.274 | 0.183 | 0.267 | 0.200 | 0.293 | **0.164** | **0.256** | 0.276 | 0.339 | 0.198 | 0.281 | 0.238 | 0.352 | 0.174 | 0.260 | 0.194 | 0.280 |
| | 336 | 0.186 | 0.274 | 0.187 | 0.274 | 0.206 | 0.289 | 0.198 | 0.283 | 0.215 | 0.308 | **0.179** | **0.272** | 0.357 | 0.393 | 0.213 | 0.296 | 0.260 | 0.348 | 0.190 | 0.277 | 0.207 | 0.296 |
| | 720 | 0.228 | 0.310 | 0.226 | 0.308 | 0.247 | 0.322 | 0.238 | 0.316 | 0.258 | 0.340 | **0.211** | **0.300** | 0.406 | 0.422 | 0.253 | 0.329 | 0.290 | 0.369 | 0.231 | 0.312 | 0.243 | 0.329 |
| | Avg | 0.186 | 0.272 | 0.186 | 0.272 | 0.208 | 0.289 | 0.198 | 0.281 | 0.218 | 0.308 | **0.176** | **0.267** | 0.316 | 0.366 | 0.216 | 0.296 | 0.251 | 0.347 | 0.190 | 0.275 | 0.210 | 0.296 |
| Traffic | 96 | 0.424 | **0.270** | 0.431 | 0.278 | 0.555 | 0.349 | 0.475 | 0.307 | 0.542 | 0.342 | **0.393** | 0.268 | 0.657 | 0.311 | 0.657 | 0.390 | 0.660 | 0.437 | 0.445 | 0.283 | 0.649 | 0.397 |
| | 192 | 0.437 | 0.274 | 0.443 | 0.282 | 0.536 | 0.339 | 0.481 | 0.308 | 0.530 | 0.334 | **0.413** | 0.277 | 0.565 | 0.315 | 0.608 | 0.370 | 0.649 | 0.438 | 0.453 | 0.285 | 0.599 | 0.371 |
| | 336 | 0.451 | 0.281 | 0.457 | 0.288 | 0.546 | 0.341 | 0.491 | 0.309 | 0.541 | 0.338 | **0.424** | 0.283 | 0.587 | 0.323 | 0.617 | 0.373 | 0.653 | 0.472 | 0.468 | 0.291 | 0.606 | 0.374 |
| | 720 | 0.483 | **0.300** | 0.489 | 0.307 | 0.582 | 0.359 | 0.524 | 0.328 | 0.578 | 0.355 | **0.457** | 0.300 | 0.589 | 0.319 | 0.659 | 0.390 | 0.639 | 0.437 | 0.501 | 0.310 | 0.646 | 0.396 |
| | Avg | 0.449 | **0.281** | 0.455 | 0.289 | 0.555 | 0.347 | 0.493 | 0.313 | 0.548 | 0.342 | **0.422** | 0.282 | 0.571 | 0.317 | 0.635 | 0.381 | 0.650 | 0.446 | 0.467 | 0.292 | 0.625 | 0.385 |

Table 1: In-domain forecasting. All models are both pre-trained and fine-tuned on the same dataset. The best results are in bold, and the second-best results are underlined.

| Models | ShuffleMTM | | TimeSiam | | PITS | | PatchTST | | SimMTM | |
|---|---|---|---|---|---|---|---|---|---|---|
| Metrics | MSE | MAE | MSE | MAE | MSE | MAE | MSE | MAE | MSE | MAE |
| ETTh1→ETTh2 | 0.375 | **0.400** | 0.374 | 0.401 | 0.376 | 0.401 | 0.381 | 0.406 | 0.382 | 0.407 |
| ETTh2→ETTh1 | **0.434** | **0.435** | 0.444 | 0.450 | 0.442 | 0.436 | 0.438 | 0.437 | 0.445 | 0.446 |
| ETTh1→ETTm1 | **0.380** | **0.393** | 0.395 | 0.404 | 0.387 | 0.398 | 0.386 | 0.395 | 0.383 | 0.398 |
| ETTh2→ETTm1 | **0.381** | **0.394** | 0.399 | 0.411 | 0.387 | 0.398 | 0.385 | 0.396 | 0.456 | 0.428 |
| ETTm2→ETTm1 | **0.378** | **0.396** | 0.395 | 0.404 | 0.387 | 0.398 | 0.379 | 0.396 | 0.396 | 0.402 |
| ETTm2→ETTh1 | 0.437 | **0.433** | 0.434 | 0.441 | 0.439 | 0.437 | 0.443 | 0.443 | 0.457 | 0.452 |
| Weather→ETTh1 | 0.436 | 0.441 | 0.438 | 0.443 | 0.439 | 0.433 | 0.441 | 0.440 | **0.426** | **0.435** |
| Weather→ETTm1 | **0.378** | **0.395** | 0.387 | 0.403 | 0.387 | 0.398 | 0.380 | 0.396 | 0.385 | 0.399 |

Table 2: Cross-domain forecasting. All models are pre-trained on source dataset and fine-tuned on target dataset.

| Dataset | Model | Accuracy | Precision | Recall | F1 score |
|---|---|---|---|---|---|
| AD | ShuffleMTM | **93.93** | **93.82** | **94.17** | **93.90** |
| | TimeSiam | 89.69 | 89.73 | 89.51 | 59.55 |
| | PITS | 76.90 | 82.85 | 76.90 | 76.56 |
| | PatchTST | 62.65 | 65.57 | 62.65 | 61.07 |
| | SimMTM | 66.98 | 75.03 | 69.67 | 65.56 |
| | COMET | 91.11 | 92.39 | 89.89 | 92.10 |
| PTB | ShuffleMTM | **91.58** | **91.82** | 86.91 | **88.90** |
| | TimeSiam | 90.09 | 92.24 | 83.17 | 83.32 |
| | PITS | 87.57 | 90.16 | 84.06 | 81.79 |
| | PatchTST | 90.36 | 90.51 | **88.84** | 86.98 |
| | SimMTM | 84.49 | 83.99 | 75.64 | 78.28 |
| | COMET | 87.37 | 87.38 | 81.13 | 83.03 |

Table 3: In-domain classification

## 4 EXPERIMENTS

### 4.1 EXPERIMENTAL SETTINGS

**Tasks and evaluation metrics.** We evaluate the performance of ShuffleMTM on two downstream tasks: time series forecasting and classification. We follow the standard self-supervised framework, where the model is pre-trained with unlabeled data and fine-tuned on the same data with labels. We also consider in-domain and cross-domain transfer, as well as limited labeled data scenarios in some experiments. We use mean squared error (MSE) and mean absolute error (MAE) for forecasting, and accuracy, precision, recall, and F1 score for classification. We report the average performance over five runs for each experiment. More implementation details are provided in Appendix A.

### 4.2 TIME SERIES FORECASTING

**Datasets and baselines.** We conduct extensive experiments on eight real-world benchmarks following Wu et al. (2021), including four ETT datasets (ETTh1, ETTh2, ETTm1, ETTm2), Exchange, Weather, Electricity, and Traffic. It is worth noting that all datasets are multivariate time series, some of which have a large number of channels, such as 321 for Electricity and 862 for Traffic. For the baselines, we consider channel-independent MTM methods, SimMTM (Dong et al., 2024b), PatchTST (Nie et al., 2023), PITS (Lee et al., 2024), and TimeSiam (Dong et al., 2024a). We choose PatchTST-backbone TimeSiam for the comparison over CI MTM method. We also compare with four state-of-the-art channel-dependent forecasting methods, iTransformer (Liu et al., 2024b), Crossformer (Zhang & Yan, 2023), CrossGNN (Huang et al., 2023), and MTGNN (Wu et al., 2020)

| Fraction | Models | AD | | | | PTB | | | |
|---|---|---|---|---|---|---|---|---|---|
| | | Accuracy | Precision | Recall | F1 score | Accuracy | Precision | Recall | F1 score |
| 20% | ShuffleMTM | **96.38** | **96.33** | **96.38** | **96.34** | 90.64 | 90.43 | **85.94** | 87.70 |
| | TimeSiam | 91.25 | 81.84 | 90.67 | 91.03 | **91.24** | **94.03** | 84.60 | **87.92** |
| | PITS | 75.82 | 78.67 | 77.36 | 75.69 | 87.29 | 90.59 | 78.11 | 81.66 |
| | PatchTST | 78.67 | 82.02 | 80.36 | 78.53 | 85.74 | 91.02 | 74.71 | 78.44 |
| | SimMTM | 70.55 | 73.91 | 72.29 | 70.16 | 85.09 | 84.43 | 76.82 | 79.36 |
| | COMET | 92.55 | 92.49 | 92.73 | 92.50 | 87.46 | 88.89 | 82.30 | 84.46 |
| 10% | ShuffleMTM | **93.75** | **93.79** | **93.60** | **93.67** | **89.11** | 89.55 | 82.82 | **85.21** |
| | TimeSiam | 93.35 | 93.52 | 93.06 | 93.24 | 88.50 | **92.95** | 79.44 | 83.34 |
| | PITS | 75.37 | 81.49 | 75.37 | 75.10 | 83.28 | 85.52 | 83.28 | 73.91 |
| | PatchTST | 68.41 | 74.15 | 68.41 | 67.92 | 86.80 | 89.15 | 83.38 | 80.56 |
| | SimMTM | 68.23 | 76.44 | 70.90 | 66.22 | 85.58 | 84.49 | 78.05 | 80.32 |
| | COMET | 92.06 | 92.07 | 91.92 | 91.96 | 87.75 | 86.68 | 82.07 | 83.48 |
| 5% | ShuffleMTM | **91.17** | 91.41 | **91.79** | **91.15** | 88.15 | 90.41 | 79.94 | 83.27 |
| | TimeSiam | 83.35 | 84.61 | 82.39 | 82.70 | 82.60 | 89.02 | 69.14 | 72.25 |
| | PITS | 71.93 | 78.59 | 74.37 | 71.23 | 85.72 | 88.06 | 75.92 | 79.28 |
| | PatchTST | 85.36 | 83.23 | 82.37 | 80.91 | 80.77 | 88.64 | 65.71 | 67.96 |
| | SimMTM | 68.27 | 75.47 | 70.81 | 66.80 | 84.09 | 84.02 | 74.64 | 77.37 |
| | COMET | 90.50 | **91.84** | 89.90 | 90.20 | **89.08** | **89.57** | 82.63 | **85.21** |

Table 4: Limited labeled data classification

and two CI forecasting methods, supervised PatchTST (Nie et al., 2023) and DLinear (Zeng et al., 2023). We follow the experimental settings and baseline results in TimeSiam, iTransformer and CrossGNN, with a default look-back window $L = 96$ and a forecast horizon $\{96, 192, 336, 720\}$.

**In-domain forecasting.** As shown in Table 11, ShuffleMTM exhibits superior performance compared to both self-supervised pre-training and supervised forecasting baselines, achieving the best or second-best results in 72 out of 80 forecasting scenarios. Notably, ShuffleMTM outperforms CI MTM baselines, demonstrating significantly improved performance on the Traffic dataset, which contains the largest number of channels among the benchmarks. Although ShuffleMTM achieves the second-best performance on the Electricity and Traffic datasets, following iTransformer, a state-of-the-art forecasting method, the results emphasize the importance of cross-channel dependence in MTS forecasting with a large number of channels.

**Cross-domain forecasting.** We investigate the robustness of the pre-trained model to mismatched data distributions that were not encountered during pre-training. Table 12 presents multiple transfer scenarios to demonstrate its effectiveness in the cross-domain forecasting setting. We compare ShuffleMTM with CI MTM baselines, as channel independence allows the pre-trained model to be transferred to target datasets with an arbitrary number of variables: Weather → ETTh1 and ETTm1. The results indicate that ShuffleMTM consistently outperforms all baselines in most scenarios.

## 4.3 TIME SERIES CLASSIFICATION

**Datasets and baselines.** We select two classification datasets in the medical domain covering two mainstream signals, electroencephalography (EEG) and electrocardiogram (ECG): **AD** (Escudero et al., 2006) and **PTB** (Goldberger et al., 2000). These datasets have multiple channels, 16 for AD and 15 for PTB. For baselines, we compare with CI MTM methods, TimeSiam, PITS, PatchTST, and SimMTM. COMET (Wang et al., 2024) is also included in baselines to verify the classification performance is comparable with the most recent self-supervised model in the medical domain.

**In-domain classification.** Table 3 demonstrates that the proposed ShuffleMTM outperforms self-supervised baseline methods in classification task. As channels in bioelectrical signals represent the different views of the same physical activity, leveraging spatial information between channels is significant for learning biosignal representations (Kiyasseh et al., 2021). This observation justifies the superior performance of ShuffleMTM compared to CI MTM baselines. Moreover, ShuffleMTM exhibits better performance than COMET, which exploits meta information in medical domain in self-supervised pre-training, such as trial or patient IDs. These results illustrates the effectiveness of ShuffleMTM in capturing cross-channel relationships, even when meta information is not available.

## 5 ABLATION STUDIES

**Fine-tuning to limited labeled data.** We investigate the effectiveness of the representation under limited labeled training data for classification. We utilize 20%, 10%, and 5% of the labeled training data during the fine-tuning stage. As shown in Table 4, ShuffleMTM outperforms the baselines in

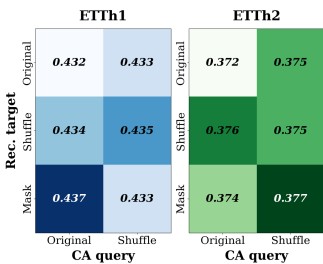

Figure 3: Reconstruction target

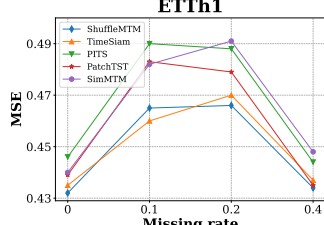

Figure 4: Robustness to missing data

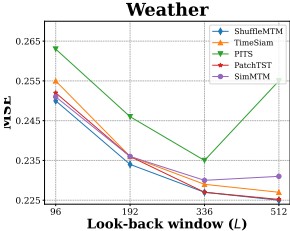

Figure 5: Look-back window

label-scarce settings. Notably, ShuffleMTM achieves state-of-the-art performance on the AD dataset across the 20%, 10%, and 5% scenarios. For the PTB dataset, ShuffleMTM performs comparably to TimeSiam. Meanwhile, COMET shows consistent performance even in the 5% scenario, as pre-training with meta information provides pseudo-label knowledge.

**Reconstruction target** We study the effect of the reconstruction target and the choice of query in the cross-attention of the decoder. We consider six variations of ShuffleMTM, involving two choices of query—original masked series and shuffled masked series—and three reconstruction targets: the original time series, the shuffled time series which is used as the input in the Siamese encoders, and only the masked patches. In Figure 3, all variations show solid performance, with our setting (i.e., reconstructing the original time series using the original masked series as the query) performing the best. This analysis indicates that integrating original and shuffled views of masked series representations is crucial for forecasting performance owing to the integration of spatial and temporal information in MTS, irrespective of the reconstruction target or query choice.

**Robustness to missing data** To demonstrate model robustness to the missing data, we randomly remove a portion of timestamps from the train and test datasets, and the model predicts the original values in the test dataset. As shown in Figure 4, ShuffleMTM shows the lowest MSE performance in various missing rate, which suggests that the pre-training architecture effectively leverages cross-channel dependence even in the presence of missing data. These results demonstrate the superior robustness of ShuffleMTM in the data corruption.

**Varying look-back window** We investigate the effectiveness of Shuf-fleMTM for time series forecasting in longer look-back windows {96, 192, 336, 512}. The less a model's performance degrades at the missing data, the more robust it is considered. We demonstrate in Figure 5 that our ShuffleMTM consistently reduces the MSE error as the look-back window increases and achieves the best performance compared to other CI MTM methods in all look-back windows. This result confirms the effectiveness of ShuffleMTM to learn from increased look-back window.

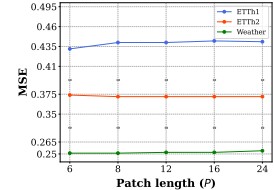

Figure 6: Patch length

**Hyperparameter sensitivity.** We study the effect of mask ratio and patch length, which are the key hyperparameters in masked modeling and patch-based Transformer encoder. We vary the patch lengths {6, 8, 12, 16, 24} and mask ratios in {0.2, 0.4, 0.6, 0.8}. As shown in Figures 7 and 6, ShuffleMTM is robust to the patch lengths. While Shuf-fleMTM is robust to masking ratios in small-channel datasets, their impact is more pronounced on high-channel datasets, such as Electricity. When the masking ratio is low, the proportion of self-supervision is insufficient, resulting in poor forecasting performance. Conversely, as the masking ratio increases, the number of potential shuffled candidate loca-

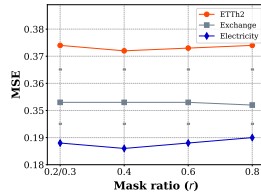

Figure 7: Mask ratio

tions decreases. As a result, the diversity of patch replacements diminishes, degrading forecasting performance. Therefore, an appropriate masking ratio is critical for forecasting performance on high-channel datasets.

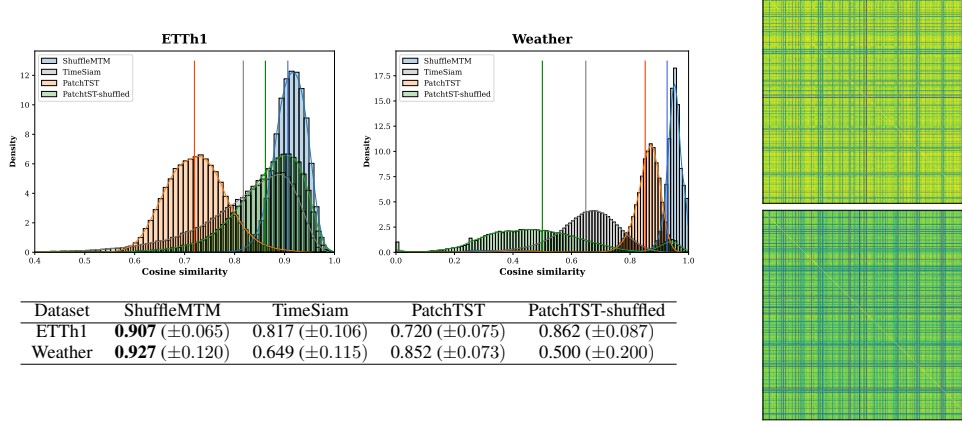

| Dataset | ShuffleMTM | TimeSiam | PatchTST | PatchTST-shuffled |
|---------|------------|----------|----------|-------------------|
| ETTh1 | **0.907** ($\pm$0.065) | 0.817 ($\pm$0.106) | 0.720 ($\pm$0.075) | 0.862 ($\pm$0.087) |
| Weather | **0.927** ($\pm$0.120) | 0.649 ($\pm$0.115) | 0.852 ($\pm$0.073) | 0.500 ($\pm$0.200) |

Figure 8: Cross-channel dependence analysis. (**Left**) Distributions of similarities between the attention map of shuffled patched series and its patch-correlation matrix on the ETTh1 and Weather datasets. (**Right**) A case visualization of cross-channel correlations of raw time series and pairwise distances of the learned channel embeddings of ShuffleMTM on the Traffic dataset.

## 6 MODEL ANALYSIS

### 6.1 CROSS-CHANNEL DEPENDENCE ANALYSIS

To validate ShuffleMTM's capacity to capture cross-channel dependence in a channel-independent encoding, we conduct two experiments, focusing on both patch-level and channel-level dependence.

**Patch-level dependence.** We perform a similarity analysis on the shuffled patched series. Specifically, we compute the cosine similarity between the self-attention map and the correlation coefficients of patches, derived from the randomly shuffled patched series across channels. A Transformer encoder that effectively learns the dependencies of input patches should yield attention scores consistent with the correlation structure (Liu et al., 2024b). We compare ShuffleMTM with patch-based CI MTM methods, PatchTST and TimeSiam. Additionally, we evaluate PatchTST pre-trained to reconstruct the original time series from shuffled masked series, referred to as PatchTST-shuffled.

As shown in Figure 8, ShuffleMTM achieves a higher cosine similarity between the attention scores of shuffled patched series and their correlation matrix compared to other models. This finding suggests that pre-training with shuffled masked series effectively captures cross-channel dependence in the channel-independent encoding. Moreover, PatchTST-shuffled demonstrate lower average similarities and higher variances in both datasets than ShuffleMTM. This result suggests that processing shuffled masked series in a single-branch, channel-independent encoding is insufficient for learning cross-channel dependencies. These analyses validate the use of Siamese networks to better capture such dependencies.

**Channel-level dependence.** While ShuffleMTM processes each univariate time series in MTS independently, its embedding space captures the the cross-channel correlations present in the raw time series. To illustrate this channel-wise dependence, we present a case visualization on the correlation coefficients of channels in the raw time series and the pairwise distances of channel embeddings on the Traffic dataset, as shown in Figure 8. Max pooling is applied to extract channel embeddings from a series of patch embeddings, a technique commonly used in time series self-supervised methods (Nie et al., 2023; Dong et al., 2024b; Lee et al., 2024). The results show that the pairwise distances between the learned channel embeddings closely align with the correlations between channels in the raw time series. These findings confirm that pre-training with shuffled series captures the dependent structure of channels in the raw time series.

### 6.2 CAPACITY-ROBUSTNESS ANALYSIS.

Analysis from Han et al. (2024) revealed that the CI forecasting models have lower capacity but better robustness than the channel-dependent forecasting models. As ShuffleMTM combines the advantages of both channel-independent and channel-dependent models, we expect ShuffleMTM enhances forecasting capacity and robustness compared to CI models. Thus, we conduct a capacity-

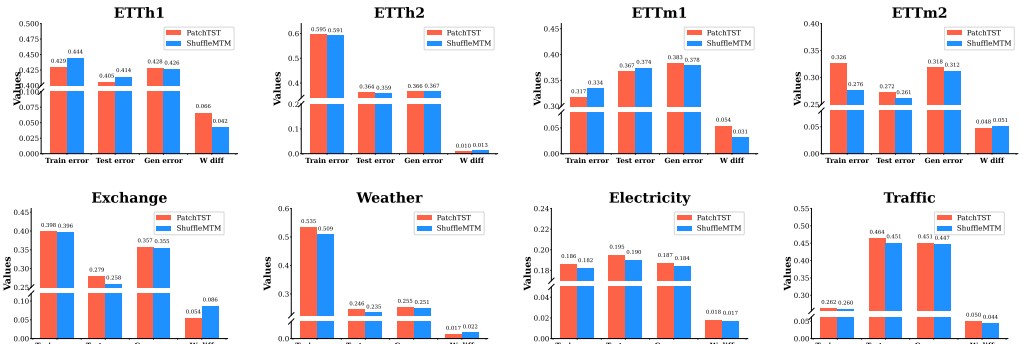

Figure 9: Capacity-robustness analysis. Train and test errors are capacity measures and Generalization error and W difference are robustness measures. The lower values indicate the higher capacity or robustness. ShuffleMTM achieves lower values on 12 out of 16 capacity measures and 11 out of 16 robustness measures across eight forecasting benchmarks.

robustness analysis based on the measures proposed in Han et al. (2024). Train error and test error measure the capacity, and generalization error and W difference measure the robustness of forecasting models. The lower values indicate the better capacity and robustness. For a fair comparison, we compare with PatchTST, a CI MTM that can be derived from ShuffleMTM. Since the measures relate to absolute forecasting performance, we set the model and training configurations for both models equally. The formulation and detailed experimental setup is explained in Appendix C.

As shown in Figure 9, ShuffleMTM consistently demonstrates greater capacity and robustness than PatchTST, achieving lower error values on 12 out of 16 capacity measures and 11 out of 16 robustness measures across eight forecasting benchmarks. While CI methods trade capacity for robust prediction, ShuffleMTM attains both by incorporating cross-channel information in a channel-independent encoding. Notably, ShuffleMTM achieves the strengths of both CI and channel-dependent approaches, despite not being explicitly pre-trained to enhance capacity and robustness. These analyses confirm the superior forecasting performance of ShuffleMTM compared to CI MTM methods.

In summary, capturing cross-channel dependence is crucial for MTS modeling, yet it is not addressed by existing MTM methods. ShuffleMTM is the first MTM framework to capture cross-channel dependence within the CI strategy, combining the advantages of temporal modeling from CI models and cross-channel modeling from channel-dependent models. We confirm that ShuffleMTM captures both fine-grained and coarse-grained dependencies across channels within the CI strategy (see Figure 8), enhancing both forecasting capacity and robustness—achieved by channel-dependent and CI forecasting models, respectively. Through ablations and analyses, we demonstrate that the shuffling method, Siamese encoders, and cross-attention decoder are crucial for extending the CI MTM task to model both cross-time and cross-channel dependencies effectively.

## 7 CONCLUSION

This paper proposes ShuffleMTM, a simple pre-training framework for masked modeling of multivariate time series. Unlike previous works that primarily focus on enhancing temporal modeling capacity within each channel, ShuffleMTM simultaneously captures both cross-time and cross-channel dependencies through its proposed masked series shuffling and Siamese encoders. Experimentally, ShuffleMTM demonstrates superior performance in time series forecasting and classification tasks compared to state-of-the-art masked modeling methods, across in-domain, cross-domain, and label-scarce settings. This work highlights the effectiveness of incorporating cross-channel dependencies during pre-training, paving the way for various future studies. For example, we aim to develop fine-tuning methods applicable to channel-independent encoders, further enhancing adaptability to diverse cross-channel patterns during fine-tuning and inference. Additionally, we plan to extend our approach to time series foundation models, which are crucial for capturing cross-time and cross-channel dependencies necessary for time series forecasting tasks.

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

## A  Dataset Description

We perform extensive experiments using 10 well-established datasets, targeting two primary tasks in time series analysis: forecasting and classification. These datasets span a broad spectrum of application domains, encompassing various signal types, channel dimensions, time series lengths, and data scales. This variety allows us to assess the generalizability of the proposed approach to complex real-world datasets.

### A.1  Time Series Forecasting

We evaluate the forecasting performance using seven datasets: ETT (Zhou et al., 2021), Weather (Wetterstation), Electricity (UCI), Traffic (PeMS), and Exchange (Lai et al., 2018). The ETT datasets, consisting of two hourly and two quarter-hourly datasets, contain two years of oil temperature and power load data from electricity transformers. The Weather dataset records 21 meteorological variables every 10 minutes. The Electricity dataset contains hourly electricity consumption data for 321 clients. The Traffic dataset tracks hourly road occupancy from 862 sensors across San Francisco Bay Area freeways. Lastly, the Exchange dataset records daily exchange rates for eight countries. For the experimental setup, we follow the standard setting from Wu et al. (2021), which splits datasets into training, validation, and test sets in chronological order. The splitting ratios are set at 6:2:2 for ETT datasets and 7:1:2 for the other datasets. For the all forecasting experiments, we fix the look-back horizon as $L = 96$ for a fair comparison. A detailed description of each dataset is summarized in Table 5.

| Datasets | Channels | Time steps | Information | Frequency |
|----------|----------|------------|-------------|-----------|
| ETT (h1,h2) | 7 | 17420 | Electricity | Hourly |
| ETT (m1,m2) | 7 | 69680 | Electricity | 15 Mins |
| Exchange | 8 | 7588 | Exchange rate | Daily |
| Weather | 21 | 52696 | Weather | 10 Mins |
| Electricity | 321 | 26304 | Electricity | Hourly |
| Traffic | 862 | 17544 | Transportation | Hourly |

Table 5: Dataset description for time series forecasting.

### A.2  Time Series Classification

We utilize two representative datasets within the medical domain: AD (Escudero et al., 2006), and PTB (Goldberger et al., 2000). The AD dataset consists of 5967 EEG recordings from 12 Alzheimer's patients and 11 healthy individuals, with each trial spanning 5 seconds across 16 channels at a sampling rate of 256Hz. The data is standardized and divided into nine overlapping 1-second segments. A binary label based on whether the patient has Alzheimer's disease is assigned to each sample. In the PTB dataset, 62370 ECG recordings from 198 patients (comprising Myocardial infarction cases and healthy controls) are captured across 15 channels at 1000Hz. The signals are down-sampled to 250Hz, normalized, and divided into heartbeat segments based on R-peak intervals. For the benchmark selection, we exclude the classification datasets with a single channel, which do not have interactions across channels. We follow the pre-processing procedure and evaluation setup described in Wang et al. (2024), which splits training, validation, and test sets in a patient-independent way. The detailed descriptions of datasets are summarized in Table 6.

## B  Implementation Details

### B.1  Baselines Implementation

**Time series forecasting** For the forecasting task, we compare our proposed ShuffleMTM to ten state-of-the-art baselines, including four self-supervised pre-training methods and six supervised

| Datasets | Channels | Length | Classes | Information | Frequency |
|----------|----------|--------|---------|-------------|-----------|
| AD | 16 | 256 | 2 | EEG | 256 Hz |
| PTB | 15 | 300 | 2 | ECG | 1000 Hz |

Table 6: Dataset description for time series classification.

methods. We implemented the baselines using their official implementations and followed the configurations from the original papers as closely as possible. For datasets not included in the original papers, we explored various configurations by adjusting key hyperparameters and reported the best performance. We conducted the experiments five times and report the average performance.

**Time series classification** The baselines for the classification task include five state-of-the-art MTM methods: COMET (Wang et al., 2024), SimMTM (Dong et al., 2024b), PITS (Lee et al., 2024), PatchTST (Nie et al., 2023), and TimeSiam (Dong et al., 2024a). To ensure a fair comparison, we implemented their official code and hyperparameters from the original papers wherever possible. In cases where the optimal configuration was not provided, we conducted a hyperparameter search for key parameters and reported the best results.

Specifically, for SimMTM, which was not validated on the AD and PTB datasets in the original papers, we adjusted the encoder's hidden dimensions {64, 128}. For PITS, we similarly searched for the best hidden dimension {128, 256 } and patch length {8, 16} on the AD dataset and {16, 30} on the PTB dataset. For PatchTST, which was not validated for classification tasks, we adopted the evaluation protocol of PITS. Although TimeSiam originally uses temporal convolutional networks as the encoder for classification, we utilized the PatchTST encoder to demonstrate improvements over the CI MTM baselines. We adjusted the hidden dimensions of the PatchTST-backbone TimeSiam {128, 256} and patch length {8, 16} on the AD dataset and {16, 30} on the PTB dataset. Lastly, we implemented the official code of COMET. We conducted the experiment five times and report the average performance.

## B.2 MODEL CONFIGURATION

ShuffleMTM includes two main sets of model hyperparameters: patch length and the Transformer encoder hyperparameters. Depending on the task and the size of the datasets (i.e., small, medium, and large), we pre-define the set of model hyperparameters and determine the best configuration on a pre-defined validation dataset. The candidate sets of model hyperparameters are summarized in Table 7.

| Task | Size | Dataset | Architecture | | | | | |
|------|------|---------|--------------|---------------|---------------|-----------------|------------------|----------|
| | | | Patch length ($P$) | encoder depth | decoder depth | Number of heads | Hidden dim. ($d_m$) | FFW dim. |
| Forecasting | Small | ETTh1 ETTh2 Exchange | {6, 8} | 3 | 1 | {8, 16} | {16, 32} | {32, 64} |
| | Medium | ETTm1 ETTm2 Weather | | | | | {64, 128} | {64, 128} |
| | Large | Electricity Traffic | | | | | 256 | 512 |
| Classification | - | AD PTB | {32, 64} {15, 30} | 3 | 1 | {8, 16} | {64, 128} {64, 128, 256} | {64, 128} {128, 256, 512} |

Table 7: Model configuration for Forecasting and Classification tasks.

## B.3 TRAINING CONFIGURATION

In self-supervised pre-training, we set the pre-training epochs to 100 and search for the pre-training learning rate in {1e-4, 5e-4} for forecasting and {5e-4, 1e-3} for classification. We also explore the mask ratio depending on the task: 0.4 for forecasting and {0.4, 0.8} for classification. During fine-tuning, we adopt the freeze-and-fine-tune strategy. In this strategy, we apply linear probing for $n$ epochs to update the downstream head while keeping the backbone frozen. Subsequently, we perform full fine-tuning of the entire network for $2n$ epochs. This two-step fine-tuning has

been shown to be effective in Nie et al. (2023) and Lee et al. (2024). For the classification, we aggregate patch representations by max-pooling over patches in each channel to generate the global representation of the sample. The candidate sets of training hyperparameters are summarized in Table 8.

| Task | Size | Dataset | Pre-training | | | | Fine-tuning | | | |
|------|------|---------|--------------|---------------|------------|--------|---------------|---------------|------------|-------------------|
| | | | Mask ratio | Learning rate | Batch size | Epochs | Learning rate | Loss function | Batch size | Epochs (LP / FT) |
| Forecasting | Small | ETTh1 ETTh2 Exchange | 0.4 | {1e-4, 5e-4} | 64 | 100 | 5e-4 | L2 | 64 | 10 / 20 |
| | Medium | ETTm1 ETTm2 Weather | | | | | | | | |
| | Large | Electricity Traffic | | | 32 | | | | 32 | |
| Classification | - | AD PTB | 0.4 {0.4, 0.8} | {5e-4, 1e-3} | 256 | 100 | {5e-4, 1e-3} | Cross-Entropy | 128 | {20 / 40, 30 / 60} |

Table 8: Training configuration for forecasting and classification tasks. Epochs (LP / FT) indicates epochs for linear probing and end-to-end fine-tuning.

## C CAPACITY-ROBUSTNESS ANALYSIS

### C.1 DEFINITION OF MEASURES

Han et al. (2024) proposes four measures, with two evaluating capacity and two evaluating robustness of a linear model. We extend these measures to a neural network forecaster. We denote the training and test sets for forecasting as $(X^{(tr)}, Y^{(tr)})$ and $(X^{(te)}, Y^{(te)})$, respectively, and the neural network forecaster as $f_\theta$, parametrized by $\theta$. The training and test sets can be decomposed into univariate time series if the forecaster adopts the channel-independent strategy. We compute the following measures to evaluate the model's performance:

**Train Error (Incapacity):** The train error is defined as:
$$L^{(tr)} = ||f_{\theta^{(tr)}}(X^{(tr)}) - Y^{(tr)}||_F^2$$

where
$$\theta^{(tr)} = argmin_\theta||f_\theta(X^{(tr)}) - Y^{(tr)}||_F^2$$

is the optimal parameter for the training data. Training error measures the capacity computed on the training set.

**Test Error (Incapacity):** The test error is defined as:
$$L^{(te)} = ||f_{\theta^{(te)}}(X^{(te)}) - Y^{(te)}||_F^2$$

where
$$\theta^{(te)} = argmin_\theta||f_\theta(X^{(te)}) - Y^{(te)}||_F^2$$

is the optimal parameter for the test data. Test error indicates the best error a forecaster can achieve on the test data, which measures the capacity computed on the test set.

**Generalization Error (Non-robustness):** The generalization error is defined as:
$$L^{(gen)} = ||f_{\theta^{(tr)}}(X^{(te)}) - Y^{(te)}||_F^2.$$

It is the performance measure on the benchmark evaluation, which represents the test set errors of a forecaster that achieves the lowest training error.

**W Difference (Non-robustness):** The W difference is computed as:
$$\text{Diff}_\theta = ||f_{\theta^{(tr)}}(X^{(te)}) - f_{\theta^{(te)}}(X^{(te)})||_F^2$$

It computes the mean squared error between best-train-error forecaster prediction and best-test-error forecaster prediction on the test dataset, where each forecaster is trained in calculating train and test erors.

For all measures, the lower value indicates the greater capacity and robustness.

## C.2 EXPERIMENTAL SETUP

To demonstrate effectiveness over CI models, we compare with PatchTST. PatchTST can be regarded as a single-branch version of ShuffleMTM, as described in Section 3.4, and both models share the same encoder architecture. Therefore, we select PatchTST for comparison. To exclude the effects of different pre-training tasks, we unify the pre-training task for both models: reconstructing the original time series. Since the measures relate to absolute forecasting performance, we set the model and training configurations equally for both models. We pre-train each model for 100 epochs, then optimize the encoder to obtain $\theta^{(tr)}$ and $\theta^{(te)}$ through 10 epochs of linear probing and 20 epochs of end-to-end fine-tuning, as conducted in the main experiment.

## D CLASSIFICATION EMBEDDING VISUALIZATION

To visualize the effectiveness of ShuffleMTM in the classification task, we present a case visualization of the learned embeddings on the AD dataset. To represent the embeddings more intuitively, we use UMAP, a dimensionality reduction method, with 80 neighbors and a minimum distance of 1. For comparison, we use TimeSiam, which has shown the best classification performance among all MTM baselines. Additionally, we compute the average pairwise Euclidean distance in the UMAP embedding space between the negative (healthy) and positive (Alzheimer's) classes. As shown in Figure 10, ShuffleMTM embeddings are more clustered within each class. The average pairwise distance between classes for ShuffleMTM is also greater than for TimeSiam, indicating better class separability.

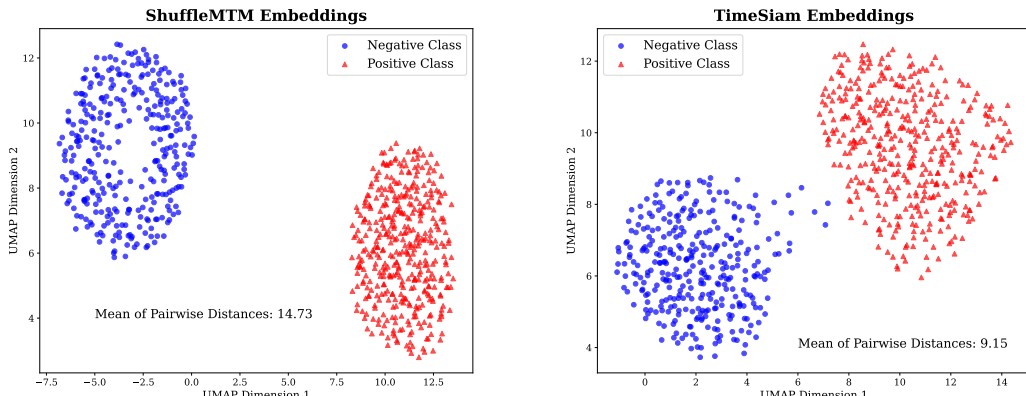

Figure 10: Comparison of learned embeddings from ShuffleMTM and TimeSiam on the AD Dataset. (**Left**) Visualization for ShuffleMTM. (**Right**) Visualization for TimeSiam. We calculate the mean Euclidean distance between pairwise samples from the two classes to assess class separability. Comparing the figures and distances, ShuffleMTM shows a larger gap between classes than TimeSiam.

## E SIMULATION EXPERIMENT

### E.1 SYNTHETIC DATASETS

We conduct simulation experiments to examine the cross-channel dependence that ShuffleMTM captures. We generate two synthetic datasets with different cross-channel dependencies. As illustrated in Figure 11, the first synthetic dataset consists of three channels, each of which exhibits lagged structures relative to the others. We generate the first channel as sequence of length-16 patches, each representing a sinusoidal function with a unique frequency. Then, the second and third channels are derived by shifting the first channel by one-patch and two-patches lengths, respectively. From this simulation, this dataset naturally exhibits apparent patch-level dependencies between channels due to the lagged relationship, which are prevalent in real-world multivariate time series (Zhao & Shen, 2024). As the attention values of Transformers for multivariate time series forecasting tend to segment, i.e., close data points have similar attention weights, it is important to capture patch-level dependencies both within and across channels (Zhang & Yan, 2023).

Figure 11: Cross-channel dependency structure of two synthetic datasets. The first synthetic dataset exhibits patch-level dependencies across channels with the lagged structure, while the channels in the second synthetic dataset share long-term temporal dependencies.

| Dataset | Pred_len | ShuffleMTM | | PatchTST | |
|---|---|---|---|---|---|
| | | MSE | MAE | MSE | MAE |
| Synthetic 1 | 32 | **1.023** | 0.896 | 1.030 | **0.895** |
| | 48 | **1.024** | **0.897** | 1.037 | 0.901 |
| | 96 | **1.026** | **0.898** | 1.033 | 0.901 |
| | 192 | **1.026** | **0.899** | 1.031 | 0.901 |
| Synthetic 2 | 32 | 0.292 | 0.477 | **0.284** | **0.470** |
| | 48 | 0.292 | 0.477 | **0.288** | **0.474** |
| | 96 | 0.294 | 0.479 | **0.292** | **0.478** |
| | 192 | 0.296 | 0.480 | **0.294** | **0.479** |

Table 9: Comparison between ShuffleMTM and PatchTST on two synthetic datasets.

| Dataset | Pred_len | ShuffleMTM | | ShuffleMTM w/o shuffle | |
|---|---|---|---|---|---|
| | | MSE | MAE | MSE | MAE |
| Synthetic 1 | 32 | **1.023** | **0.896** | 1.034 | 0.899 |
| | 48 | **1.024** | **0.897** | 1.034 | 0.899 |
| | 96 | **1.026** | 0.898 | 1.033 | 0.898 |
| | 192 | 1.026 | 0.899 | 1.029 | **0.897** |
| Synthetic 2 | 32 | **0.292** | **0.477** | 0.293 | 0.479 |
| | 48 | **0.292** | **0.477** | 0.293 | 0.479 |
| | 96 | **0.294** | **0.479** | 0.295 | **0.479** |
| | 192 | **0.296** | 0.480 | **0.296** | 0.480 |

Table 10: Comparison between ShuffleMTM and the ShuffleMTM without the shuffled view on two synthetic datasets.

The second synthetic dataset also consists of three channels, all of which share the same long-term trend. First, we randomly generate three sequences of length-16 patches of sinusoidal function with distinct frequencies, ensuring no overlaps of local patterns between patches. Next, we generate a low-frequency sinusoidal waveform spanning the whole time series length as a long-term trend. Then, we add each sequence of length-16 patches to the long-term trend to get three channels that share the same trend. In this synthetic dataset, each time step in one channel is dependent on the previous time step as it has a long-term trend but is not dependent on current time step's information in other channels as it does not share the local sinusoidal patterns.

### E.2 COMPARISON WITH CHANNEL-INDEPENDENT MTM

We evaluate the forecasting performance of ShuffleMTM and PatchTST on two synthetic datasets exhibiting different cross-channel dependencies. Table 9 presents the forecasting performance of ShuffleMTM and PatchTST on two synthetic datasets. We used the same model configuration for both models to ensure a fair comparison. Both models are evaluated in forecasting scenarios with prediction lengths of {32, 64, 96, 192} and a fixed input length of 96, with the patch length set to 16. The results indicate that ShuffleMTM consistently outperforms PatchTST across all prediction lengths on the first dataset, demonstrating its ability to capture patch-level dependencies between channels with lagged structures. This analysis confirms that ShuffleMTM can capture fine-grained cross-channel dependencies. However, ShuffleMTM exhibits in Table 10 that greater forecasting errors than PatchTST on the second dataset. Since each channel in the second dataset contains long-term trends and lacks local dependencies on other channels, ShuffleMTM is less effective for short-term forecasting. However, as prediction lengths increase, the performance difference between the two models decreases. As the channels share the same long-term context, ShuffleMTM effectively captures the long-term dependence, resulting in enhanced performance in long-term forecasting.

In summary, ShuffleMTM effectively captures fine-grained patch-level dependencies between channels such as lagged dependencies, as shown in the analysis on the first synthetic dataset. In time series that shares global temporal patterns, ShuffleMTM is ineffective in short-term forecasting, if each channel is not dependent on the others in a local context. However, ShuffleMTM becomes effective in long-term forecasting on this data if the channels share the long-term context.

### E.3 COMPARISON WITH THE SHUFFLEMTM WITHOUT THE SHUFFLED VIEW

We evaluate the performance of ShuffleMTM on these synthetic datasets after removing the shuffled view and using the original view as the query, key, and value in the decoder. This variant of

ShuffleMTM in fact reduces to PatchTST with a self-attention decoder, which originally utilizes the original masked series and decodes the representation with a linear layer. Comparing this variant with ShuffleMTM demonstrates the effectiveness of utilizing the shuffled masked series for pre-training channel-independent MTM. We denote this model variant as ShuffleMTM w/o shuffle.

Table 10 presents the forecasting performance of ShuffleMTM and its variant without the shuffled view on these two synthetic datasets. ShuffleMTM outperformed its variant across both datasets, with a larger performance gap observed in the first dataset than in the second. These experiments suggest that the shuffling method enables the channel-independent encoder to capture fine-grained dependencies between channels. However, the shuffling method is less effective on datasets with weak local dependencies compared to those with strong local dependencies.

# F FULL EXPERIMENTAL RESULTS

Due to space constraints, we present the full experiments for time series forecasting and classification, including mean and standard deviations under five random seeds, for both in-domain and cross-domain scenarios.

Table 11: In-domain forecasting. All models are both pre-trained and fine-tuned on the same dataset. The best results are in bold, and the second-best results are underlined.

| | | Self-supervised | | | | | | | | | | Supervised | | | | | | | | | | | |
| | | ShuffleMTM | | TimeSiam | | PITS | | PatchTST | | SimMTM | | iTransformer | | Crossformer | | CrossGNN | | MTGNN | | PatchTST (sup) | | DLinear | |
| Models | Metrics | MSE | MAE | MSE | MAE | MSE | MAE | MSE | MAE | MSE | MAE | MSE | MAE | MSE | MAE | MSE | MAE | MSE | MAE | MSE | MAE | MSE | MAE |
|---|---|---|---|---|---|---|---|---|---|---|---|---|---|---|---|---|---|---|---|---|---|---|---|
| ETTh1 | 96 | 0.376 | 0.397 | 0.379 | 0.402 | 0.377 | 0.395 | 0.379 | 0.399 | 0.367 | 0.389 | 0.386 | 0.405 | 0.391 | 0.418 | 0.389 | 0.399 | 0.515 | 0.517 | 0.392 | 0.407 | 0.390 | 0.404 |
| | 192 | 0.420 | 0.425 | 0.425 | 0.431 | 0.430 | 0.425 | 0.425 | 0.427 | 0.424 | 0.423 | 0.443 | 0.437 | 0.450 | 0.453 | 0.441 | 0.430 | 0.553 | 0.522 | 0.445 | 0.434 | 0.451 | 0.446 |
| | 336 | 0.456 | 0.446 | 0.459 | 0.451 | 0.478 | 0.458 | 0.470 | 0.446 | 0.473 | 0.456 | 0.489 | 0.460 | 0.526 | 0.503 | 0.484 | 0.452 | 0.612 | 0.577 | 0.483 | 0.451 | 0.498 | 0.474 |
| | 720 | 0.474 | 0.471 | 0.475 | 0.478 | 0.499 | 0.475 | 0.482 | 0.466 | 0.494 | 0.493 | 0.508 | 0.494 | 0.643 | 0.593 | 0.483 | 0.472 | 0.609 | 0.597 | 0.477 | 0.469 | 0.511 | 0.505 |
| ETTh2 | 96 | 0.288 | 0.338 | 0.291 | 0.343 | 0.289 | 0.338 | 0.299 | 0.346 | 0.299 | 0.347 | 0.301 | 0.351 | 0.683 | 0.562 | 0.295 | 0.345 | 0.384 | 0.454 | 0.299 | 0.346 | 0.378 | 0.413 |
| | 192 | 0.368 | 0.390 | 0.370 | 0.393 | 0.374 | 0.393 | 0.382 | 0.398 | 0.383 | 0.398 | 0.379 | 0.399 | 0.824 | 0.628 | 0.383 | 0.400 | 0.457 | 0.464 | 0.382 | 0.398 | 0.447 | 0.452 |
| | 336 | 0.412 | 0.426 | 0.414 | 0.428 | 0.415 | 0.426 | 0.427 | 0.433 | 0.420 | 0.430 | 0.422 | 0.432 | 0.966 | 0.701 | 0.427 | 0.440 | 0.515 | 0.540 | 0.427 | 0.433 | 0.515 | 0.497 |
| | 720 | 0.421 | 0.441 | 0.422 | 0.443 | 0.423 | 0.443 | 0.438 | 0.452 | 0.425 | 0.444 | 0.435 | 0.450 | 1.395 | 0.853 | 0.436 | 0.453 | 0.532 | 0.576 | 0.438 | 0.452 | 0.688 | 0.593 |
| ETTm1 | 96 | 0.317 | 0.357 | 0.317 | 0.359 | 0.332 | 0.363 | 0.318 | 0.356 | 0.327 | 0.365 | 0.343 | 0.377 | 0.360 | 0.395 | 0.345 | 0.372 | 0.379 | 0.446 | 0.321 | 0.359 | 0.346 | 0.372 |
| | 192 | 0.361 | 0.382 | 0.357 | 0.383 | 0.366 | 0.384 | 0.356 | 0.380 | 0.368 | 0.389 | 0.381 | 0.394 | 0.390 | 0.410 | 0.379 | 0.388 | 0.470 | 0.428 | 0.362 | 0.382 | 0.383 | 0.393 |
| | 336 | 0.390 | 0.402 | 0.386 | 0.403 | 0.396 | 0.406 | 0.385 | 0.403 | 0.396 | 0.409 | 0.419 | 0.419 | 0.452 | 0.419 | 0.410 | 0.408 | 0.473 | 0.430 | 0.393 | 0.403 | 0.415 | 0.416 |
| | 720 | 0.446 | 0.435 | 0.444 | 0.438 | 0.457 | 0.441 | 0.444 | 0.439 | 0.452 | 0.440 | 0.490 | 0.458 | 0.542 | 0.516 | 0.469 | 0.441 | 0.553 | 0.479 | 0.453 | 0.437 | 0.475 | 0.454 |
| ETTm2 | 96 | 0.175 | 0.259 | 0.176 | 0.259 | 0.177 | 0.261 | 0.176 | 0.262 | 0.186 | 0.276 | 0.184 | 0.269 | 0.184 | 0.353 | 0.179 | 0.259 | 0.203 | 0.299 | 0.178 | 0.260 | 0.188 | 0.282 |
| | 192 | 0.240 | 0.302 | 0.241 | 0.303 | 0.244 | 0.304 | 0.242 | 0.306 | 0.253 | 0.317 | 0.252 | 0.313 | 0.379 | 0.432 | 0.243 | 0.302 | 0.265 | 0.328 | 0.245 | 0.304 | 0.269 | 0.343 |
| | 336 | 0.299 | 0.340 | 0.302 | 0.342 | 0.304 | 0.342 | 0.304 | 0.346 | 0.317 | 0.356 | 0.315 | 0.352 | 0.520 | 0.535 | 0.304 | 0.342 | 0.365 | 0.374 | 0.307 | 0.343 | 0.351 | 0.400 |
| | 720 | 0.399 | 0.398 | 0.400 | 0.398 | 0.400 | 0.397 | 0.406 | 0.405 | 0.417 | 0.412 | 0.412 | 0.406 | 1.453 | 0.875 | 0.405 | 0.399 | 0.461 | 0.459 | 0.406 | 0.401 | 0.492 | 0.484 |
| ETTm2/Exchange | 96 | 0.082 | 0.199 | 0.085 | 0.199 | 0.084 | 0.200 | 0.085 | 0.203 | 0.087 | 0.208 | 0.087 | 0.207 | 0.256 | 0.453 | 0.084 | 0.200 | 0.102 | 0.228 | 0.088 | 0.205 | 0.088 | 0.218 |
| Exchange | 192 | 0.173 | 0.296 | 0.182 | 0.304 | 0.182 | 0.297 | 0.182 | 0.302 | 0.180 | 0.303 | 0.179 | 0.302 | 0.531 | 0.554 | 0.177 | 0.296 | 0.267 | 0.335 | 0.176 | 0.297 | 0.176 | 0.315 |
| | 336 | 0.324 | 0.411 | 0.334 | 0.419 | 0.340 | 0.421 | 0.331 | 0.415 | 0.330 | 0.417 | 0.335 | 0.420 | 0.886 | 0.732 | 0.340 | 0.418 | 0.393 | 0.457 | 0.344 | 0.424 | 0.313 | 0.427 |
| | 720 | 0.833 | 0.687 | 0.866 | 0.702 | 0.855 | 0.696 | 0.869 | 0.696 | 0.852 | 0.696 | 0.853 | 0.697 | 1.571 | 1.016 | 1.017 | 0.761 | 1.090 | 0.811 | 0.904 | 0.716 | 0.839 | 0.695 |
| Weather | 96 | 0.168 | 0.211 | 0.173 | 0.215 | 0.184 | 0.223 | 0.170 | 0.212 | 0.170 | 0.216 | 0.175 | 0.215 | 0.158 | 0.230 | 0.180 | 0.259 | 0.230 | 0.329 | 0.178 | 0.219 | 0.199 | 0.260 |
| | 192 | 0.215 | 0.253 | 0.222 | 0.257 | 0.229 | 0.262 | 0.215 | 0.253 | 0.216 | 0.255 | 0.225 | 0.258 | 0.206 | 0.298 | 0.227 | 0.302 | 0.263 | 0.322 | 0.225 | 0.259 | 0.237 | 0.294 |
| | 336 | 0.271 | 0.293 | 0.273 | 0.294 | 0.283 | 0.300 | 0.273 | 0.294 | 0.271 | 0.294 | 0.281 | 0.299 | 0.272 | 0.335 | 0.282 | 0.342 | 0.354 | 0.396 | 0.278 | 0.298 | 0.283 | 0.332 |
| | 720 | 0.348 | 0.343 | 0.353 | 0.346 | 0.357 | 0.349 | 0.348 | 0.345 | 0.346 | 0.343 | 0.359 | 0.350 | 0.398 | 0.386 | 0.360 | 0.399 | 0.409 | 0.371 | 0.351 | 0.347 | 0.347 | 0.383 |
| Electricity | 96 | 0.161 | 0.248 | 0.161 | 0.247 | 0.187 | 0.269 | 0.174 | 0.258 | 0.198 | 0.291 | 0.148 | 0.240 | 0.226 | 0.308 | 0.199 | 0.279 | 0.217 | 0.318 | 0.166 | 0.252 | 0.195 | 0.278 |
| | 192 | 0.170 | 0.257 | 0.171 | 0.257 | 0.190 | 0.274 | 0.183 | 0.267 | 0.200 | 0.293 | 0.164 | 0.256 | 0.276 | 0.339 | 0.198 | 0.281 | 0.238 | 0.352 | 0.174 | 0.260 | 0.194 | 0.280 |
| | 336 | 0.188 | 0.274 | 0.187 | 0.274 | 0.206 | 0.289 | 0.198 | 0.283 | 0.215 | 0.308 | 0.179 | 0.272 | 0.357 | 0.393 | 0.213 | 0.296 | 0.260 | 0.348 | 0.190 | 0.277 | 0.207 | 0.296 |
| | 720 | 0.228 | 0.310 | 0.226 | 0.308 | 0.247 | 0.322 | 0.238 | 0.316 | 0.258 | 0.340 | 0.211 | 0.300 | 0.406 | 0.422 | 0.253 | 0.329 | 0.290 | 0.369 | 0.231 | 0.312 | 0.243 | 0.329 |
| Traffic | 96 | 0.424 | 0.220 | 0.431 | 0.278 | 0.555 | 0.349 | 0.475 | 0.307 | 0.542 | 0.342 | 0.393 | 0.268 | 0.549 | 0.311 | 0.657 | 0.390 | 0.660 | 0.437 | 0.445 | 0.283 | 0.649 | 0.397 |
| | 192 | 0.437 | 0.274 | 0.443 | 0.282 | 0.536 | 0.339 | 0.481 | 0.308 | 0.530 | 0.334 | 0.413 | 0.277 | 0.565 | 0.315 | 0.608 | 0.370 | 0.649 | 0.438 | 0.453 | 0.285 | 0.599 | 0.371 |
| | 336 | 0.451 | 0.281 | 0.457 | 0.288 | 0.546 | 0.338 | 0.491 | 0.309 | 0.541 | 0.338 | 0.424 | 0.283 | 0.580 | 0.323 | 0.617 | 0.373 | 0.653 | 0.472 | 0.468 | 0.291 | 0.606 | 0.374 |
| | 720 | 0.483 | 0.300 | 0.489 | 0.307 | 0.582 | 0.359 | 0.524 | 0.328 | 0.578 | 0.355 | 0.457 | 0.300 | 0.589 | 0.319 | 0.659 | 0.390 | 0.639 | 0.437 | 0.501 | 0.310 | 0.646 | 0.396 |

| Models | ShuffleMTM | | TimeSiam | | PITS | | PatchTST | | SimMTM | |
|---|---|---|---|---|---|---|---|---|---|---|
| Metrics | MSE | MAE | MSE | MAE | MSE | MAE | MSE | MAE | MSE | MAE |
| ETTh1→ETTh2 | $\underline{0.375}_{(\pm0.001)}$ | $\mathbf{0.400}_{(\pm0.001)}$ | $\mathbf{0.374}_{(\pm0.002)}$ | $\underline{0.401}_{(\pm0.004)}$ | $0.376_{(\pm0.002)}$ | $\underline{0.401}_{(\pm0.002)}$ | $0.381_{(\pm0.003)}$ | $0.406_{(\pm0.002)}$ | $0.382_{(\pm0.003)}$ | $0.407_{(\pm0.002)}$ |
| ETTh2→ETTh1 | $\mathbf{0.434}_{(\pm0.001)}$ | $\mathbf{0.435}_{(\pm0.001)}$ | $0.444_{(\pm0.003)}$ | $0.450_{(\pm0.001)}$ | $0.442_{(\pm0.001)}$ | $\underline{0.436}_{(\pm0.001)}$ | $\underline{0.438}_{(\pm0.003)}$ | $0.437_{(\pm0.001)}$ | $0.445_{(\pm0.002)}$ | $0.446_{(\pm0.001)}$ |
| ETTh1→ETTm1 | $\mathbf{0.380}_{(\pm0.001)}$ | $\mathbf{0.393}_{(\pm0.001)}$ | $0.395_{(\pm0.001)}$ | $0.404_{(\pm0.001)}$ | $0.387_{(\pm0.001)}$ | $0.398_{(\pm0.001)}$ | $0.386_{(\pm0.002)}$ | $\underline{0.395}_{(\pm0.001)}$ | $\underline{0.383}_{(\pm0.003)}$ | $0.398_{(\pm0.003)}$ |
| ETTh2→ETTm1 | $\mathbf{0.381}_{(\pm0.001)}$ | $\mathbf{0.394}_{(\pm0.001)}$ | $0.399_{(\pm0.002)}$ | $0.411_{(\pm0.001)}$ | $0.387_{(\pm0.001)}$ | $0.398_{(\pm0.001)}$ | $\underline{0.385}_{(\pm0.001)}$ | $\underline{0.396}_{(\pm0.001)}$ | $0.456_{(\pm0.017)}$ | $0.428_{(\pm0.007)}$ |
| ETTm2→ETTm1 | $\mathbf{0.378}_{(\pm0.001)}$ | $\mathbf{0.396}_{(\pm0.002)}$ | $0.395_{(\pm0.004)}$ | $0.404_{(\pm0.002)}$ | $0.387_{(\pm0.002)}$ | $\underline{0.398}_{(\pm0.001)}$ | $\underline{0.379}_{(\pm0.004)}$ | $\mathbf{0.396}_{(\pm0.002)}$ | $0.396_{(\pm0.004)}$ | $0.402_{(\pm0.003)}$ |
| ETTm2→ETTh1 | $\underline{0.437}_{(\pm0.003)}$ | $\mathbf{0.433}_{(\pm0.003)}$ | $\mathbf{0.434}_{(\pm0.005)}$ | $0.441_{(\pm0.003)}$ | $0.439_{(\pm0.002)}$ | $\underline{0.437}_{(\pm0.001)}$ | $0.443_{(\pm0.007)}$ | $0.443_{(\pm0.003)}$ | $0.457_{(\pm0.001)}$ | $0.452_{(\pm0.001)}$ |
| Weather→ETTh1 | $\underline{0.436}_{(\pm0.004)}$ | $0.441_{(\pm0.003)}$ | $0.438_{(\pm0.004)}$ | $0.443_{(\pm0.006)}$ | $0.439_{(\pm0.002)}$ | $\underline{0.433}_{(\pm0.001)}$ | $0.441_{(\pm0.003)}$ | $0.440_{(\pm0.002)}$ | $\mathbf{0.426}_{(\pm0.004)}$ | $\mathbf{0.435}_{(\pm0.003)}$ |
| Weather→ETTm1 | $\mathbf{0.378}_{(\pm0.001)}$ | $\mathbf{0.395}_{(\pm0.001)}$ | $0.387_{(\pm0.003)}$ | $0.403_{(\pm0.001)}$ | $0.387_{(\pm0.001)}$ | $0.398_{(\pm0.001)}$ | $\underline{0.380}_{(\pm0.001)}$ | $\underline{0.396}_{(\pm0.001)}$ | $0.385_{(\pm0.003)}$ | $0.399_{(\pm0.001)}$ |

Table 12: Cross-domain forecasting. All models are pre-trained on source dataset and fine-tuned on target dataset.

| Dataset | Model | Accuracy | Precision | Recall | F1 score |
|---|---|---|---|---|---|
| AD | ShuffleMTM | $\mathbf{93.93}_{(\pm1.14)}$ | $\mathbf{93.82}_{(\pm1.51)}$ | $\mathbf{94.17}_{(\pm1.24)}$ | $\mathbf{93.90}_{(\pm0.98)}$ |
| | TimeSiam | $89.69_{(\pm2.61)}$ | $89.73_{(\pm2.69)}$ | $89.51_{(\pm2.52)}$ | $59.55_{(\pm2.63)}$ |
| | PITS | $76.90_{(\pm6.28)}$ | $82.85_{(\pm3.37)}$ | $76.90_{(\pm6.28)}$ | $76.56_{(\pm6.62)}$ |
| | PatchTST | $62.65_{(\pm12.09)}$ | $65.57_{(\pm13.08)}$ | $62.65_{(\pm12.09)}$ | $61.07_{(\pm13.16)}$ |
| | SimMTM | $66.98_{(\pm6.43)}$ | $75.03_{(\pm1.30)}$ | $69.67_{(\pm5.48)}$ | $65.56_{(\pm8.08)}$ |
| | COMET | $\underline{91.11}_{(\pm3.16)}$ | $\underline{92.39}_{(\pm2.19)}$ | $\underline{89.89}_{(\pm4.08)}$ | $\underline{92.10}_{(\pm5.23)}$ |
| PTB | ShuffleMTM | $\mathbf{91.58}_{(\pm1.58)}$ | $\mathbf{91.82}_{(\pm2.25)}$ | $86.91_{(\pm2.22)}$ | $\mathbf{88.90}_{(\pm2.36)}$ |
| | TimeSiam | $\underline{90.09}_{(\pm0.28)}$ | $\underline{92.24}_{(\pm0.81)}$ | $83.17_{(\pm1.12)}$ | $83.32_{(\pm0.65)}$ |
| | PITS | $87.57_{(\pm1.25)}$ | $90.16_{(\pm1.94)}$ | $84.06_{(\pm4.26)}$ | $81.79_{(\pm2.25)}$ |
| | PatchTST | $90.36_{(\pm2.50)}$ | $90.51_{(\pm2.64)}$ | $\mathbf{88.84}_{(\pm5.68)}$ | $\underline{86.98}_{(\pm3.55)}$ |
| | SimMTM | $84.49_{(\pm0.91)}$ | $83.99_{(\pm1.45)}$ | $75.64_{(\pm1.16)}$ | $78.28_{(\pm1.29)}$ |
| | COMET | $87.37_{(\pm1.40)}$ | $87.38_{(\pm2.77)}$ | $81.13_{(\pm3.67)}$ | $83.03_{(\pm2.33)}$ |

Table 13: In-domain classification. All models are both pre-trained and fine-tuned on the same dataset.

| Fraction | Models | AD | | | | PTB | | | |
|---|---|---|---|---|---|---|---|---|---|
| | | Accuracy | Precision | Recall | F1 score | Accuracy | Precision | Recall | F1 score |
| 20% | ShuffleMTM | $\mathbf{96.38}_{(\pm1.20)}$ | $\mathbf{96.33}_{(\pm1.18)}$ | $\mathbf{96.38}_{(\pm0.89)}$ | $\mathbf{96.34}_{(\pm1.39)}$ | $\underline{90.64}_{(\pm2.25)}$ | $90.43_{(\pm3.23)}$ | $\mathbf{85.94}_{(\pm3.45)}$ | $\underline{87.70}_{(\pm3.12)}$ |
| | TimeSiam | $91.25_{(\pm1.99)}$ | $81.84_{(\pm1.56)}$ | $90.67_{(\pm2.26)}$ | $91.03_{(\pm2.09)}$ | $\mathbf{91.24}_{(\pm0.69)}$ | $\mathbf{94.03}_{(\pm0.71)}$ | $\underline{84.60}_{(\pm1.24)}$ | $\mathbf{87.92}_{(\pm1.06)}$ |
| | PITS | $75.82_{(\pm2.87)}$ | $78.67_{(\pm2.48)}$ | $77.36_{(\pm2.66)}$ | $75.69_{(\pm2.98)}$ | $87.29_{(\pm0.53)}$ | $90.59_{(\pm1.39)}$ | $78.11_{(\pm1.76)}$ | $81.66_{(\pm1.31)}$ |
| | PatchTST | $78.67_{(\pm6.50)}$ | $82.02_{(\pm5.73)}$ | $80.36_{(\pm6.26)}$ | $78.53_{(\pm6.60)}$ | $85.74_{(\pm1.55)}$ | $\underline{91.02}_{(\pm0.61)}$ | $74.71_{(\pm2.97)}$ | $78.44_{(\pm3.10)}$ |
| | SimMTM | $70.55_{(\pm7.26)}$ | $73.91_{(\pm4.99)}$ | $72.29_{(\pm6.52)}$ | $70.16_{(\pm7.75)}$ | $85.09_{(\pm1.20)}$ | $84.43_{(\pm1.97)}$ | $76.82_{(\pm1.49)}$ | $79.36_{(\pm1.65)}$ |
| | COMET | $\underline{92.55}_{(\pm1.88)}$ | $\underline{92.49}_{(\pm1.96)}$ | $\underline{92.73}_{(\pm1.57)}$ | $\underline{92.50}_{(\pm1.86)}$ | $87.46_{(\pm3.25)}$ | $88.89_{(\pm4.98)}$ | $82.30_{(\pm7.90)}$ | $84.46_{(\pm7.30)}$ |
| 10% | ShuffleMTM | $\mathbf{93.75}_{(\pm1.31)}$ | $\mathbf{93.79}_{(\pm1.56)}$ | $\mathbf{93.60}_{(\pm1.28)}$ | $\mathbf{93.67}_{(\pm1.19)}$ | $\mathbf{89.11}_{(\pm1.31)}$ | $\underline{89.55}_{(\pm0.32)}$ | $82.82_{(\pm3.05)}$ | $\mathbf{85.21}_{(\pm2.23)}$ |
| | TimeSiam | $\underline{93.35}_{(\pm1.49)}$ | $\underline{93.52}_{(\pm1.34)}$ | $\underline{93.06}_{(\pm1.65)}$ | $\underline{93.24}_{(\pm1.54)}$ | $\underline{88.50}_{(\pm0.22)}$ | $\mathbf{92.95}_{(\pm0.36)}$ | $79.44_{(\pm0.30)}$ | $83.34_{(\pm0.33)}$ |
| | PITS | $75.37_{(\pm2.43)}$ | $81.49_{(\pm1.76)}$ | $75.37_{(\pm2.43)}$ | $75.10_{(\pm2.57)}$ | $83.28_{(\pm3.78)}$ | $85.52_{(\pm0.62)}$ | $\underline{83.28}_{(\pm0.38)}$ | $73.91_{(\pm0.99)}$ |
| | PatchTST | $68.41_{(\pm2.55)}$ | $74.15_{(\pm0.52)}$ | $68.41_{(\pm2.55)}$ | $67.92_{(\pm3.05)}$ | $86.80_{(\pm1.23)}$ | $89.15_{(\pm1.38)}$ | $\mathbf{83.38}_{(\pm6.59)}$ | $80.56_{(\pm1.97)}$ |
| | SimMTM | $68.23_{(\pm13.47)}$ | $76.44_{(\pm6.68)}$ | $70.90_{(\pm12.05)}$ | $66.22_{(\pm16.08)}$ | $85.58_{(\pm2.15)}$ | $84.49_{(\pm2.54)}$ | $78.05_{(\pm3.50)}$ | $80.32_{(\pm3.39)}$ |
| | COMET | $92.06_{(\pm2.02)}$ | $92.07_{(\pm2.26)}$ | $91.92_{(\pm1.84)}$ | $91.96_{(\pm2.01)}$ | $87.75_{(\pm3.76)}$ | $86.68_{(\pm2.23)}$ | $82.07_{(\pm7.97)}$ | $\underline{83.48}_{(\pm6.55)}$ |
| 5% | ShuffleMTM | $\mathbf{91.17}_{(\pm1.14)}$ | $\underline{91.41}_{(\pm1.51)}$ | $\mathbf{91.79}_{(\pm1.24)}$ | $\mathbf{91.15}_{(\pm0.98)}$ | $\underline{88.15}_{(\pm1.58)}$ | $\mathbf{90.41}_{(\pm2.25)}$ | $\underline{79.94}_{(\pm2.22)}$ | $\underline{83.27}_{(\pm2.36)}$ |
| | TimeSiam | $83.35_{(\pm3.19)}$ | $84.61_{(\pm2.64)}$ | $82.39_{(\pm3.74)}$ | $82.70_{(\pm3.63)}$ | $82.60_{(\pm1.05)}$ | $89.02_{(\pm1.34)}$ | $69.14_{(\pm1.73)}$ | $72.25_{(\pm2.08)}$ |
| | PITS | $71.93_{(\pm5.37)}$ | $78.59_{(\pm3.40)}$ | $74.37_{(\pm4.4)}$ | $71.23_{(\pm5.96)}$ | $85.72_{(\pm0.65)}$ | $88.06_{(\pm1.28)}$ | $75.92_{(\pm0.80)}$ | $79.28_{(\pm0.93)}$ |
| | PatchTST | $85.36_{(\pm6.18)}$ | $83.23_{(\pm1.84)}$ | $82.37_{(\pm1.58)}$ | $80.91_{(\pm1.46)}$ | $80.77_{(\pm1.41)}$ | $88.64_{(\pm1.12)}$ | $65.71_{(\pm2.55)}$ | $67.96_{(\pm3.38)}$ |
| | SimMTM | $68.27_{(\pm9.64)}$ | $75.47_{(\pm3.94)}$ | $70.81_{(\pm8.48)}$ | $66.80_{(\pm11.79)}$ | $84.09_{(\pm0.84)}$ | $84.02_{(\pm0.72)}$ | $74.64_{(\pm1.99)}$ | $77.37_{(\pm1.78)}$ |
| | COMET | $\underline{90.50}_{(\pm2.34)}$ | $\mathbf{91.84}_{(\pm1.84)}$ | $\underline{89.90}_{(\pm2.84)}$ | $\underline{90.20}_{(\pm2.57)}$ | $\mathbf{89.08}_{(\pm0.41)}$ | $\underline{89.57}_{(\pm0.41)}$ | $\mathbf{82.63}_{(\pm0.75)}$ | $\mathbf{85.21}_{(\pm0.64)}$ |

Table 14: Limited labeled data classification.

