# OpenReview forum: "ShuffleMTM: Learning Cross-channel Dependence in Multivariate Time Series from Shuffled Patches"
_ICLR.cc/2025/Conference — Submitted to ICLR 2025_

### Official Review · Reviewer_riEy · 2024-10-23

**Soundness:** 1
**Presentation:** 2
**Contribution:** 1
**Rating:** 3
**Confidence:** 5

**Summary:**

This paper proposes a simple method to improve the multivariate time series self-supervised learning paradigm. In order to capture cross-channel dependencies, it proposes method named ShuffleMTM, which first shuffle unmasked patch along the channel dimension for data processing, and secondly performs information extraction based on cross attention based on the original mask sequence and the shuffled mask sequence.

**Strengths:**

1. The problem is clearly defined and the method is clear and simple；
2. Experimental results show that although this method is simple, it brings certain gains on some data sets.

**Weaknesses:**

1. This paper lacks novelty. Except for the shuffle operation on the unmasked time series , the use of Siamese network does not give me a bright feeling. At the same time, as the author expressed, part of the work differs from patchstst in that the linear layer is replaced by the transformer decoder structure, which seems to be an operation driven by experimental results and has nothing novel.
2. This paper claims to enhance the prediction ability of multivariate time series by capturing cross-channel dependencies. However, in Table 1 we can find that the experimental results do not seem to prove this point. It is also a data set with strong channel correlation, and the weather dataset shows Good results, however, the electricity dataset and traffic dataset performed poorly. Firstly, experimental results prove that this method cannot capture cross-channel dependencies well. Secondly, it cannot prove that the improvements to other data sets are based on cross-channel information. Gains from extraction.
3. The method proposed in this paper has problems in the reasoning stage. In the training and verification stage, some data operations can indeed be performed through shuffle to improve model performance. However, in the reasoning stage, if the random seeds are different each time, then for the same input, due to the uncertainty of other channel patches introduced by the shuffle operation, the output results are inconsistent each time, which is unacceptable in real scenarios.
4. In the robustness exploration experiments based on different proportions of missing values, all selected baseline methods, including this paper, showed that as the proportion of missing values increases, the performance first decreases, and then the performance increases. The author does not give a reasonable explanation for this phenomenon.

**Questions:**

See Weaknesses

---

> ### Author Response · Authors · 2024-11-22
>
> ### W1. This paper lacks novelty. Except for the shuffle operation on the unmasked time series , the use of Siamese network does not give me a bright feeling. At the same time, as the author expressed, part of the work differs from patchstst in that the linear layer is replaced by the transformer decoder structure, which seems to be an operation driven by experimental results and has nothing novel.
>
> ---
>
> -	Thank you for your valuable comment. Your feedback has prompted us to clearly articulate the contribution of ShuffleMTM and to differentiate our work from existing literature.
>
> -	Channel-independent masked time-series modeling (MTM) focuses on learning temporal dependency through encoding each channel separately, but this independent channel encoding inherently overlooks cross-channel dependencies in multivariate time series. As the patterns of individual channels intricately influence one another, neglecting these correlations leads to suboptimal performance in downstream tasks. To address this issue, we propose ShuffleMTM, a novel MTM framework that captures both cross-channel and cross-time dependencies in the channel independent strategy, to achieve advantages of channel-independent and channel-dependent methods.
>
> -	Here, we propose that the shuffling operation on masked time series, Siamese networks, and a cross-attention decoder are essential components for capturing cross-channel dependence in the channel-independent strategy. Unlike previous CI methods that recover masked patches from unmasked patches in the same channel, ShuffleMTM randomly shuffles unmasked patches on masked series, not on unmasked series. After obtaining the original and shuffled masked series, we use Siamese networks to simultaneously encode two views of masked patch representations, leveraging both the temporal relationships within channels and the spatial relationships between patches across different channels. A cross-attention decoder then integrates these original and shuffled views of representations to reconstruct the original time series, effectively capturing the temporal and spatial dependencies in multivariate time series. We demonstrate the effectiveness of these components in capturing cross-channel dependencies in the cross-channel dependence analysis in Section 6.1. The capacity-robustness analysis in Section 6.2 demonstrates superior forecasting capacity and robustness of ShuffleMTM over channel-independent method, validating our framework achieves both advantages of channel-independent and channel-dependent forecasting methods.
>
> -	ShuffleMTM is technically distinct from PatchTST [1] in that PatchTST employs a single-branch encoder and does not utilize the shuffling operation. As aforementioned in the previous paragraph, we emphasize that replacing the linear decoder in PatchTST with a cross-attention decoder is not a modification driven by trial and error but a crucial enhancement for integrating temporal and cross-channel relationships in multivariate time series.
>
> -	We are grateful for your valuable feedback, which has led to meaningful discussions that strengthen the motivation behind and clarify the academic contribution and technical novelty of our work. Based on this response, we will revise Section 1 to clearly convey the motivation and contributions of ShuffleMTM. Thank you again for your comment.
>
> ---
>
> [1] Nie, Y., Nguyen, N. H., Sinthong, P., & Kalagnanam, J. (2023) A Time Series is Worth 64 Words: Long-term Forecasting with Transformers. In The Eleventh International Conference on Learning Representations.

---

> ### Author Response · Authors · 2024-11-22
>
> ### W2. This paper claims to enhance the prediction ability of multivariate time series by capturing cross-channel dependencies. However, in Table 1 we can find that the experimental results do not seem to prove this point. It is also a data set with strong channel correlation, and the weather dataset shows Good results, however, the electricity dataset and traffic dataset performed poorly. Firstly, experimental results prove that this method cannot capture cross-channel dependencies well. Secondly, it cannot prove that the improvements to other data sets are based on cross-channel information. Gains from extraction.
>
> ---
>
> -	We appreciate your careful review and constructive comment, which prompted us to verify the effect of capturing cross-channel dependencies on multivariate time series forecasting performance. Your comment consists of two questions: one regarding the superiority of forecasting performance and another on the effect of cross-channel information on forecasting performance. We will address each question one by one.
>
> -	In Table 1 of the submitted manuscript, ShuffleMTM demonstrates superior performance compared to state-of-the-art channel-independent MTM methods. ShuffleMTM outperforms channel-independent MTMs on multivariate datasets with a high number of channels, such as Weather and Traffic, while showing comparable performance on the Electricity dataset. Additionally, ShuffleMTM achieves the best or second-best performance on multivariate time series with a high number of channels alongside iTransformer, a state-of-the-art forecasting model explicitly designed to capture channel-wise dependencies. These experimental results validate the importance of capturing cross-channel dependence and the superiority of ShuffleMTM for multivariate time series forecasting.
>
> -	To examine the effect of cross-channel information on forecasting performance, we conducted an additional ablation study. We defined a variant of ShuffleMTM in which the shuffle operation is removed, and the cross-attention decoder uses the original masked series representations as query, key, and value. Since this variant cannot utilize the Siamese network, it reduces to PatchTST with a self-attention decoder, which overlooks cross-channel dependencies. We refer to this variant as “ShuffleMTM without shuffle.” Table 1 compares the forecasting performance of ShuffleMTM and ShuffleMTM without shuffle on the ETTh2 and Electricity datasets. The results indicate that removing the shuffle operation decreases forecasting performance, implying that cross-channel information enhances the forecasting capability of ShuffleMTM. This phenomenon is evident in the Electricity dataset, which contains a large number of channels and exhibits strong channel correlations. These findings confirm that capturing cross-channel dependencies through the proposed shuffle operation and Siamese architecture is effective for multivariate time series forecasting.
>
> -	Once again, we sincerely appreciate your comment, which has facilitated meaningful analyses and experiments to strengthen the technical contributions of our work. Based on the response above, we will revise the manuscript to emphasize the superior performance of ShuffleMTM over channel-independent MTM methods and include the ablation study on the effect of cross-channel information in Section 5.
>
> ---
>
> Table 1. Comparision with ShuffleMTM after removing the shuffled view on ETTh2 and Weather. All experiments are repeated three times.
>
> |   |   |   &nbsp;Shuffle | MTM&nbsp;&nbsp;&nbsp;&nbsp;  |  ShuffleMTM  |  w/o shuffle |
> |:---:|:---:|:---:|:---:|:---:|:---:|
> | Dataset      | Length | MSE | MAE | MSE | MAE |
> |ETTh2         |96    |**0.288** |**0.338**|0.289 |**0.338** |
> |              |192   |**0.368**|**0.390**|0.370 |0.391 |
> |              |336   |**0.412** |**0.426**|**0.412** |**0.426** |
> |              |720   |**0.421** |**0.441** |0.422 |0.442 |
> |              |avg   |**0.372** |**0.399** |0.373 |**0.399** |
> |Electricity   |96    |**0.161** |**0.248** |0.162 |0.250 |
> |              |192   |**0.170** |**0.257** |0.172 |0.258 |
> |              |336   |**0.186** |**0.274** |0.188 |0.275 |
> |              |720   |0.228 |**0.310** |**0.227** |**0.310** |
> |              |avg   |**0.186** |**0.272** |0.187 |0.273 |

---

> ### Author Response · Authors · 2024-11-22
>
> ### W3. The method proposed in this paper has problems in the reasoning stage. In the training and verification stage, some data operations can indeed be performed through shuffle to improve model performance. However, in the reasoning stage, if the random seeds are different each time, then for the same input, due to the uncertainty of other channel patches introduced by the shuffle operation, the output results are inconsistent each time, which is unacceptable in real scenarios.
>
> ---
>
> -	Thank you for your careful review and comment. We would like to clarify that ShuffleMTM performs the shuffle operation on masked series during the pre-training stage but does not perform the shuffle operation on unmasked series during the reasoning (fine-tuning) stage.
>
> -	During the pre-training stage, ShuffleMTM randomly shuffles unmasked patches on masked series along the channel and integrates the shuffled view of masked series representations with the original masked series representations to reconstruct the original time series. Through the random shuffling mechanism in pre-training, ShuffleMTM learns both cross-time and cross-channel dependencies in multivariate time series. In the reasoning stage, the weights of the encoder are transferred to project each channel (univariate time series) into a deep representation, which is then fine-tuned for the downstream task. As no shuffling is performed during the reasoning stage, the output results remain consistent regardless of the random seeds.
>
> -	In response to your feedback, we recognize that the mechanism of ShuffleMTM was not clearly presented. We will revise the methodology section to clarify the learning process of ShuffleMTM during both the pre-training and reasoning stages. Once again, thank you for your comment.

---

> ### Author Response · Authors · 2024-11-22
>
> ### W4. In the robustness exploration experiments based on different proportions of missing values, all selected baseline methods, including this paper, showed that as the proportion of missing values increases, the performance first decreases, and then the performance increases. The author does not give a reasonable explanation for this phenomenon.
>
> -	Thank you for your careful review and insightful comment. As you pointed out, the forecasting performance of all selected baselines showed a similar trend, as it first decreases and then increases again as the missing ratio increases. Although ETT contains evident cyclic patterns, the period is not fixed, and the mean shifts over time [2]. For non-stationary time series such as ETT, even small amounts of noise in the training data can abruptly alter the temporal patterns. Transformer-based methods are particularly prone to capturing spurious features present in the training data [2]. Due to overfitting to sudden changes, Transformer-based methods experience significant accuracy degradation [3] However, as the proportion of missing values increases, the amount of correct supervision for the masked reconstruction task decreases, resulting in a self-supervisory task that cannot be easily solved by learning spurious features. This enables the model to focus on learning robust temporal features, which leads to improved forecasting performance [4]. Among the baseline models exhibiting a similar trend, ShuffleMTM consistently demonstrates superior forecasting performance across various missing rates.
>
> -	We sincerely appreciate your detailed review and comments, which have enhanced the clarity of our work. Thanks to your comment, we could conduct in-depth analysis on the experimental results and further contribute to the time series literature. We will include this in-depth discussion on ablation results in the revised manuscript. Thank you again.
>
>
> ---
>
> [2] Shao, Z., Wang, F., Xu, Y., Wei, W., Yu, C., Zhang, Z., ... & Cheng, X. (2024). Exploring progress in multivariate time series forecasting: Comprehensive benchmarking and heterogeneity analysis. IEEE Transactions on Knowledge and Data Engineering.
>
> [3] Zeng, A., Chen, M., Zhang, L., & Xu, Q. (2023). Are transformers effective for time series forecasting?. In Proceedings of the AAAI conference on artificial intelligence.
>
> [4] Chen, Z., Agarwal, D., Aggarwal, K., Safta, W., Balan, M. M., & Brown, K. (2023). Masked image modeling advances 3d medical image analysis. In Proceedings of the IEEE/CVF Winter Conference on Applications of Computer Vision.

---

> ### Comment · Reviewer_riEy · 2024-11-25
>
> Thanks to the author’s reply, which made me clearly understand the modeling method of this paper. However, I still retain my concerns about the innovation and usefulness of this article. Except for the shuffle operation, the other components of this article are a combination of existing work.
>
> First of all, in the main table 1, the method proposed in this paper performs worse than iTransformer (an algorithm used to capture the correlation between channels) in the Electricity and Traffic datasets with strong correlation. Although the author claims that the existing performance can reflect the superiority of the proposed method for multivariate time series prediction tasks. However, if the proposed method cannot surpass existing work (such as iTransformer) in strongly correlated multivariate time series prediction data sets (electricity, transportation, etc.) , it seems that the significance of this article cannot be well proved.
>
> Secondly, according to the ablation experiments supplemented by the author, there is no significant change in performance after removing the shuffle operation. For example, for the electricity dataset, which the authors claim to perform better, ***the performance comparison is only 0.001 for most prediction lengths***, and even at some prediction lengths (target length 720), the performance is even poor, which is further evidenced the proposed shuffle operation in this paper does not bring the gain claimed by the authors said to the strongly correlated multivariate time series forecasting dataset.
>
> Therefore, I will maintain my score.

---

> ### Author Response · Authors · 2024-11-27
>
> Thank you for your active engagement in this review and rebuttal. We appreciate your constructive comments. We understand your concerns regarding (1) the novelty of ShuffleMTM, (2) the performance of ShuffleMTM compared to the supervised learning methods specialized in considering cross-channel dependencies (e.g., iTransformer), and (3) the ablation experiment on the effectiveness of the shuffling method in ShuffleMTM. In response to these concerns, we would like to clarify the following points. We sincerely hope that these clarifications address your concerns and improve the presentation of our contributions.
>
> ---
>
> ### **(1) The novelty of ShuffleMTM**
>
> First of all, please note that ShuffleMTM is a **novel MTM method for self-supervised learning** of multivariate time series. The key contribution of ShuffleMTM lies in its pre-training approach, which is specifically designed to integrate cross-channel dependencies into the channel-independent encoder. The MTM pre-training task of reconstructing masked segments has been demonstrated to be crucial and effective for capturing cross-time dependencies, thereby enhancing time series forecasting and classification tasks [1, 2]. The key contribution of ShuffleMTM lies in its ability to reflect channel dependencies during pre-training, aligning with other MTM studies that focus on enhancing the pre-training stage.
>
> While channel-independent MTM methods focus on separately processing each channel, ShuffleMTM introduces a novel shuffling mechanism. This mechanism incorporates unmasked segments from other channels to capture patterns within this strategy. Also, while channel-independent MTM methods process each channel in a single branch, we extend the single-branch encoders to Siamese encoders to simultaneously learn the cross-time and cross-channel dependencies from the original and shuffled masked series. Then, to effectively incorporate two views of masked series representations, we utilize cross-attention decoder for the reconstruction task. While these components themselves are not originally developed by us, **the entire architectural extension from channel-independent MTM to ShuffleMTM is significant and novel, such as the shuffling method for locating patterns from other channels within the channel-independent strategy, and the architectural extension from a single-branch encoder to Siamese encoders and utilization of cross-attention decoder.**
>
> ---
>
> ### **(2) The performance of ShuffleMTM compared to the supervised learning methods specialized in cross-channel dependencies**
>
> Once again, we would highlight that contribution of our research is to develop a **novel MTM approach** that enables channel-independent models to incorporate cross-channel information during the pre-training stage. **As other MTM studies focus on enhancing the pre-training method, it is important to validate the effectiveness of the pre-training stage for downstream tasks with other self-supervised baselines.** We experimentally demonstrate that ShuffleMTM outperforms state-of-the-art MTM methods which focus on developing an effective pre-training approach. While our model demonstrates comparable performance with a state-of-the-art **supervised** forecasting model that explicitly consider cross-channel dependence on some datasets, it is important that our channel-independent encoder achieves the competitive performance with those channel-dependent **supervised** forecasting models through the proposed pre-training. In addition, It is noteworthy that **ShuffleMTM achieves superior performance in time series classification as well, which confirms the universal effectiveness of its pre-training method across multiple downstream tasks. Thus, we emphasize that ShuffleMTM, as a novel pre-training method, is innovative and contributes to diverse tasks in time series learning.**
>
> ---
>
> [1] Dong et al., (2024). Simmtm: A simple pre-training framework for masked time-series modeling. Advances in Neural Information Processing Systems.
>
> [2] Lee et al., (2024). Learning to Embed Time Series Patches Independently. In The Twelfth International Conference on Learning Representations.

---

> ### Author Response · Authors · 2024-11-27
>
> ### **(3) The ablation experiment on effectiveness of the shuffling method in ShuffleMTM**
>
> The ablation experiment evaluating the shuffled view of ShuffleMTM, which we presented several days ago, does not diminish the utility of ShuffleMTM. **Through extensive experiments, we have demonstrated the effectiveness and technical contributions of our work.** ShuffleMTM has shown superior performance in time series forecasting and classification tasks across both in-domain and cross-domain scenarios. Furthermore, we have demonstrated that ShuffleMTM effectively captures various cross-channel dependencies compared to channel-independent MTM methods. It also enhances both forecasting capacity and robustness—achievements typically associated with channel-dependent and channel-independent forecasting models, respectively.
>
> Regarding the comparison between ShuffleMTM and its variant without the shuffled view, we analyzed the small performance differences on the ETTh2 and Electricity datasets. Shao et al. (2022) [3] found that the ETT and Electricity datasets exhibit low spatial dependencies. We believe the small performance gap between ShuffleMTM and the variant is due to the low spatial dependencies inherent in these datasets. To promptly address reviewers’ comments during the rebuttal period with limited resources available, we initially selected relatively small datasets that were feasible for immediate analysis. We now understand that selecting ETT and Electricity datasets was not ideal for the purpose of that ablation study. Recognizing the importance of evaluating our method on datasets with varying spatial dependencies, we have now extended our experiments as detailed below, taking the opportunity provided by the rebuttal extension for ICLR 2025.
>
> Thanks to the extension of the rebuttal for ICLR 2025, we have proactively conducted additional experiments to compare ShuffleMTM with its variant without the shuffled view, so that we can address your concern. We generated two synthetic datasets: the first consists of three channels, each exhibiting lagged structures relative to one another. The second dataset also consists of three channels, all of which share the same long-term trend but exhibit distinct local patterns. Thus, the first dataset has high local dependencies between channels, whereas the second has weak local dependencies. Table 2 below presents the forecasting performance of ShuffleMTM and its variant without the shuffled view on these two synthetic datasets. Both models were evaluated in forecasting scenarios with prediction lengths of {32, 64, 96, 192} and a fixed input length of 96, with the patch length set to 16. ShuffleMTM outperformed its variant across both datasets, with a larger performance gap observed in the first dataset than in the second. These experiments suggest that **the shuffling method enables the channel-independent encoder to capture fine-grained dependencies between channels.** However, **the shuffling method is less effective on datasets with weak local dependencies compared to those with strong local dependencies**. Based on the experimental results from the synthetic datasets and this analysis, we explain that **the small performance difference between the two models on ETTh2 and Electricity is due to the low spatial dependencies in these datasets.**
>
> Furthermore, to confirm the abovementioned finding and to further validate the utility of ShuffleMTM in capturing cross-channel dependencies, we will compare ShuffleMTM with its variant without the shuffled view using other benchmark datasets with high spatial dependencies as well. For example, we will use the PEMS dataset, which collects traffic flow data in urban road networks and is known for its strong spatial dependencies. We will conduct this comparison and upload the results during the rebuttal period. We kindly ask you for a few days to complete this experiment.
>
> ---
>
> Table 2. Comparison between ShuffleMTM and its variant without the shuffled view on two synthetic datasets. All experiments were repeated three times.
>
> |Dataset||&nbsp;&nbsp;Shuffle|MTM&nbsp;| &nbsp;&nbsp;ShuffleMTM  |  w/o shuffle &nbsp; &nbsp;|
> |:---:|:---:|:---:|:---:|:---:|:---:|
> ||Pred_len|MSE|MAE|MSE|MAE|
> |Synthetic|32|**1.023**|**0.896**|1.034|0.899|
> |data 1|48|**1.024**|**0.897**|1.034|0.899|
> ||96|**1.026**|**0.898**|1.033|0.898|
> ||192|**1.026**|0.899|1.029|**0.897**|
> |Synthetic|32|**0.292**|**0.477**|0.293|0.479|
> |data 2|48|**0.292**|**0.477**|0.293|0.479|
> ||96|**0.294**|**0.479**|0.295|**0.479**|
> ||192|**0.296**|**0.480**|**0.296**|**0.480**|
>
>
> [3] Shao et al., (2024). Exploring progress in multivariate time series forecasting: Comprehensive benchmarking and heterogeneity analysis. IEEE Transactions on Knowledge and Data Engineering.

---

> ### Author Response · Authors · 2024-11-27
>
> Regarding our responses (1), (2), and (3) mentioned above, we would like to emphasize that the **key contribution of our work lies in its pre-training approach, which is specifically designed to integrate cross-channel dependencies into the channel-independent encoder.** As all channel-independent encoders inevitably processes each channel independently, these methods are only able to implicitly learn inter-channel relationships by separating and processing all channels in a multivariate time series within the same batch. However, these methods cannot learn patch-level dependencies between different channels, as they process each channel independently. This prevents the model from learning relationships between channels, such as cross-correlation at lagged positions, which are commonly observed in multivariate time series [4].
>
> To verify this and to further confirm the contribution of ShuffleMTM in capturing patch-level dependencies between different channels, we compare the performance of ShuffleMTM with PatchTST on the aforementioned first synthetic dataset, which exhibits clear patch-level dependencies between channels. As shown in Table 3 below, ShuffleMTM consistently outperforms PatchTST across all prediction lengths, demonstrating its ability to capture patch-level dependencies in channels with lagged structures. This result confirms that **the pre-training of ShuffleMTM is effective for forecasting multivariate time series (MTS) with patch-level dependencies across channels.**
>
> Furthermore, please note that from our cross-channel dependence analysis in Section 6.1, we conducted a similarity analysis on the shuffled patches series. Specifically, we compared cosine similarities between the self-attention map and the correlation coefficients of patches derived from the randomly shuffled patch series across channels. In this experiment, **ShuffleMTM achieves a higher cosine similarity compared to other channel-independent MTM methods. This result further supports the conclusion that the pre-training of ShuffleMTM enables the channel-independent encoder to better capture patch-level dependencies.**
>
> Once again, we greatly appreciate your insightful feedback, which has enabled us to strengthen our work further. With these additional experimental results and responses, we sincerely hope that we have clarified our contributions in MTM research as an indispensable approach for advancing temporal as well as spatial modeling capabilities of MTM. We sincerely hope we have clarified our contributions and improved the presentation of our work by addressing your comments. We hope that these improvements have increased your confidence in our work and provided you with a more favorable view of its novelty and technical validity.
>
> ---
>
> Table 3. Evaluation of ShuffleMTM and PatchTST on the first synthetic dataset. All experiments are repeated three times.
>
> ||&nbsp;&nbsp;Shuffle|MTM&nbsp;|&nbsp;Patch|TST&nbsp;|
> |:---:|:---:|:---:|:---:|:---:|
> |Pred_len|MSE|MAE|MSE|MAE|
> |32|**1.023**|0.896|1.030|**0.895**|
> |48|**1.024**|**0.897**|1.037|0.901|
> |96|**1.026**|**0.898**|1.033|0.901|
> |192|**1.026**|**0.899**|1.031|0.901|
>
> [4] Zhao, L., & Shen, Y. (2024) Rethinking Channel Dependence for Multivariate Time Series Forecasting: Learning from Leading Indicators. In The Twelfth International Conference on Learning Representations.

---

> ### Comment · Reviewer_riEy · 2024-11-30
>
> I appreciate the author's efforts in the rebuttal stage, but unfortunately the author did not address my concerns.
>
> First of all, as I mentioned before, the motivation of this paper is to improve the performance of downstream tasks by increasing the interaction between channels. Unfortunately, through the experiments in Main Table 1, we can see that the method proposed in this paper does not compare existing cross-channel information extraction methods (such as iTransformer, etc.) work better, the authors acknowledge this, and believe that their method is more effective than existing self-supervised learning methods and achieves the design goals. But I don't agree with this. If authors are comparing the pre-training plus fine-tuning paradigm, can I ask the author to compare it with the existing time series foundation models? (Such as Moirai[1] et al.), these works also belong to the self-supervised learning paradigm, but use more datasets in the pre-training stage. In real-life scenarios, what we need is the algorithm with the best performance, rather than selecting an algorithm in a custom subset(Within the range allowed by computing power).
>
> Secondly, in the reconstruction target part of the ablation experiment, corresponding to the visualization in Figure 3, we can find that the difference in comparative experimental results between different CA queries and different reconstruction targets is basically negligible, which further proves that the CA strategy used in this paper does not brings the useful benefits claimed by the author.
>
> Combining the above points and the feedback from other reviewers, I will maintain my rating and believe that the current work is not suitable for publication at this conference.
>
> [1] Woo, G., Liu, C., Kumar, A., Xiong, C., Savarese, S., & Sahoo, D. (2024). Unified training of universal time series forecasting transformers. *arXiv preprint arXiv:2402.02592*.

---

> ### Author Response · Authors · 2024-12-02
>
> ### **(1) Comparison with time series foundation models**
>
> ---
>
> Thank you for your continued active engagement in this review and rebuttal.
>
> In response to your comment regarding a comparison between ShuffleMTM and time series foundation models, we evaluated ShuffleMTM against Moirai [1] and Timer [2] in long-term forecasting scenarios. For ShuffleMTM and Timer, we followed the standard protocol in which the model is trained and fine-tuned on the same dataset to predict future windows of lengths {96, 192, 336, 720}. Meanwhile, since Moirai is a zero-shot forecasting model, we compared our method against the zero-shot performance of Moirai using the results reported in its original paper. Table 4 shows the forecasting performance of ShuffleMTM and Timer on the ETT, Exchange, Weather, Electricity, and Traffic datasets, averaged across all prediction lengths. ShuffleMTM outperforms Timer in long-term forecasting across all datasets. Timer, which predicts future windows autoregressively using generative modeling, tends to accumulate errors in long-term forecasting. Table 5 presents the forecasting performance of ShuffleMTM and Moirai on the ETT, Weather, and Electricity datasets, also averaged across all prediction lengths. ShuffleMTM demonstrates performance comparable to Moirai. These experimental results underscore **the effectiveness of ShuffleMTM over time series foundation models in learning representations specific to the dataset for time series forecasting tasks.**
>
> While Moirai and Timer are pre-trained on a large-scale dataset and learn various time series patterns from vast amounts of data, ShuffleMTM is more effective in learning representations by capturing channel dependencies specific to each dataset. We emphasize that ShuffleMTM pursues different research objectives compared to time series foundation models, such as Moirai and Timer. While **time series foundation models aim to establish a pretrained model** that can be efficiently applied to a wide range of time series tasks by leveraging large-scale datasets, **ShuffleMTM, as a pretraining method,** is the first MTM method to learn representations that captures both cross-time and cross-channel dependencies within a channel-independent framework for a specific dataset. Compared to iTransformer [3], ShuffleMTM outperforms it on six out of eight datasets and achieves second-best performance on two datasets. ShuffleMTM achieves state-of-the-art performance compared to all other existing channel-independent self-supervised methods. These results underscore that **ShuffleMTM provides an effective pre-training strategy for capturing both cross-time and cross-channel dependencies.**
>
> In response to your comment, the discussion on the channel-independent strategy and time series foundation models has inspired us to propose a future research direction: the development of time series foundation models capable of capturing both cross-time and cross-channel dependencies. While Moirai and Timer offer efficient architectures for training on large-scale datasets, they face challenges in learning dataset-specific cross-channel dependencies. By combining the strengths of ShuffleMTM and existing foundation models, we believe that future foundation models should be designed to adaptively capture dynamic cross-channel dependencies across various numbers of channels in time series data.
>
> We sincerely appreciate the discussion with you, which has greatly clarified and enhanced the academic contributions of our work while improving its presentation. We have included this discussion and the proposed future research direction in Section 7 of the revised manuscript and Section E of the Appendix. Following this rebuttal, we plan to further conduct and include an in-depth analysis on cross-channel dependence and comparison with time series foundation models, should our work be accepted at ICLR 2025. Once again, we thank you for your invaluable insights and inspiration. We sincerely hope you could kindly acknowledge our unique contributions of ShuffleMTM to the MTM methodology literature and efforts throughout this rebuttal.

---

> > ### Author Response · Authors · 2024-12-02
> >
> > Table 4. Comparison with Timer. All experiments are implemented three times. We report the average forecasting performance for all prediction lengths \{96, 192, 336, 720\}.
> >
> > |   |&nbsp;&nbsp;Shuffle|MTM&nbsp;|Timer   |   |
> > |:-----------:|:-----:|:-----:|:-----:|:-----:|
> > |Dataset    |MSE  |MAE  |MSE  |MAE  |
> > |ETTh1      |**0.432**|**0.435**|0.439|0.436|
> > |ETTh2      |**0.372**|**0.399**|0.395|0.411|
> > |ETTm1      |**0.379**|**0.394**|0.430|0.420|
> > |ETTm2      |**0.278**|**0.325**|0.296|0.333|
> > |Exchange   |**0.353**|**0.398**|0.357|0.401|
> > |Weather    |**0.250**|**0.275**|0.279|0.294|
> > |Electricity|**0.186**|**0.272**|0.230|0.303|
> > |Traffic    |**0.449**|**0.281**|0.492|0.312|
> >
> > Table 5. Comparison with MOIRAI. We report the average forecasting performance for all prediction lengths \{96, 192, 336, 720\}.
> >
> > |   |&nbsp;&nbsp;Shuffle|MTM&nbsp;|MOIRAI   |   |
> > |:-----------:|:-----:|:-----:|:-----:|:-----:|
> > |Dataset    |MSE  |MAE  |MSE  |MAE  |
> > |ETTh1      |0.432|0.435|**0.400**|**0.424**|
> > |ETTh2      |0.372|0.399|**0.341**|**0.379**|
> > |ETTm1      |**0.379**|**0.394**|0.448|0.410|
> > |ETTm2      |**0.278**|**0.325**|0.300|0.341|
> > |Weather    |0.250|0.275|**0.242**|**0.267**|
> > |Electricity|**0.186**|**0.272**|0.233|0.320|
> >
> > [1] Woo, G., Liu, C., Kumar, A., Xiong, C., Savarese, S., & Sahoo, D. (2024) Unified Training of Universal Time Series Forecasting Transformers. In Forty-first International Conference on Machine Learning.
> >
> > [2] Liu, Y., Zhang, H., Li, C., Huang, X., Wang, J., & Long, M. (2024) Timer: Generative Pre-trained Transformers Are Large Time Series Models. In Forty-first International Conference on Machine Learning.

---

> ### Author Response · Authors · 2024-12-02
>
> ### **(2) The validity of cross-attention decoder in ShuffleMTM**
>
> ---
>
> Finally, please allow us to address your question regarding Figure 3: The results of the ablation study on the reconstruction target demonstrate that integrating original and shuffled views of masked series representations is crucial for forecasting performance. This is due to the integration of spatial and temporal information in multivariate time series, irrespective of the reconstruction target or query choice. The robust performance, regardless of the reconstruction target and query choice, indicates that adopting a cross-attention decoder effectively integrates cross-time and cross-channel dependencies.
>
> Furthermore, in our previous response, we also demonstrated in Table 2 the effectiveness of adopting the shuffled view and cross-attention decoder on a synthetic dataset with high channel dependencies. As we had promised, we have further evaluated ShuffleMTM against its variant without the shuffled view on the PEMS08 dataset as well, which exhibits high periodicity and spatial dependencies. Table 6 presents the forecasting performance of both models for prediction lengths {96, 192, 336, 720}. ShuffleMTM consistently outperforms its variant across all prediction lengths. These results, combined with the findings from Table 2, confirm that the **shuffling method and cross-attention decoder are effective for forecasting in datasets with high channel dependencies.**
>
> In summary, through the ablation study on reconstruction targets and the additional experiment on the shuffled view and cross-attention decoder, we conclude that ShuffleMTM effectively captures the dependent structure across channels.
>
>
>
> ---
>
> Table 6. Comparison between ShuffleMTM and its variant without the shuffled view on the PEMS08 datasets. All experiments were repeated three times.
>
> |Dataset||&nbsp;&nbsp;Shuffle|MTM&nbsp;| &nbsp;&nbsp;ShuffleMTM  |  w/o shuffle &nbsp; &nbsp;|
> |:---:|:---:|:---:|:---:|:---:|:---:|
> ||Pred_len|MSE|MAE|MSE|MAE|
> |PEMS08|96|**0.380**|**0.418**|0.383|0.421|
> ||192|**0.438**|**0.446**|0.442|0.449|
> ||336|**0.394**|**0.407**|0.397|0.409|
> ||720|**0.465**|**0.459**|0.468|0.461|

---

> ### Comment · Reviewer_riEy · 2024-12-03
>
> I appreciate that the author conducted sufficient experiments during the rebuttal stage, but unfortunately, the experimental results do not address my concerns.
>
> First of all, as I mentioned before, the performance of the Traffic and Electricity datasets provided in main table 1 raises my concerns about the motivation of this paper, which is to "enhance the prediction performance by capturing the correlation between different channels". The views have not been verified on the corresponding datasets. It is worth mentioning that I have also carefully reviewed the feedback from reviewer wko4. I agree that the proposed method does not need to show the best results on all types of datasets. However, unfortunately, if the proposed method cannot Excellent performance is shown on strongly correlated datasets (the method proposed in this paper considers the information interaction between channels, so datasets such as Traffic are strongly related to them), then I think the motivation of the paper cannot be verified.
>
> Secondly, regarding the visualization results in Figure 3 corresponding to the CA operation, the author believes that this reflects the robustness of its design. I don’t think so. This kind of result with no significant difference reflects that the CA operation proposed in this paper is meaningless. Because different CA queries and CA keys will determine the calculation of the attention map, and the final feature combination is for the CA value.
>
> Finally, it needs to be emphasized that the shuffle operation, the core contribution of this paper, does not bring the excellent performance emphasized by the author. In the rebuttal stage, the author conducted ablation experiments on multiple datasets, among which ETTh2, Electricity, and PEMS08 are real datasets, and Synthetic data2 is a synthetic dataset. Unfortunately, these experiments show that the presence or absence of the shuffle operation will not have a significant impact on the final prediction results (the ablation experimental results are basically reflected in the difference to three decimal places). This also makes me very confused. Obviously the ablation experiment results show that the core contribution point is meaningless. Why does reviewer wko4 recognize this paper so much?
>
> Combining the above points and the feedback from other reviewers, I will maintain my rating and believe that the current work is not suitable for publication at this conference. I encourage the author to spend more time thinking about the core contribution of this paper, and to prove its design motivation as much as possible.

---

> > ### Author Response · Authors · 2024-12-04
> >
> > We sincerely thank Reviewer riEy for the detailed feedback. To address the reviewer’s concerns, we would like to provide a clear explanation of the contributions and experimental results of the proposed ShuffleMTM methodology. Below, we respond to the three main points inquired by the reviewer and summarize the overall contributions of this study.
> >
> > First of all, as we clarified in Meta Response 2, the essential contribution of our research is the development of **a novel MTM method that enables channel-independent models to incorporate cross-channel information during the pre-training stage**. Thus, to validate the effectiveness of the pre-training stage, it is essential to compare with channel-independent MTM methods. We experimentally demonstrate that ShuffleMTM outperforms state-of-the-art channel-independent MTM methods in all benchmark datasets, including high-channel datasets. ShuffleMTM achieves superior performance compared to the state-of-the-art forecaster that considers channel dependencies, outperforming it on six out of eight datasets. Thus, we emphasize that ShuffleMTM achieves notable performance in forecasting tasks.
> >
> > Secondly, the core mechanism of the cross-attention decoder involves attending a query to different keys and values and integrating distinct information from the query, key, and value. This core mechanism remains unchanged regardless of whether the original view or the shuffled view is used as the query. Thus, **the stable performance across all query choices and reconstruction targets shows that the integration of spatial and temporal information encoded in both views is crucial for forecasting performance**. The core mechanism of the cross-attention decoder in ShuffleMTM is critical for capturing both cross-time and cross-channel dependencies.
> >
> > Lastly, the contribution of ShuffleMTM lies not only in its superior performance but also in its capability to capture cross-channel dependencies within a channel-independent strategy. We demonstrate that ShuffleMTM effectively captures various granularities of cross-channel dependencies through the analyses in Section 6.1, and it outperforms the MTM variant without the shuffled view on the highly correlated synthetic dataset. While channel-independent methods are well known to be effective for time series forecasting, the shuffle operation consistently improves performance across real-world datasets. **These experimental results support the conclusion that the shuffle operation effectively captures cross-channel dependencies and that the observed improvements are attributed to the design of ShuffleMTM.**
> >
> > Once again, we sincerely thank you for your active engagement and constructive comments during the rebuttal stage. Your comments have greatly helped us refine the academic and technical contributions of our work.

---

### Official Review · Reviewer_5ATj · 2024-11-01

**Soundness:** 3
**Presentation:** 2
**Contribution:** 3
**Rating:** 6
**Confidence:** 4

**Summary:**

This paper proposes a simple time series masking modeling method based on a Siamese network and a specially designed decoder. It establishes the relationship between masked patches within a channel and shuffled patches across channels at the patch level, enabling learning of both intra-channel and inter-channel temporal dependencies. This approach demonstrated superior performance on the main temporal analysis task.​

**Strengths:**

- The proposed method effectively models temporal correlations and dependencies within and across channels through a simple yet effective masking approach.
- Comprehensive experiments and analyses support the method.

**Weaknesses:**

- Limited generalizability: The method relies on a channel-independent time series model as the backbone, limiting its applicability to more advanced temporal models, especially those that inherently model inter-channel relationships.

- The motivation of the paper could be more precisely stated. The CI method does not completely ignore inter-channel modeling; Rather, they construct these relations implicitly rather than explicitly, as multiple channels are simultaneously optimized in a single iteration.​

**Questions:**

- **Main Experiments**:

Effectiveness should be validated on larger datasets. Numerous masking and generative pre-training methods, such as Moirai [1] and Timer [2], have achieved notable results on large-scale time series data. Comparisons of these approaches could shed light on the effectiveness of this mask modeling approach as data size scales, and whether it holds advantages over other unsupervised pre-training paradigms in larger data scenarios.​

- **Ablation Study**:

The performance of ShuffleMTM is most likely affected by the masking ratio and the number of channels. The masking ratio controls the potential shuffled candidate locations, while the number of channels determines the diversity of patch replacements. A more detailed examination of their relationship is recommended. For example, on high-channel datasets (such as Traffic and ECL), it would be beneficial to explore how model performance changes with increasing masking ratios.

​Another ablation experiment could evaluate the effect of enforcing that the current position in one channel is always replaced by another channel during shuffling.​

[1] Unified Training of Universal Time Series Forecasting Transformers

[2] Timer: Generative Pre-trained Transformers Are Large Time Series Models

---

> ### Author Response · Authors · 2024-11-22
>
> ### W1. Limited generalizability: The method relies on a channel-independent time series model as the backbone, limiting its applicability to more advanced temporal models, especially those that inherently model inter-channel relationships.
>
> ---
>
> -	We thank you for your constructive comment. We present ShuffleMTM as a solution to address the disregard of cross-channel dependence in multivariate time series for channel-independent masked time series modeling (MTM) methods. Here, we would like to emphasize that ShuffleMTM is not a modular architecture designed to be applied to channel-independent MTM. Instead, we would like to highlight that ShuffleMTM is a channel-independent MTM framework that captures both cross-channel and cross-time dependencies, similar to other advanced temporal models that inherently model these relationships [1, 2]. Whereas these approaches propose enhanced attention mechanisms [1, 2], ShuffleMTM proposes a novel self-supervised pre-training task to capture this dependence by reconstructing each channel through the integration of original and shuffled views of masked series. To integrate both views of masked series, our shuffling method generates the original and shuffled masked series, Siamese encoders simultaneously encode both views, and a cross-attention decoder integrates cross-time and cross-channel dependencies encoded in the representations to reconstruct each channel. The articulated design of ShuffleMTM effectively captures cross-time and cross-channel dependencies within the channel-independent strategy, demonstrating superior performance in time series forecasting and classification tasks. Since our work captures cross-channel dependence through a pre-training task rather than a meticulously designed attention mechanism, it represents a novel and important contribution to the time series literature.
>
> -	Specifically, ShuffleMTM provides technical advantages over existing models that capture inter-channel relationships. These methods can be categorized into two approaches: those using sequential cross-time and cross-variate attention on multivariate time series, and those applying 1D attention on flattened time series [1, 2]. However, sequentially applying self-attention in horizontal and vertical manners is inefficient for learning dependencies between patches from other channels at lagged locations. In contrast, shuffling unmasked patches across channels dynamically imposes patches at lagged locations and enables the channel-independent encoder to capture dynamic patch-level dependencies across channels in different positions. Applying 1D self-attention to flattened patches from all channels allows the encoder and decoder to access unmasked patch embeddings from different channels with identical temporal information. This simplifies the reconstruction task, negatively impacting the training of the encoder. By contrast, shuffling unmasked patches from other channels in different locations integrates cross-channel information into the channel-independent reconstruction task without relying on patches with identical temporal information.
>
> -	In summary, ShuffleMTM is not a modular component that can be directly applied to channel-independent models. Instead, it is a novel framework that captures both cross-channel and cross-time dependencies while retaining the channel-independent strategy. While existing models consider cross-channel dependence through enhanced attention mechanisms, ShuffleMTM proposes a novel self-supervised pre-training task to capture this dependence, offering technical advantages over these inter-channel forecasting models
>
> -	We sincerely appreciate your review and constructive comment, which has helped us strengthen the positioning of our work in the time series literature. Thanks to your feedback, we believe we can further enhance the technical contributions and novelty of our work.
>
> ---
>
> [1] Zhang, Y., & Yan, J (2023). Crossformer: Transformer utilizing cross-dimension dependency for multivariate time series forecasting. In The eleventh international conference on learning representations.
>
> [2] Liu, J., Liu, C., Woo, G., Wang, Y., Hooi, B., Xiong, C., & Sahoo, D. (2024). UniTST: Effectively Modeling Inter-Series and Intra-Series Dependencies for Multivariate Time Series Forecasting. arXiv preprint arXiv:2406.04975.

---

> ### Author Response · Authors · 2024-11-22
>
> ### W2. The motivation of the paper could be more precisely stated. The CI method does not completely ignore inter-channel modeling; Rather, they construct these relations implicitly rather than explicitly, as multiple channels are simultaneously optimized in a single iteration.
>
> ---
>
> -	Thank you for your careful review and constructive comment. While CI methods explicitly focus on the temporal patterns within each channel, these methods implicitly learn inter-channel relationship by separating and processing all channels in a multivariate time series within the same batch. However, these methods cannot learn patch-level dependencies between different channels, as they process each channel independently. This prevents the model from capturing various relationships between channels, such as cross-correlation at lagged positions, which are commonly observed in multivariate time series [3]. To address this limitation, we propose ShuffleMTM to capture these fine-grained patch-level dependencies across channels by reconstructing the original series by incorporating the original and shuffled view of masked series. In the cross-channel dependence analysis in Section 6.1, we demonstrate that the ShuffleMTM effectively captures patch-level cross-channel dependence whereas CI methods fail to learn these dependencies.
>
> - We appreciate your insightful comment and thorough feedback. Based on your feedback, we will clarify the limitations of the existing works and sharpen the contributions of our work in the revised manuscript.
>
> ---
>
> [3] Zhao, L., & Shen, Y (2024). Rethinking Channel Dependence for Multivariate Time Series Forecasting: Learning from Leading Indicators. In The Twelfth International Conference on Learning Representations.

---

> ### Author Response · Authors · 2024-11-22
>
> ### Q1. Effectiveness should be validated on larger datasets. Numerous masking and generative pre-training methods, such as Moirai [1] and Timer [2], have achieved notable results on large-scale time series data. Comparisons of these approaches could shed light on the effectiveness of this mask modeling approach as data size scales, and whether it holds advantages over other unsupervised pre-training paradigms in larger data scenarios.
>
> ---
>
> -	We thank you for your critical and insightful comments. We emphasize that ShuffleMTM pursues different research objectives compared to Moirai and Timer. While Moirai and Timer propose a unified framework to address all time series tasks efficiently by pre-training models on large-scale datasets, ShuffleMTM is the first MTM framework to learn representations that captures both cross-time and cross-channel dependencies on the specific dataset in the channel-independent strategy. Since the channel-independent strategy flattens the channel dimensions into the same batch, channel-independent methods inherently require that the batch of samples originates from the same dataset and involve significant memory usage. Consequently, channel-independent methods are unsuitable for learning on large-scale datasets containing time series with varying channel dimensions.
>
> -	Although ShuffleMTM is not suitable for learning from large-scale datasets, we concur with the reviewer’s opinion that it is worth comparing ShuffleMTM with Moirai and Timer. As such, we evaluate the performance of ShuffleMTM against Moirai and Timer in long-term forecasting scenarios to demonstrate its effectiveness in learning time series representations for a specific target dataset. For ShuffleMTM and Timer, we follow the standard protocol in which the model is trained and fine-tuned on the same dataset to predict a future window of lengths {96, 192, 336, 720}. However, since Moirai is a zero-shot forecasting model, we compare our method against the zero-shot performance of Moirai using results already reported in its paper.

---

> > ### Author Response · Authors · 2024-11-22
> >
> > ### Q1. (Cont'd) Effectiveness should be validated on larger datasets. Numerous masking and generative pre-training methods, such as Moirai [1] and Timer [2], have achieved notable results on large-scale time series data. Comparisons of these approaches could shed light on the effectiveness of this mask modeling approach as data size scales, and whether it holds advantages over other unsupervised pre-training paradigms in larger data scenarios.
> >
> > ---
> >
> > -	Table 1 displays the forecasting performance of ShuffleMTM and Timer on four ETT, Exchange, Weather, Electricity, and Traffic datasets, averaged across all prediction lengths. ShuffleMTM outperforms Timer in long-term forecasting for all datasets. Since Timer predicts the future window autoregressively through generative modeling, it accumulates errors in long-term forecasting. Table 2 presents the forecasting performance of ShuffleMTM and Moirai on four ETT, Weather, and Electricity datasets, averaged across all prediction lengths. ShuffleMTM demonstrates comparable performance to Moirai. These experimental results underscore the effectiveness of ShuffleMTM over time series foundation models in learning representations specific to the dataset for time series forecasting tasks.
> >
> > -	In response to your comment, the discussion on the channel-independent strategy and time series foundation models has inspired us to propose a future research direction; time series foundation models that capture cross-time and cross-channel dependencies are necessary for time series forecasting task. While Moirai and Timer has an efficient architecture to train large-scale datasets, it shows difficulty in learning cross-channel dependence specific to the dataset. Taking both advantages of ShuffleMTM and foundation models, foundation models should be developed to adaptively capture various cross-channel dependencies on various number of channels of time series. We will include this discussion and future research direction in the revised manuscript.
> >
> > -	We sincerely appreciate your careful review and constructive suggestions, which have encouraged us to compare our work with foundation models and demonstrate competitive performance. We believe that these evaluations validate the technical contributions of our work. Once again, thank you.
> >
> > ---
> >
> > Table 1. Comparison with Timer. All experiments are implemented three times. We report the average forecasting performance for all prediction lengths \{96, 192, 336, 720\}.
> >
> > |   |&nbsp;&nbsp;Shuffle|MTM&nbsp;|Timer   |   |
> > |:-----------:|:-----:|:-----:|:-----:|:-----:|
> > |Dataset    |MSE  |MAE  |MSE  |MAE  |
> > |ETTh1      |**0.432**|**0.435**|0.439|0.436|
> > |ETTh2      |**0.372**|**0.399**|0.395|0.411|
> > |ETTm1      |**0.379**|**0.394**|0.430|0.420|
> > |ETTm2      |**0.278**|**0.325**|0.296|0.333|
> > |Exchange   |**0.353**|**0.398**|0.357|0.401|
> > |Weather    |**0.250**|**0.275**|0.279|0.294|
> > |Electricity|**0.186**|**0.272**|0.230|0.303|
> > |Traffic    |**0.449**|**0.281**|0.492|0.312|
> >
> >
> >
> > Table 2. Comparison with MOIRAI. We report the average forecasting performance for all prediction lengths \{96, 192, 336, 720\}.
> >
> > |   |&nbsp;&nbsp;Shuffle|MTM&nbsp;|MOIRAI   |   |
> > |:-----------:|:-----:|:-----:|:-----:|:-----:|
> > |Dataset    |MSE  |MAE  |MSE  |MAE  |
> > |ETTh1      |0.432|0.435|**0.400**|**0.424**|
> > |ETTh2      |0.372|0.399|**0.341**|**0.379**|
> > |ETTm1      |**0.379**|**0.394**|0.448|0.410|
> > |ETTm2      |**0.278**|**0.325**|0.300|0.341|
> > |Weather    |0.250|0.275|**0.242**|**0.267**|
> > |Electricity|**0.186**|**0.272**|0.233|0.320|

---

> ### Author Response · Authors · 2024-11-22
>
> ### Q2. The performance of ShuffleMTM is most likely affected by the masking ratio and the number of channels. The masking ratio controls the potential shuffled candidate locations, while the number of channels determines the diversity of patch replacements. A more detailed examination of their relationship is recommended. For example, on high-channel datasets (such as Traffic and ECL), it would be beneficial to explore how model performance changes with increasing masking ratios.
>
> ---
>
> -	Responding to your feedback, we conducted an ablation study on various masking ratios using the ECL (Electricity) dataset including the ETTh2 and Exchange datasets in Figure 5 of the submitted manuscript. As shown in Table 3, the MSE error decreases until the masking ratio reaches 0.4, after which it increases as the masking ratio continues to rise. While masking ratios do not have much influence on forecasting performance for time series with a small number of channels such as ETTh2 and Exchange, the effect of masking ratios is more pronounced on high-channel datasets, such as ECL. When the masking ratio is low, the proportion of self-supervision is insufficient, resulting in poor forecasting performance. However, as the masking ratio increases, the number of potential shuffled candidate locations decreases. Consequently, the diversity of patch replacements diminishes, degrading forecasting performance. Therefore, an appropriate masking ratio may be critical for forecasting performance on high-channel datasets, whereas performance remains robust to masking ratios on low-channel datasets. Based on this empirical analysis, we recommend setting the masking ratio for ShuffleMTM at 0.4.
>
> -	We sincerely appreciate your valuable comment, which has helped us to conduct in-depth analysis on the ablation results. We will improve the ablation study on the effect of masking ratios in Section 5 of the revised manuscript to strengthen our technical contribution. Thank you again.
>
> ---
>
> Table 3.  Ablation experiment on the masking ratio. All experiments are repeated three times. We report the average MSE performance for all prediction lengths \{96, 192, 336, 720\}. We experimented on masking ratios {0.2, 0.4, 0.6, 0.8} for the ECL dataset. For the ETTh2 and Exchange datasets, we experimented on masking ratios {0.3, 0.4, 0.5, 0.6 0.7, 0.8} and collected the results of masking ratios {0.3, 0.4, 0.6, 0.8} to ensure comparability of results.
>
> |Masking ratio|0.2/0.3|&nbsp;&nbsp;&nbsp;0.4  |&nbsp;&nbsp;&nbsp;0.6  |&nbsp;&nbsp;&nbsp;0.8  |
> |:-----------:|:-------:|:-----:|:-----:|:-----:|
> |ETTh2      |0.374  |**0.372**|0.373|0.374|
> |Exchange   |0.353  |0.353|0.353|**0.352**|
> |ECL|0.188  |**0.186**|0.188|0.190|

---

> ### Author Response · Authors · 2024-11-22
>
> ### Q3. Another ablation experiment could evaluate the effect of enforcing that the current position in one channel is always replaced by another channel during shuffling.
>
> ---
>
> -	Thank you very much for your insightful suggestion. Following your comment, we examined the effect of different shuffling mechanisms for ShuffleMTM. Specifically, we compared our random shuffling method with strict shuffling, which enforces that the current position in a channel is always replaced by another channel. As shown in Table 4, ShuffleMTM with random shuffling achieves better forecasting performance than ShuffleMTM with strict shuffling. Strict shuffling decreases the diversity of patch replacements and reduces the temporal information within the original channel, making it harder to reconstruct the masked patches of that channel during pre-training.
>
> -	Once again, we sincerely appreciate your valuable comment, which has significantly contributed to the improvement of our work. The results of the ablation experiment on the shuffling method will be updated in Section 5 of the revised manuscript. Thanks to your feedback, we are confident that this additional ablation study will validate the technical effectiveness of ShuffleMTM.
>
> ---
>
> Table 4. Ablation experiments on the shuffling method. All experiments are repeated three times. We report the average MSE performance for all prediction lengths \{96, 192, 336, 720\}.
>
> |Dataset   |ShuffleMTM|ShuffleMTM-strict|
> |:----------:|:----------:|:-----------------:|
> |ETTh1     |**0.432**     |**0.432**            |
> |ETTh2     |**0.372**     |0.373            |
> |Exchange  |**0.353**     |0.354            |

---

> ### Comment · Reviewer_5ATj · 2024-11-25
>
> ​Thanks for the response, I will improve my score.​

---

> ### Author Response · Authors · 2024-11-25
>
> Thank you for your response. We greatly appreciate the reviewer’s decision to increase the rating towards acceptance, recognizing the academic contribution of ShuffleMTM to time series self-supervised research. As the reviewer has suggested, we have enhanced our manuscript by clarifying the positioning and technical validity of our work and improving the ablation study. Please refer to the blue-colored texts in the revised manuscript.
>
> Once again, we are pleased to engage with you as our reviewer and appreciate your incisive comments, which have significantly helped us improve our work. Please let us know if you have any further concerns, questions, or suggestions. We would be glad to incorporate your feedback.

---

### Official Review · Reviewer_wko4 · 2024-11-03

**Soundness:** 3
**Presentation:** 3
**Contribution:** 3
**Rating:** 8
**Confidence:** 4

**Summary:**

This paper proposes a simple strategy called ShuffleMTM for handling multi-channel time series. During pre-training, time series patches are randomly masked and shuffled across channels. Two views of each channel's univariate time series (the first is the one after random masking, and the second is after both random masking and shuffling) goes into a Siamese encoder followed by a decoder that aims to reconstruct the raw input time series. The paper demonstrates that ShuffleMTM works very well in practice.

**Strengths:**

- The proposed strategy is well-explained and very simple.
- The experimental results of Section 4 show that ShuffleMTM works quite well in practice.
- I think the experiments in Sections 5 and 6 are very helpful in checking on the robustness of ShuffleMTM to various conditions (such as limited data, missing data, hyperpameters) and in capturing patch-level and channel-level dependence.

**Weaknesses:**

- While ShuffleMTM is fairly simple to describe, I think the key question in my mind is, from a theoretical viewpoint, what sort of cross-channel structure can be captured by the specific shuffling performed, and what sort of cross-channel structure can *not* be captured. I think discussing this in more detail would really strengthen the paper. Just to give a simple example, suppose that the patch size is 1 so that each patch is just a single time step, and that it turns out that the true underlying structure is that each time step depends on *previous but not the current* time step's information in other channels. If I understand ShuffleMTM correctly, it seems like the shuffling done by ShuffleMTM in this case would not be helpful in capturing the cross-channel structure? Or maybe I'm misunderstanding something about ShuffleMTM? In any case, what I'm trying to do here is to just try to think of a setting where ShuffleMTM wouldn't work well (I'd guess that the authors could come up with various examples).
- Figure 2 I think is very important and helpful, but it took me some time to figure out what I am looking at. I'd suggest making the caption more descriptive and maybe adding some text (or some sort of drawing over the diagram) to better explain what is going on. If I understand the figure correctly, each colored cell is a patch. Blue/pink/green correspond to three channels. Gray cells are masked. Only some columns are randomly shuffled (I'm inclined to suggest that you circle or box the columns that were chosen for random shuffling across channels). Meanwhile, the two views that are inputs to the Siamese encoder correspond in this case to channel 1's data (actually stating this to be the case would be helpful to the reader).
- The experimental results would benefit from having error bars reported (such as standard deviation across some fixed number of experimental repeats with different random seeds) in Tables 1-4 and Figures 4-6. If there is a concern about tables becoming too large, one solution is to just defer, for instance, MAE results (with newly added error bars) to the appendix and keep MSE results (with newly added error bars) in the main paper.

Minor:
- There are a number of English grammar issues and typos. For example, in the first sentence of the introduction (line 33), the word "traffics" is used incorrectly. Perhaps this part of this first sentence can be reworded as "including energy, transportation, and medicine" (note that "medicines" here should be singular). At line ~66-67, "Unlike previous methods that recovers" should say "Unlike previous methods that recover". I'd suggest proofreading the paper carefully and using, for instance, a grammar checker.

**Questions:**

Please address the weakness points raised.

---

> ### Author Response · Authors · 2024-11-22
>
> ### W1. What sort of cross-channel structure can be captured by the specific shuffling performed, and what sort of cross-channel structure cannot be captured. I think discussing this in more detail would really strengthen the paper. Suppose that the patch size is 1 so that each patch is just a single time step, and that it turns out that the true underlying structure is that each time step depends on previous but not the current time step's information in other channels. It seems like the shuffling done by ShuffleMTM in this case would not be helpful in capturing the cross-channel structure? In any case, what I'm trying to do here is to just try to think of a setting where ShuffleMTM wouldn't work well (I'd guess that the authors could come up with various examples).
>
> ---
>
> -	Thank you for your careful review and insightful comment. To address your comment, we conducted simulation experiments to examine the cross-channel dependence that ShuffleMTM captures. Specifically, we used two synthetic data with different cross-channel dependencies and evaluated the forecasting performance of ShuffleMTM and PatchTST on the equivalent model configuration.
>
> -	The first synthetic dataset consists of three channels, each of which exhibits **lagged structures** relative to the others. We generate the first channel as sequence of length-16 patches, each representing a sinusoidal function with a unique frequency. Then, the second and third channels are derived by shifting the first channel by one-patch and two-patches lengths, respectively. From this simulation, this dataset naturally exhibits apparent **patch-level dependencies between channels** due to the lagged relationship, which are prevalent in real-world multivariate time series [1]. As the attention values of Transformers for multivariate time series forecasting tend to segment, i.e., close data points have similar attention weights, it is important to capture patch-level dependencies both within and across channels [2].
>
> -	Table 1 presents the forecasting performance of ShuffleMTM and PatchTST on the first synthetic dataset. Both models are evaluated in forecasting scenarios with prediction lengths of {32, 64, 96, 192} and a fixed input length of 96. The patch length is set to 16. The results indicate that ShuffleMTM consistently outperforms PatchTST across all prediction lengths, demonstrating its capability to capture patch-level dependencies between channels with lagged structures. This analysis confirms that ShuffleMTM can capture fine-grained cross-channel dependencies.
>
> ---
>
> Table 1. Evaluation of ShuffleMTM and PatchTST on the first synthetic dataset. All experiments are repeated three times.
>
> ||&nbsp;&nbsp;Shuffle|MTM&nbsp;|&nbsp;Patch|TST&nbsp;|
> |:---:|:---:|:---:|:---:|:---:|
> |Pred_len|MSE|MAE|MSE|MAE|
> |32|**1.023**|0.896|1.030|**0.895**|
> |48|**1.024**|**0.897**|1.037|0.901|
> |96|**1.026**|**0.898**|1.033|0.901|
> |192|**1.026**|**0.899**|1.031|0.901|
>
> ---
>
> [1] Zhao, L., & Shen, Y (2024). Rethinking Channel Dependence for Multivariate Time Series Forecasting: Learning from Leading Indicators. In The Twelfth International Conference on Learning Representations.
>
> [2] Zhang, Y., & Yan, J (2023). Crossformer: Transformer utilizing cross-dimension dependency for multivariate time series forecasting. In The eleventh international conference on learning representations.

---

> > ### Author Response · Authors · 2024-11-22
> >
> > ### W1. (Cont'd) What sort of cross-channel structure can be captured by the specific shuffling performed, and what sort of cross-channel structure cannot be captured. I think discussing this in more detail would really strengthen the paper. Suppose that the patch size is 1 so that each patch is just a single time step, and that it turns out that the true underlying structure is that each time step depends on previous but not the current time step's information in other channels. It seems like the shuffling done by ShuffleMTM in this case would not be helpful in capturing the cross-channel structure? In any case, what I'm trying to do here is to just try to think of a setting where ShuffleMTM wouldn't work well (I'd guess that the authors could come up with various examples).
> >
> > ---
> >
> > -	The second synthetic dataset also consists of three channels, all of which share the same long-term trend. First, we randomly generate three sequences of length-16 patches of sinusoidal function with distinct frequencies, ensuring no overlaps of local patterns between patches. Next, we generate a low-frequency sinusoidal waveform spanning the whole time series length as a long-term trend. Then, we add each sequence of length-16 patches to the long-term trend to get three channels that share the same trend. In this synthetic dataset, each time step in one channel is dependent on the previous time step as it has a long-term trend but is not dependent on current time step’s information in other channels as it does not share the local sinusoidal patterns. Hence, this dataset can be regarded as a length-16-patch case of the example that you provided.
> >
> > -	Table 2 presents the forecasting performance of ShuffleMTM and PatchTST on the second synthetic dataset. The evaluation setup is equivalent to the scenarios for the first synthetic dataset. The results indicate that ShuffleMTM demonstrates greater forecasting errors than PatchTST. As each channel contains long-term trends and is not dependent on the others in a local context, ShuffleMTM is less effective in the short-term forecasting. However, as the prediction length increases, the performance gap between two models becomes small. As the channels share the same long-term context, ShuffleMTM effectively captures the long-term dependence, resulting in enhanced performance in long-term forecasting.
> >
> > -	In summary, ShuffleMTM effectively captures fine-grained patch-level dependencies between channels such as lagged dependencies, as shown in the analysis on the first synthetic dataset. In time series that shares global temporal patterns, ShuffleMTM is ineffective in short-term forecasting, if each channel is not dependent on the others in a local context. However, ShuffleMTM becomes effective in long-term forecasting on this data if the channels share the long-term context.
> >
> > -	We sincerely appreciate your valuable comment, which has helped us to investigate the ability of ShuffleMTM to capture cross-channel dependencies under various patterns. These experiments not only address the concerns you raised but also enhance the clarity of our work. We will include this simulation experiment on the revised manuscript. Thank you again.
> >
> >
> > ---
> >
> > Table 2. Evaluation of ShuffleMTM and PatchTST on the second synthetic dataset. All experiments are repeated three times.
> >
> > ||&nbsp;&nbsp;Shuffle|MTM&nbsp;|&nbsp;Patch|TST&nbsp;|
> > |:---:|:---:|:---:|:---:|:---:|
> > |Pred_len|MSE|MAE|MSE|MAE|
> > |32|0.292|0.477|**0.284**|**0.470**|
> > |48|0.292|0.477|**0.288**|**0.474**|
> > |96|0.294|0.479|**0.292**|**0.478**|
> > |192|0.296|0.480|**0.294**|**0.479**|

---

> ### Author Response · Authors · 2024-11-22
>
> ### W2. Figure 2 I think is very important and helpful, but it took me some time to figure out what I am looking at. I'd suggest making the caption more descriptive and maybe adding some text (or some sort of drawing over the diagram) to better explain what is going on. If I understand the figure correctly, each colored cell is a patch. Blue/pink/green correspond to three channels. Gray cells are masked. Only some columns are randomly shuffled (I'm inclined to suggest that you circle or box the columns that were chosen for random shuffling across channels). Meanwhile, the two views that are inputs to the Siamese encoder correspond in this case to channel 1's data (actually stating this to be the case would be helpful to the reader).
>
> ---
>
> -	Thank you for your detailed review and valuable feedback. We acknowledge that Figure 2 is not clearly illustrated enough to explain the architecture of our method despite its importance in the manuscript. Based on your feedback, we will enhance the presentation of Figure 2.
>
> -	As shown in Figure 2, ShuffleMTM proposes a novel self-supervised pre-training task to capture cross-channel dependence by reconstructing each channel through the integration of original and shuffled views of masked series. To integrate both views of masked series, our shuffling method generates the original and shuffled masked series, Siamese encoders simultaneously encode both views, and a cross-attention decoder integrates the cross-time and cross-channel dependencies encoded in the representations to reconstruct each channel. Thus, the shuffling method, Siamese encoders, and cross-attention decoder are essential components for capturing cross-time and cross-channel dependencies within the channel-independent strategy. Leveraging this design, ShuffleMTM extends the channel-independent reconstruction task to effectively capture both cross-time and cross-channel dependencies.
>
>
> -	Once again, thank you for your detailed suggestion, which has led to improve the clarity of our framework and presentation of our work.

---

> ### Author Response · Authors · 2024-11-22
>
> ### W3. The experimental results would benefit from having error bars reported (such as standard deviation across some fixed number of experimental repeats with different random seeds) in Tables 1-4 and Figures 4-6. If there is a concern about tables becoming too large, one solution is to just defer, for instance, MAE results (with newly added error bars) to the appendix and keep MSE results (with newly added error bars) in the main paper.
>
> ---
>
> -	Thank you for your detailed suggestion. Based on your feedback, we will provide the experimental results with the standard deviation. Due to the page limit, we will upload these results in the appendices. Table 3 presents the subset of Table 1 in the submitted manuscript with the standard deviation, which will be reported in our revised manuscript.
>
> -	Once again, thank you for your careful review and detailed suggestion.
>
> ---
>
> Table 3. Example of the experimental results with the standard deviation. This is the case of multivariate forecasting performance on ETTh1. Bold and italic letters indicate the best and second-best performance, respectively.
>
>
> |||&nbsp;&nbsp;&nbsp;&nbsp;&nbsp;ShuffleMTM||&nbsp;&nbsp;&nbsp;&nbsp;&nbsp;TimeSiam||&nbsp;&nbsp;&nbsp;&nbsp;&nbsp;&nbsp;&nbsp;&nbsp;&nbsp;PITS||&nbsp;&nbsp;&nbsp;&nbsp;&nbsp;PatchTST||&nbsp;&nbsp;&nbsp;&nbsp;&nbsp;SimMTM||
> |:---:|:---:|:---:|:---:|:---:|:---:|:---:|:---:|:---:|:---:|:---:|:---:|
> |Dataset|Pred_len|MSE|MAE|MSE|MAE|MSE|MAE|MSE|MAE|MSE|MAE|
> |ETTh1|96|*0.376* (&pm;0.0020)|0.397 (&pm;0.0014)|	0.379 (&pm;0.0008)|	0.402 (&pm;0.0007)|	0.377 (&pm;0.0007)|	*0.395* (&pm;0.0006)|	0.379 (&pm;0.0067)|	0.399 (&pm;0.0035)|	**0.367** (&pm;0.0008)|	**0.389** (&pm;0.0005)|	0.386 (&pm;0.0013)|	0.405 (&pm;0.0007)|
> ||192|**0.420** (&pm;0.0014)|*0.425* (&pm;0.0007)|	0.425 (&pm;0.0028)|	0.431 (&pm;0.0018)|	0.430 (&pm;0.0021)|	*0.425* (&pm;0.0010)|	0.425 (&pm;0.0013)|	0.427 (&pm;0.0019)|	0.424 (&pm;0.0014)|	**0.423** (&pm;0.0011)|	0.443 (&pm;0.0007)|	0.437 (&pm;0.0004)|
> ||336|**0.456** (&pm;0.0019)|**0.446** (&pm;0.0009)|	*0.459* (&pm;0.0016)|	0.451 (&pm;0.0017)|	0.478 (&pm;0.0030)|	*0.448* (&pm;0.0026)|	0.470 (&pm;0.0055)|	**0.446** (&pm;0.0021)|	0.473 (&pm;0.0025)|	0.456 (&pm;0.0012)|	0.489 (&pm;0.0043)|	0.460 (&pm;0.0030)|
> ||720|**0.474** (&pm;0.0095)|*0.471* (&pm;0.0049)|	*0.475* (&pm;0.0098)|	0.478 (&pm;0.0052)|	0.499 (&pm;0.0195)|	0.475 (&pm;0.0103)|	0.482 (&pm;0.0040)|	**0.466** (&pm;0.0020)|	0.494 (&pm;0.0041)|	0.493 (&pm;0.0025)|	0.508 (&pm;0.0109)|	0.494 (&pm;0.0062)|

---

> > ### Author Response · Authors · 2024-11-22
> >
> > ### Minor. There are a number of English grammar issues and typos. For example, in the first sentence of the introduction (line 33), the word "traffics" is used incorrectly. Perhaps this part of this first sentence can be reworded as "including energy, transportation, and medicine" (note that "medicines" here should be singular). At line ~66-67, "Unlike previous methods that recovers" should say "Unlike previous methods that recover". I'd suggest proofreading the paper carefully and using, for instance, a grammar checker.
> >
> > ---
> >
> > -	Thank you for your review and thoughtful feedback. We acknowledge that the submitted manuscript contains English grammar mistakes. In response to your comment, we have proofread the paper and used a grammar checker to correct errors and improve the clarity of the sentences. We kindly encourage you to review our revised manuscript. Thank you again for your kind suggestion.
> >
> > -	Once again, we are grateful for your detailed comments, which have facilitated to improve the presentation and clarity of our work.

---

> > > ### Comment · Reviewer_wko4 · 2024-11-25
> > >
> > > Thanks for the very detailed response to my review. After reading your response as well as your response to the other reviews, I am increasing my score.

---

> ### Author Response · Authors · 2024-11-25
>
> Thank you for your response. We greatly appreciate the reviewer’s decision to increase the rating towards acceptance and suggestions for improving the quality of our work. We have included the simulation experiment and the full forecasting results with standard deviations in the appendices. Please refer to Sections E and F. We also enhance the presentation of Figure 2 by adding the detailed explanation on the colors and notations, as well as in Figure 1, so that we can clearly position our work in the existing masked time-series modeling research and improve the clarity of our framework. Please refer to the captions of Figures 1 and 2 in L62-64 and L177-182.
>
> Once again, we are pleased to engage with you as our reviewer and greatly appreciate your insightful comments, which have significantly enhanced our work. Please let us know if you have any further concerns, questions, or suggestions. We would be happy to address your feedback.

---

> > ### Comment · Reviewer_wko4 · 2024-12-03
> >
> > Note that I have read the recent threads with reviewers g464 and riEy. I would emphasize to the authors a point I had already indicated in my original review (*"While ShuffleMTM is fairly simple to describe, I think the key question in my mind is, from a theoretical viewpoint, what sort of cross-channel structure can be captured by the specific shuffling performed, and what sort of cross-channel structure can not be captured. I think discussing this in more detail would really strengthen the paper."*): basically, the paper needs to very clearly articulate when the proposed ShuffleMTM method should be expected to work well and when it should not, with clear examples given of both.
> >
> > At least for me personally, I think it's fine if ShuffleMTM does not outperform all baselines for all datasets (note that I also have the same view when I review other papers in that if the paper is about a proposed method, I do not think the proposed method needs to definitively outperform all other baselines on all datasets); it should show reasonably competitive performance against some now-standard SOTA baselines on ideally most datasets considered (which I think is the case for ShuffleMTM). However, I think justifying *why* a proposed method works particularly well in some contexts and poorly in others compared to various methods is extremely important. To this end, I think that the additional synthetic experiments that have been run during this author/reviewer discussion period have been very informative and help explain some of the experimental results we're seeing on specific real datasets (such as the electricity one).
> >
> > In any case, I think reviewers g464 and riEy have provided feedback that should lead to the paper getting even stronger, with more experimental results and explanations of why ShuffleMTM does not perform well in some settings. (The comparison with Moirai and Timer is also helpful.)
> >
> > Overall, my score still stays the same where I favor acceptance.

---

> > > ### Author Response · Authors · 2024-12-04
> > >
> > > Dear reviewer wko4,
> > >
> > > We sincerely appreciate your recognition of the academic contributions of our work to the time series learning literature. In response to your insightful comment, we gained a deeper understanding of the mechanisms and strengths of our approach, which allowed us to further enhance its technical contributions. Based on your suggestion, we are continuing to explore advancements in time series self-supervised learning. Thank you once again for your valuable feedback.

---

### Official Review · Reviewer_g464 · 2024-11-04

**Soundness:** 3
**Presentation:** 3
**Contribution:** 2
**Rating:** 5
**Confidence:** 4

**Summary:**

This paper addresses the challenge of capturing cross-channel dependencies in multivariate time series pre-training. Unlike previous channel-independent methods, the proposed model, ShuffleMTM, masks a multivariate instance to create an original masked view and then shuffles the channels within each time step to generate a shuffled masked view. The original view serves as the query, while the shuffled view serves as the key and value in a cross-attention decoder, with the objective of reconstructing the original instance. Experimental results demonstrate that ShuffleMTM outperforms several channel-independent baselines in both forecasting and classification tasks. Additional experiments assess its performance in limited data scenarios, robustness, and other settings.

**Strengths:**

1. The paper is well-structured and accessible, making it easy to follow.

2. The focus on capturing cross-channel dependencies in multivariate time series pre-training is both important and relevant.

3. A capacity-robustness analysis is provided, offering insights into the model’s forecasting accuracy as well as its robustness, training-testing gap, and other performance metrics.

**Weaknesses:**

1. Some crucial parts about the proposed model is missing: how to adapt the pre-trained model to downstream tasks. Is the shuffled view utilized during fine-tuning? If not, it is unclear how the channel-independent encoder effectively captures cross-channel dependencies.

2. The proposed shuffling method is  not straightforward. Why not simply apply 2D attention, as done in Crossformer, to the masked patches, or flatten the time series into a 1D sequence and use self-attention? What specific advantages does shuffling offer over these approaches, aside from efficiency?

3. The ablation study could be improved. The following experiment should be included: removing the shuffled view by using the original view as the query, key, and value in the decoder, and comparing this configuration with the proposed method, which uses the original view as the query and the shuffled view as the key and value.

4. In the forecasting task, the lookback window is fixed to 96, which is shorter than the future window to forecast. It would be valuable to assess the model’s performance with an extended lookback window.

5. Please explain Section 6.1 in detail, including:
 - How to comupte the "self-attention map" and "patch-correlation" in patch-level dependence paragraph as there is no cross-channel attention in the proposed method?
 - How to compute the "channel correlations in the raw time series" in channel-level dependence?

**Questions:**

Please see the Weaknesses section for detailed questions.

---

> ### Author Response · Authors · 2024-11-22
>
> ### W1. Some crucial parts about the proposed model is missing: how to adapt the pre-trained model to downstream tasks. Is the shuffled view utilized during fine-tuning? If not, it is unclear how the channel-independent encoder effectively captures cross-channel dependencies.
>
> ---
>
> - Thank you for your careful review and important question. During the pre-training stage, ShuffleMTM encodes both the original and shuffled views of input time series using Siamese networks and performs a channel-independent reconstruction task by simultaneously learning temporal and cross-channel dependencies. Since ShuffleMTM captures cross-channel dependencies of multivariate time series in a channel-independent setting during pre-training, the shuffled view is not utilized during fine-tuning. Instead, the ShuffleMTM encoder projects the original time series into a deep representation, which is transferred to downstream tasks during the fine-tuning process.
>
> - As such, its representations effectively capture the cross-channel dependencies of multivariate time series. As demonstrated in the cross-channel dependence analysis in Section 6.1, the ShuffleMTM encoder produces an attention score on the shuffled patched series that more closely aligns with its correlation matrix than other channel-independent MTM encoders. This finding suggests that pre-training with shuffled masked series effectively captures cross-channel dependencies in channel-independent encoding. Please note that our random shuffling, which dynamically introduces unmasked patches from different channels into masked series, enables the capture of various patch-level dependencies across channels. However, its random nature is not suitable during fine-tuning and inference, when exact values need to be forecasted.
>
> - Once again, thank you for your thorough comment, which has helped us to enhance the presentation of our work. Based on your feedback, we will clarify the mechanism of ShuffleMTM during the fine-tuning process in the revised manuscript.

---

> > ### Comment · Reviewer_g464 · 2024-11-25
> > **Confusion about capturing cross-channel dependencies in channel-independent encoding**
> >
> > In my understanding, during fine-tuning, the encoder processes each channel independently. Therefore, a specific channel is encoded regardless of the other channels. More specifically, even though I replace other channels with pure noise, the encoding results still do not change. I do not understand how this method can capture channel dependence.

---

> ### Author Response · Authors · 2024-11-22
>
> ### W2. The proposed shuffling method is not straightforward. Why not simply apply 2D attention, as done in Crossformer, to the masked patches, or flatten the time series into a 1D sequence and use self-attention? What specific advantages does shuffling offer over these approaches, aside from efficiency?
>
> ---
>
> - Thank you for your careful review and constructive comment. In response to your comment, we have been able to further differentiate ShuffleMTM in the time series literature that learns cross-channel dependence and improve the technical contribution of our work. Please see our response below.
>
> - Crossformer [1] proposes a sequential two-stage attention mechanism for multivariate time series to capture cross-channel dependence. However, applying cross-time and cross-variate attention sequentially struggles to model the dynamic dependencies between patches from different channels. Crossformer simply applies self-attention horizontally and vertically, which is ineffective for learning dependencies between patches from other channels at lagged locations. In contrast, shuffling unmasked patches across channels dynamically introduces patches at lagged locations, enabling the channel-independent encoder to capture various patch-level dependencies across channels at different positions.
>
> - Flattening multivariate time series and applying 1D self-attention to patches across all channels [2] simplifies the reconstruction task, allowing the encoder and decoder to access unmasked patch embeddings from different channels with identical temporal information. This simplification, however, can allow the encoder to learn spurious information, hindering the training of the encoder. Shuffling unmasked patches from other channels at different locations integrates cross-channel information into the channel-independent reconstruction task without relying on patches with identical temporal information.
>
> - The fundamental advantage of our shuffling method over these approaches is that ShuffleMTM achieves the advantages of both channel-independent and channel-dependent methods. The channel-independent strategy flattens the channel dimensions into the same batch and processes various patterns in each channel independently, improving adaptability to diverse temporal patterns within the channel [3]. Consequently, channel-independent models achieve greater robustness, whereas channel-dependent forecasting models are known for their higher capacity [4]. By capturing cross-channel dependence in the channel-independent strategy through the shuffling method, ShuffleMTM achieves both greater forecasting capacity and robustness compared to channel-independent methods (refer to the capacity-robustness analysis in Section 6.2). However, adopting 2D attention or applying 1D attention to the flattened time series is incompatible with the channel-independent method.
>
> - In addition to the advantages of shuffling for learning cross-channel dependence, both two-stage attention mechanisms and flattened time series approaches require the entire multivariate time series to fully capture cross-channel dependencies. If some channels are missing, discrepancies between training and testing phases arise, leading to incomplete cross-channel information. Moreover, the high computational cost of Transformer-based methods makes self-attention on flattened time series computationally expensive. These approaches are thus inefficient for learning cross-channel dependencies through self-attention.
>
> - We sincerely appreciate your valuable feedback, which has helped us strengthen the positioning of our work in the literature. Thanks to your comment, we believe we can further enhance the technical contribution and novelty of our work. We will incorporate this discussion into the revised manuscript.
>
> ---
>
> [1] Zhang, Y., & Yan, J (2023). Crossformer: Transformer utilizing cross-dimension dependency for multivariate time series forecasting. In The eleventh international conference on learning representations.
>
> [2] Liu, J., Liu, C., Woo, G., Wang, Y., Hooi, B., Xiong, C., & Sahoo, D. (2024). UniTST: Effectively Modeling Inter-Series and Intra-Series Dependencies for Multivariate Time Series Forecasting. arXiv preprint arXiv:2406.04975.
>
> [3] Nie, Y., Nguyen, N. H., Sinthong, P., & Kalagnanam, J. (2023) A Time Series is Worth 64 Words: Long-term Forecasting with Transformers. In The Eleventh International Conference on Learning Representations.
>
> [4] Han, L., Ye, H. J., & Zhan, D. C. (2024). The capacity and robustness trade-off: Revisiting the channel independent strategy for multivariate time series forecasting. IEEE Transactions on Knowledge and Data Engineering.

---

> ### Author Response · Authors · 2024-11-22
>
> ### W3. The ablation study could be improved. The following experiment should be included: removing the shuffled view by using the original view as the query, key, and value in the decoder, and comparing this configuration with the proposed method, which uses the original view as the query and the shuffled view as the key and value.
>
> ---
>
> -	Thank you for your valuable feedback. Following your comment, we evaluated the performance of ShuffleMTM after removing the shuffled view and using the original view as the query, key, and value in the decoder. This variant of ShuffleMTM in fact reduces to PatchTST [3] with a self-attention decoder, which originally utilizes the original masked series and decodes the representation with a linear layer. Comparing this variant with ShuffleMTM demonstrates the effectiveness of utilizing the shuffled masked series for pre-training channel-independent MTM. The purpose of this experiment aligns with the ablation study on the decoder and reconstruction targets in Section 5 of the submitted manuscript. Thank you again, and please see the result as follows.
>
> -	Table 1 presents the MSE forecasting performance of ShuffleMTM and ShuffleMTM without the shuffled view. The results show that removing the shuffled view and using a self-attention decoder with only the original view decreases the forecasting performance of ShuffleMTM. These findings are consistent with the ablation study on the decoder and reconstruction targets in Figure 3 of the submitted manuscript. The results in Table 1 of this rebuttal and Figure 3 in the manuscript indicate that integrating the original and shuffled views of masked series is crucial for forecasting performance, as it facilitates the integration of spatial and temporal information in multivariate time series. Thus, the shuffling method and Siamese encoders are essential components for capturing cross-channel dependencies in the channel-independent strategy.
>
> -	Once again, we sincerely appreciate your insightful comment, which was instrumental to enhance the technical validity of the proposed method. Based on your feedback, we will improve the ablation study by incorporating this experimental result into the analysis of the decoder and reconstruction targets in the manuscript.
>
> ---
>
> Table 1.  Comparison with ShuffleMTM after removing the shuffled view on ETTh2 and Weather. All experiments are repeated three times.
>
> |   |   |   &nbsp;Shuffle | MTM&nbsp;&nbsp;&nbsp;&nbsp;  |  ShuffleMTM  |  w/o shuffle |
> |:---:|:---:|:---:|:---:|:---:|:---:|
> | Dataset      | Length | MSE | MAE | MSE | MAE |
> |ETTh2         |96    |**0.288** |**0.338**|0.289 |**0.338** |
> |              |192   |**0.368**|**0.390**|0.370 |0.391 |
> |              |336   |**0.412** |**0.426**|**0.412** |**0.426** |
> |              |720   |**0.421** |**0.441** |0.422 |0.442 |
> |              |avg   |**0.372** |**0.399** |0.373 |**0.399** |
> |Electricity   |96    |**0.161** |**0.248** |0.162 |0.250 |
> |              |192   |**0.170** |**0.257** |0.172 |0.258 |
> |              |336   |**0.186** |**0.274** |0.188 |0.275 |
> |              |720   |0.228 |**0.310** |**0.227** |**0.310** |
> |              |avg   |**0.186** |**0.272** |0.187 |0.273 |

---

> ### Author Response · Authors · 2024-11-22
>
> ### W4. In the forecasting task, the lookback window is fixed to 96, which is shorter than the future window to forecast. It would be valuable to assess the model’s performance with an extended lookback window.
>
> ---
>
> -	Thank you again for your careful review and constructive suggestions. Following your comment, we conducted an ablation experiment on the lengths of the lookback window. Specifically, we evaluated ShuffleMTM on extended lookback windows {96, 192, 336, 512} using the ETTm1 and Weather datasets. As shown in Table 2, ShuffleMTM demonstrates superior or competitive performance compared to channel-independent self-supervised baselines on extended lookback windows as well. Furthermore, ShuffleMTM outperforms iTransformer—a state-of-the-art forecasting model that explicitly captures cross-channel dependence through variate-wise attention—on all lookback windows. These experimental results demonstrate the robust forecasting performance of ShuffleMTM across various input lengths.
>
> -	We are grateful for your constructive comments, which have facilitated the validation of our work's robustness in various forecasting scenarios. We will include the ablation study on lookback window size in the revised manuscript.
>
> ---
>
> Table 2.  Ablation experiment on the lengths of the lookback window. All experiments are repeated three times. Bold and italic letters indicate the best and second-best performance, respectively.
>
> |  |  |Shuffle  |MTM |  &nbsp;&nbsp;&nbsp;&nbsp;&nbsp;&nbsp;PI| TS&nbsp;&nbsp;&nbsp;&nbsp;&nbsp;&nbsp; |Patch  |TST  |&nbsp;&nbsp;Sim  | MTM&nbsp; &nbsp; |Time  | Siam |iTrans  | former |
> |:--------:|:------:|:------:|:------:|:------:|:------:|:------:|:------:|:------:|:------:|:------:|:------:|:------:|:------:|
> |Dataset |Length|MSE   |MAE   |MSE   |MAE   |MSE   |MAE   |MSE   |MAE   |MSE   |MAE   |MSE   |MAE   |
> |ETTm1   |96    |*0.379* |**0.394** |0.388 |0.399 |**0.376** |*0.395* |0.386 |0.401 |**0.376** |0.396 |0.408 |0.412 |
> |        |192   |**0.353** |*0.383* |0.354 |**0.382** |**0.353**|**0.382** |0.354 |0.384 |0.356 |0.384 |0.371 |0.394 |
> |        |336   |**0.344** |**0.377** |0.355 |0.385 |*0.345* |*0.381* |0.356 |0.387 |0.351 |0.384 |0.369 |0.396 |
> |        |512   |*0.350* |**0.382** |0.354 |0.386 |**0.347** |**0.382** |0.359 |0.391 |0.356 |0.390 |0.368 |0.398 |
> |Weather |96    |**0.250** |**0.275** |0.263 |0.284 |0.252 |*0.276* |*0.251* |0.277 |0.255 |0.278 |0.260 |0.281 |
> |        |192   |**0.234** |**0.264** |0.246 |0.274 |*0.236* |*0.266* |*0.236* |0.268 |*0.236* |0.267 |0.248 |0.277 |
> |        |336   |**0.226** |**0.261** |0.235 |0.270 |*0.227* |*0.263* |0.230 |0.268 |0.229 |0.266 |0.238 |0.273 |
> |        |512   |**0.224** |**0.262** |0.255 |0.269 |*0.225* |**0.262** |0.231 |0.270 |0.227 |*0.267* |0.247 |0.279 |

---

> ### Author Response · Authors · 2024-11-22
>
> ### W5.1. How to compute the "self-attention map" and "patch-correlation" in patch-level dependence paragraph as there is no cross-channel attention in the proposed method?
>
> ---
>
> -	Thank you for your detailed feedback. We examined that ShuffleMTM captures cross-channel dependence in a channel-independent encoding, focusing on fine-grained dependence between channels. In analyzing fine-grained cross-channel dependence, we concentrated on the patches. First, we randomly shuffled the patches across different channels that are positioned at the same index. Subsequently, a self-attention map of the shuffled patched series is obtained from the ShuffleMTM encoder. At the same time, we calculated the correlation coefficient between the patches in the shuffled patched series, producing the patch-correlation matrix. Then, we analyzed the cosine similarity between the self-attention map and patch-correlation matrix of the shuffled patched series. A Transformer encoder that successfully learns the dependencies of input patches should yield attention scores consistent with the correlation structure [5].
>
> -	Through the patch-level analysis, ShuffleMTM achieves higher cosine similarity between the attention scores and patch-correlation matrices compared to other channel-independent models. In addition, ShuffleMTM attains a higher cosine similarity than a single-branch version of ShuffleMTM with reconstructing the original time series using only shuffled masked series. These findings confirm that pre-training with shuffled masked series effectively captures cross-channel dependence in channel-independent encoding and utilizing Siamese networks is essential for integrating spatial information from shuffled series representations into the channel-independent reconstruction task.
>
> -	We appreciate your detailed review and comments, which have helped improve the presentation of our work and the clarity of our analyses. Based on your suggestions, we will clarify the procedure for computing the measures used in our analyses.
>
> ---
>
> [5] Liu, Y., Hu, T., Zhang, H., Wu, H., Wang, S., Ma, L., & Long, M. (2024). iTransformer: Inverted Transformers Are Effective for Time Series Forecasting. In The Twelfth International Conference on Learning Representations.

---

> ### Author Response · Authors · 2024-11-22
>
> ### W5.2. How to compute the "channel correlations in the raw time series" in channel-level dependence?
>
> ---
>
> -	Thank you for your constructive feedback. We examined that ShuffleMTM captures coarse-grained dependence between channels in the channel-independent encoding. In analyzing coarse-grained cross-channel dependence, this time, we focused on the entire univariate series. First, we computed the channel correlation as the correlation coefficient between channels in a raw multivariate time series. We also calculated the pairwise distances between channel embeddings of ShuffleMTM, applying max pooling to extract these embeddings from a series of patch embeddings, as was commonly done in time series self-supervised methods [3]. By comparing the channel correlation matrix and pairwise distance matrix of the channel embeddings, we observed that two matrices align closely with each other, These experimental results confirm that pre-training with shuffled series captures dependent structure of channels present in the raw time series.
>
> -	Once again, we appreciate your detailed review and comments, which have helped improve the presentation of our work and the clarity of our analyses. Based on your suggestions, we will clarify the procedure for computing the measures used in our analyses.

---

> ### Comment · Reviewer_g464 · 2024-11-26
> **Concerns about capturing cross-channel dependency in channel-independent encoding**
>
> Thanks for the author's rebuttal. My fundamental concern is how this channel-independent encoding can capture channel dependency. As my last comment states, a specific channel's forecasting result will not change during inference, even when all other channels are replaced by pure noise. Moreover, the ablation study shows that the improvement over the ablated version without shuffle is insignificant.
>
> Therefore, I have reason to question whether the model captures channel dependence as claimed. If the authors do not directly address the question above, I will lower the score to 3.

---

> ### Author Response · Authors · 2024-11-26
>
> Thank you again for your precise review and constructive comments. We understand your question regarding how our method captures channel dependencies, particularly since the encoder processes each channel independently during fine-tuning and inference. In response to your question, we would like to clarify the following points. We sincerely hope that these clarifications address your concerns and improve the presentation of our contributions.
>
> First of all, we would like to highlight that **our work aims to address the limitation of channel-independent MTM (masked time series modeling) research, where the pre-training stage cannot explicitly capture cross-channel dependencies**. In existing channel-independent MTMs, each channel is processed independently during both the pre-training and fine-tuning stages, preventing the utilization of cross-channel information.
>
> **The key contribution of ShuffleMTM lies in its pre-training approach, which is specifically designed to integrate cross-channel dependencies into the channel-independent encoder**. During pre-training, ShuffleMTM shuffles the multivariate time series to encourage the model to learn relationships across different channels, embedding cross-channel information into the encoder's parameters. As a result, the learned representations inherently capture cross-channel information. Although the encoder processes channels independently during fine-tuning and inference, this allows the model to capture and utilize channel dependencies without altering the channel-independent nature of the encoder during fine-tuning. Thus, while the encoder processes each channel independently during inference, the pre-trained representations reflect the cross-channel dependencies already learned during pre-training.
>
> As such, the contribution of our research is to develop **a novel MTM approach that enables channel-independent models to incorporate cross-channel information during encoding**. The MTM pre-training task of reconstructing masked segments has been demonstrated to be crucial and effective for capturing cross-time dependencies, thereby enhancing time series forecasting and classification tasks [1, 2]. The key contribution of ShuffleMTM lies in its ability to reflect channel dependencies during pre-training, aligning with other MTM studies that focus on enhancing the pre-training stage.
>
> We acknowledge that in the example you suggest, our model might not encode sudden noise from other channels in the inference stage. Given the above clarification, however, the encoder's outputs can be influenced by the cross-channel dependencies learned during pre-training, even if the other channels are noisy during inference. More specifically, during fine-tuning, the ShuffleMTM encoder implicitly learns those errors since it processes all channels in multivariate time series within the same batch. Meanwhile, **all channel-independent MTM methods involve the same operations, as the channel-independent encoder inevitably processes each channel independently during inference. As such, we emphasize that pre-training to integrate cross-channel information into the channel-independent encoder is an indispensable approach for MTM.** We believe that our pre-training method significantly enhances the model’s ability to integrate cross-channel information during downstream tasks, even if certain edge cases, like the one you describe, may exhibit such behavior.
>
> Meanwhile, based on your insights, we could explore future research directions to develop fine-tuning methods that can capture sudden changes from other channels (i.e., not only incorporating the cross-channel information in the pre-training stage but also considering it in the fine-tuning and inference stage as well). We believe such research could also benefit all channel-independent encoders and further enhance adaptability to various cross-channel patterns during fine-tuning and inference. Nonetheless, **we emphasize that our work represents a significant extension of the current MTM literature by enabling the incorporation of cross-channel information during pre-training.** This contribution provides a basis for future studies on time series deep learning models to address cross-channel patterns in different ways.
>
> Thank you again for your valuable feedback, which has helped us clarify our work and identify new directions for future research. We will include this discussion in the conclusion as part of future work. We sincerely hope we have clarified our contributions and improved the presentation of our work by addressing your comments.
>
> [1] Dong et al., (2024). Simmtm: A simple pre-training framework for masked time-series modeling. Advances in Neural Information Processing Systems.
>
> [2] Lee et al., (2024). Learning to Embed Time Series Patches Independently. In The Twelfth International Conference on Learning Representations.

---

> > ### Comment · Reviewer_g464 · 2024-11-26
> >
> > Thank you for the response. I will summarize it briefly as follows:
> >
> > 1. The pre-training process gets access to other channels through shuffling and cross-attention so that the co-occurrence knowledge among channels is encoded into model parameters.
> > 2. During inference, given a channel, the model can implicitly utilize the co-occurrence knowledge for forecasting, regardless of other channels.
> >
> > However, I remain unconvinced, as the effect of this co-occurrence appears to be quite weak. In fact, previous channel-independent models could make similar claims, as channels are aligned, coupled, and fed into the model together during training. This co-occurrence influences the optimization process of gradient descent. Furthermore, the most significant issue is that historical data from other channels does not explicitly impact the forecasting during inference. As reviewer riEy also noted, there is no substantial change in performance after removing the shuffle operation, which raises doubts about whether the cross-channel dependency is truly captured as claimed.
> >
> > Given these concerns, I have maintained my original score of 5. I think this work is not yet ready for publication.

---

> > > ### Comment · Reviewer_g464 · 2024-12-03
> > >
> > > I appreciate the authors' effort in providing additional experimental results, but my main concern is related to the core idea, which remains unsolved, that is: **It is impossible to capture cross-channel dependency in a channel-independent way**. By raising the example of replacing other channels with pure noise in the previous response, I mean **cross-channel dependency cannot be captured without knowing what other channels are**.
> > >
> > > To clarify this, let's consider a much more simple example: for a 2-dimensional vector $(x,y)$ the ground truth $R^2 \rightarrow R^2$function is $f(x, y) = (x+y, x-2y)$. Using the idea of this work, $f$ can be approximated independently by $f_\theta(x,y) = (f_\theta(x), f_\theta(y))$. How could it be possible to model $x+y$ with only $f_\theta(x)$? Authors may argue that during the pre-training stage, the relation between $x,y$ can be encoded into network parameters such that $y = g_\phi(x)$ and $x+y$ can be approximated by $f_\theta(x, g_\phi(x))$. If so, why do we need cross-channel dependency?
> > >
> > > In general, cross-channel dependency is important because it provides supplementary information that a channel does not have. However, independent fine-tuning neglects other channels and cannot capture any dependency among channels. I agree with reviewer riEy that this work is not yet ready for publication and authors should rethink their core idea.

---

> ### Author Response · Authors · 2024-11-27
>
> Thank you for your active engagement in this review and rebuttal. Once again, we appreciate your constructive comments.
>
> Regarding the comparison between ShuffleMTM and its variant without the shuffled view, we analyzed the small performance differences on the ETTh2 and Electricity datasets. Shao et al. (2022) [1] found that the ETT and Electricity datasets exhibit low spatial dependencies. We believe the small performance gap between ShuffleMTM and the variant is due to the low spatial dependencies inherent in these datasets. To promptly address reviewers’ comments during the rebuttal period with limited resources available, we initially selected relatively small datasets that were feasible for immediate analysis. We now understand that selecting ETT and Electricity datasets was not ideal for the purpose of that ablation study. Recognizing the importance of evaluating our method on datasets with varying spatial dependencies, we have now extended our experiments as detailed below, taking the opportunity provided by the rebuttal extension for ICLR 2025.
>
> Thanks to the extension of the rebuttal for ICLR 2025, we have proactively conducted additional experiments to compare ShuffleMTM with its variant without the shuffled view, so that we can address your concern. We generated two synthetic datasets: the first consists of three channels, each exhibiting lagged structures relative to one another. The second dataset also consists of three channels, all of which share the same long-term trend but exhibit distinct local patterns. Thus, the first dataset has high local dependencies between channels, whereas the second has weak local dependencies. Table 3 below presents the forecasting performance of ShuffleMTM and its variant without the shuffled view on these two synthetic datasets. Both models were evaluated in forecasting scenarios with prediction lengths of {32, 64, 96, 192} and a fixed input length of 96, with the patch length set to 16. ShuffleMTM outperformed its variant across both datasets, with a larger performance gap observed in the first dataset than in the second. These experiments suggest that **the shuffling method enables the channel-independent encoder to capture fine-grained dependencies between channels.** However, **the shuffling method is less effective on datasets with weak local dependencies compared to those with strong local dependencies**. Based on the experimental results from the synthetic datasets and this analysis, we explain that **the small performance difference between the two models on ETTh2 and Electricity is due to the low spatial dependencies in these datasets.**
>
> Furthermore, to confirm the abovementioned finding and to further validate the utility of ShuffleMTM in capturing cross-channel dependencies, we will compare ShuffleMTM with its variant without the shuffled view using other benchmark datasets with high spatial dependencies as well. For example, we will use the PEMS dataset, which collects traffic flow data in urban road networks and is known for its strong spatial dependencies. We will conduct this comparison and upload the results during the rebuttal period. We kindly ask you for a few days to complete this experiment.
>
> Table 3. Comparison between ShuffleMTM and its variant without the shuffled view on two synthetic datasets. All experiments were repeated three times.
>
> |Dataset||&nbsp;&nbsp;Shuffle|MTM&nbsp;| &nbsp;&nbsp;ShuffleMTM  |  w/o shuffle &nbsp; &nbsp;|
> |:---:|:---:|:---:|:---:|:---:|:---:|
> ||Pred_len|MSE|MAE|MSE|MAE|
> |Synthetic|32|**1.023**|**0.896**|1.034|0.899|
> |data 1|48|**1.024**|**0.897**|1.034|0.899|
> ||96|**1.026**|**0.898**|1.033|0.898|
> ||192|**1.026**|0.899|1.029|**0.897**|
> |Synthetic|32|**0.292**|**0.477**|0.293|0.479|
> |data 2|48|**0.292**|**0.477**|0.293|0.479|
> ||96|**0.294**|**0.479**|0.295|**0.479**|
> ||192|**0.296**|**0.480**|**0.296**|**0.480**|
>
>
> [1] Shao et al., (2024). Exploring progress in multivariate time series forecasting: Comprehensive benchmarking and heterogeneity analysis. IEEE Transactions on Knowledge and Data Engineering.

---

> ### Author Response · Authors · 2024-11-27
>
> Once again, we would like to emphasize that the key contribution of our work lies in its **pre-training approach,** which is specifically designed to integrate cross-channel dependencies into the channel-independent encoder. As all channel-independent encoders inevitably processes each channel independently, these methods are only able to implicitly learn inter-channel relationships by separating and processing all channels in a multivariate time series within the same batch. However, these methods cannot learn patch-level dependencies between different channels, as they process each channel independently. This prevents the model from learning relationships between channels, such as cross-correlation at lagged positions, which are commonly observed in multivariate time series [2].
>
> To verify this and to further confirm the contribution of ShuffleMTM in capturing patch-level dependencies between different channels, we compare the performance of ShuffleMTM with PatchTST on the aforementioned first synthetic dataset, which exhibits clear patch-level dependencies between channels. As shown in Table 4, ShuffleMTM consistently outperforms PatchTST across all prediction lengths, demonstrating its ability to capture patch-level dependencies in channels with lagged structures. This result confirms that **the pre-training of ShuffleMTM is effective for forecasting multivariate time series (MTS) with patch-level dependencies across channels.**
>
> Furthermore, please note that from our cross-channel dependence analysis in Section 6.1, we conducted a similarity analysis on the shuffled patches series. Specifically, we compared cosine similarities between the self-attention map and the correlation coefficients of patches derived from the randomly shuffled patch series across channels. In this experiment, **ShuffleMTM achieves a higher cosine similarity compared to other channel-independent MTM methods. This result further supports the conclusion that the pre-training of ShuffleMTM enables the channel-independent encoder to better capture patch-level dependencies.**
>
> Once again, we greatly appreciate your insightful feedback, which has enabled us to strengthen our work further. With these additional experimental results and responses, we sincerely hope that we have clarified our contributions in MTM research as an indispensable approach for advancing temporal as well as spatial modeling capabilities of MTM. We sincerely hope we have clarified our contributions and improved the presentation of our work by addressing your comments. We hope that these improvements have increased your confidence in our work and provided you with a more favorable view of its novelty and technical validity.
>
> ---
>
> Table 4. Evaluation of ShuffleMTM and PatchTST on the first synthetic dataset. All experiments are repeated three times.
>
> ||&nbsp;&nbsp;Shuffle|MTM&nbsp;|&nbsp;Patch|TST&nbsp;|
> |:---:|:---:|:---:|:---:|:---:|
> |Pred_len|MSE|MAE|MSE|MAE|
> |32|**1.023**|0.896|1.030|**0.895**|
> |48|**1.024**|**0.897**|1.037|0.901|
> |96|**1.026**|**0.898**|1.033|0.901|
> |192|**1.026**|**0.899**|1.031|0.901|
>
> [2] Zhao, L., & Shen, Y. (2024) Rethinking Channel Dependence for Multivariate Time Series Forecasting: Learning from Leading Indicators. In The Twelfth International Conference on Learning Representations.

---

> ### Author Response · Authors · 2024-12-02
>
> In our previous response, we demonstrated in Table 3 the effectiveness of adopting the shuffled view and cross-attention decoder on a synthetic dataset with high channel dependencies. As we had promised, we have further evaluated ShuffleMTM against its variant without the shuffled view on the PEMS08 dataset as well, which exhibits high periodicity and spatial dependencies [1]. Table 5 presents the forecasting performance of both models for prediction lengths {96, 192, 336, 720}. ShuffleMTM consistently outperforms its variant across all prediction lengths. These results, combined with the findings from Table 3, confirm that **the shuffling method and cross-attention decoder are effective for forecasting in datasets with high channel dependencies.**
>
> We would like to highlight that ShuffleMTM is the first MTM method that captures cross-channel dependence within the channel-independent framework during pre-training. We have shown that ShuffleMTM demonstrates the effectiveness of its pre-training method in capturing cross-channel dependence by comparing with PatchTST on synthetic datasets and showing superior performance over all existing channel-independent MTM baselines on all benchmark datasets. In addition, we have demonstrated the validity of our method by comparing with the ShuffleMTM with the shuffled view on synthetic datasets and real-world datasets. We would be grateful if you could kindly acknowledge our academic and technical contributions to the timeseries learning literature.
>
> We sincerely appreciate the discussion with you, which has been instrumental in clarifying and enhancing the academic contributions of our work and improving its presentation. This discussion, along with future research directions, is now included in Section 7 of the revised manuscript and Section E of the Appendix. Furthermore, previous discussions on the fine-tuning method and cross-channel dependence analysis are detailed in Sections 3.4 and 6.1, respectively. Following this rebuttal, we plan to conduct and include a more in-depth analysis of cross-channel dependence, should our work be accepted at ICLR 2025. Once again, we thank you for your valuable insights and inspiration, and we hope you could kindly acknowledge the unique contributions of ShuffleMTM to the MTM methodology literature and our efforts throughout this rebuttal.
>
>
> ---
>
> Table 5. Comparison between ShuffleMTM and its variant without the shuffled view on the PEMS08 datasets. All experiments were repeated three times.
>
> |Dataset||&nbsp;&nbsp;Shuffle|MTM&nbsp;| &nbsp;&nbsp;ShuffleMTM  |  w/o shuffle &nbsp; &nbsp;|
> |:---:|:---:|:---:|:---:|:---:|:---:|
> ||Pred_len|MSE|MAE|MSE|MAE|
> |PEMS08|96|**0.380**|**0.418**|0.383|0.421|
> ||192|**0.438**|**0.446**|0.442|0.449|
> ||336|**0.394**|**0.407**|0.397|0.409|
> ||720|**0.465**|**0.459**|0.468|0.461|
>
> [1] Shao et al., (2024). Exploring progress in multivariate time series forecasting: Comprehensive benchmarking and heterogeneity analysis. IEEE Transactions on Knowledge and Data Engineering.

---

> ### Author Response · Authors · 2024-12-04
>
> Dear Reviewer g464,
>
> Thank you for your inquiry about the ShuffleMTM’s mechanism of capturing cross-channel dependence in a channel-independent framework. We understand that your concern lies in the challenge of capturing cross-channel dependence without explicitly modeling for relationships between channels. Below, we provide a detailed response to address this issue.
>
> First, we would like to clarify our research topic and objective again. **ShuffleMTM is a novel MTM method that is a pre-training method.** The goal of our study is not to explicitly model cross-channel dependence in the fine-tuning stage but to **inject cross-channel information during the pre-training process of channel-independent masked time-series modeling (MTM)**. By doing so, we can **preserve the advantages of channel-independent learning** while updating the model parameters to produce the representations **capturing dependent information across channels**, achieving both advantages of channel-independent and channel-dependent methods.
>
> In the example you provided with $f(x, y)$, ShuffleMTM utilizes shuffled patches of $\tilde{x}$ and $\tilde{y}$, which represent masked channels, to predict the masked parts of each channel during pre-training. It produces the encoder parameters $\theta_{c|{\tilde{X}\tilde{Y}}}$, where $c \in {x, y}$, which are updated by considering the joint distribution of the two masked channels $\tilde{X}\tilde{Y}$ at each channel reconstruction. In other words, the learned parameters in ShuffleMTM implicitly model the joint distribution of multivariate channels to reconstruct each channel. In contrast, the existing channel-independent MTM pre-trains separate encoder parameters $\theta_{x|\tilde{x}}$ and $\theta_{y|\tilde{y}}$ for each channel $x$ and $y$, independently of the joint distribution at each channel reconstruction.
>
> Consequently, the pre-trained parameters serve as effective initialization, encoding cross-channel dependencies for fine-tuning the channel-independent encoder. This structure aligns with previous analyses of masked modeling, which highlight that the decoder significantly influences the encoder’s learned representations [1].
>
> In addition, we emphasize that ShuffleMTM consistently outperforms its counterparts without the shuffled view across various datasets. For instance, ShuffleMTM demonstrates the superior forecasting performance on synthetic dataset with lagged channel dependencies. Also, our cross-channel dependence analysis in Section 6.1 of the manuscript suggests that pre-training with shuffled masked series captures cross-channel dependence more effectively than the channel-independent MTM methods.
>
> We believe this explanation highlights the technical validity of our architectural design to capture cross-channel dependence within a channel-independent framework. Your valuable feedback has provided an opportunity to better present the technical validity of our work. Thank you again for your review and for the opportunity to refine our work further.
>
> [1] Cao, S., Xu, P., & Clifton, D. A. (2022). How to understand masked autoencoders. arXiv preprint arXiv:2202.03670.

---

### Author Response · Authors · 2024-12-03
**Meta Response to all reviewers**

Dear Reviewers,

We sincerely appreciate your insightful and constructive feedback. The discussions with the reviewers have been invaluable in providing meaningful clarifications and enhancing the quality of our work. Based on your comments, we have made the following improvements: (1) clarified the motivation and academic contributions of our work within the context of time-series research; (2) strengthened the technical contributions of ShuffleMTM; (3) conducted additional ablation experiments; (4) improved the overall presentation of our work.

We would like to use this space to summarize the key points raised by the reviewers and provide our meta responses. These meta responses, addressing the four aspects mentioned above, aim to offer a clearer understanding of the overall revisions made to our work.

---

> ### Author Response · Authors · 2024-12-03
> **Meta Response 1**
>
> ### **Meta Response 1. Clarifying the motivation and academic contributions of our work within the context of time-series research**
>
> ---
>
> Our work addresses a key limitation of channel-independent masked time-series modeling (MTM) research, where the pre-training stage cannot explicitly capture cross-channel dependencies. The existing MTM pre-training method, which focuses on reconstructing masked segments, has been demonstrated to be crucial for capturing cross-time dependencies, thereby enhancing time series forecasting and classification tasks. However, despite the importance of integrating spatial relationships across channels in multivariate time series, the pre-training of previous channel-independent MTM methods have primarily focused on capturing cross-time dependency within each channel, neglecting cross-channel dependencies.
>
> We propose **ShuffleMTM as a novel channel-independent MTM framework that captures both cross-channel and cross-time dependencies.** ShuffleMTM introduces a pre-training approach specifically designed to integrate cross-channel dependencies into the channel-independent encoder—an area not addressed by existing MTM methods. In response to reviewers g464 and 5ATj, we clarified the distinction between ShuffleMTM and forecasting models that address cross-channel dependencies via enhanced attention mechanisms [1, 2]. Unlike these models, which rely on advanced attention mechanisms, **ShuffleMTM introduces a novel self-supervised pre-training task to capture cross-channel dependencies by reconstructing each channel through the integration of original and shuffled views of masked series.** Since our work captures cross-channel dependencies through a pre-training task rather than a meticulously designed attention mechanism, it represents a novel and significant contribution to the time-series learning literature. By capturing cross-channel dependencies within a channel-independent strategy, **ShuffleMTM combines the strengths of both channel-independent and channel-dependent methods.** We have demonstrated that ShuffleMTM enhances both forecasting capability and robustness, which are achieved by channel-dependent and channel-independent methods, respectively.
>
> As research on time series foundation models has gained momentum recently, reviewers 5ATj and riEy suggested comparing ShuffleMTM with these models. We would like to first emphasize that ShuffleMTM pursues distinct research objectives compared to time series foundation models. **While time series foundation models aim to establish pre-trained models** that can be efficiently applied to a wide range of time series tasks by leveraging large-scale datasets, **ShuffleMTM, as a pre-training method, is the first MTM approach** designed to learn representations that capture both cross-time and cross-channel dependencies within a channel-independent framework for specific datasets.
>
> Nevertheless, we agree with the reviewer’s opinion that it is worth comparing ShuffleMTM with time series foundation models. Accordingly, we evaluated the performance of ShuffleMTM against Moirai [3] and Timer [4] in long-term forecasting scenarios to demonstrate its effectiveness in learning time series representations for specific target dataset. For ShuffleMTM and Timer, we followed the standard protocol in which the model is trained and fine-tuned on the same dataset. Meanwhile, since Moirai is a zero-shot forecasting model, we compared ShuffleMTM against the zero-shot performance of Moirai using results reported in its original paper. Table 1 presents the forecasting performance of ShuffleMTM and Timer on four ETT, Exchange, Weather, Electricity, and Traffic datasets, averaged across all prediction lengths. ShuffleMTM outperforms Timer in long-term forecasting for all datasets. Since Timer predicts the future window autoregressively through generative modeling, it accumulates errors in long-term forecasting. Table 2 presents the forecasting performance of ShuffleMTM and Moirai on four ETT, Weather, and Electricity datasets, averaged across all prediction lengths. ShuffleMTM demonstrates comparable performance to Moirai. **These experimental results underscore the effectiveness of ShuffleMTM over time series foundation models in learning representations specific to the dataset for time series forecasting tasks.**

---

> > ### Author Response · Authors · 2024-12-03
> > **(Cont'd) Meta Response 1**
> >
> > Table 1. Comparison with Timer. All experiments are implemented three times. We report the average forecasting performance for all prediction lengths \{96, 192, 336, 720\}.
> >
> > |   |&nbsp;&nbsp;Shuffle|MTM&nbsp;|Timer   |   |
> > |:-----------:|:-----:|:-----:|:-----:|:-----:|
> > |Dataset    |MSE  |MAE  |MSE  |MAE  |
> > |ETTh1      |**0.432**|**0.435**|0.439|0.436|
> > |ETTh2      |**0.372**|**0.399**|0.395|0.411|
> > |ETTm1      |**0.379**|**0.394**|0.430|0.420|
> > |ETTm2      |**0.278**|**0.325**|0.296|0.333|
> > |Exchange   |**0.353**|**0.398**|0.357|0.401|
> > |Weather    |**0.250**|**0.275**|0.279|0.294|
> > |Electricity|**0.186**|**0.272**|0.230|0.303|
> > |Traffic    |**0.449**|**0.281**|0.492|0.312|
> >
> >
> >
> > Table 2. Comparison with MOIRAI. We report the average forecasting performance for all prediction lengths \{96, 192, 336, 720\}.
> >
> > |   |&nbsp;&nbsp;Shuffle|MTM&nbsp;|MOIRAI   |   |
> > |:-----------:|:-----:|:-----:|:-----:|:-----:|
> > |Dataset    |MSE  |MAE  |MSE  |MAE  |
> > |ETTh1      |0.432|0.435|**0.400**|**0.424**|
> > |ETTh2      |0.372|0.399|**0.341**|**0.379**|
> > |ETTm1      |**0.379**|**0.394**|0.448|0.410|
> > |ETTm2      |**0.278**|**0.325**|0.300|0.341|
> > |Weather    |0.250|0.275|**0.242**|**0.267**|
> > |Electricity|**0.186**|**0.272**|0.233|0.320|
> >
> > ---
> >
> > [1] Zhang, Y., & Yan, J (2023). Crossformer: Transformer utilizing cross-dimension dependency for multivariate time series forecasting. In The eleventh international conference on learning representations.
> >
> > [2] Liu, J., Liu, C., Woo, G., Wang, Y., Hooi, B., Xiong, C., & Sahoo, D. (2024). UniTST: Effectively Modeling Inter-Series and Intra-Series Dependencies for Multivariate Time Series Forecasting. arXiv preprint arXiv:2406.04975.
> >
> > [3] Woo, G., Liu, C., Kumar, A., Xiong, C., Savarese, S., & Sahoo, D. (2024) Unified Training of Universal Time Series Forecasting Transformers. In Forty-first International Conference on Machine Learning.
> >
> > [4] Liu, Y., Zhang, H., Li, C., Huang, X., Wang, J., & Long, M. (2024) Timer: Generative Pre-trained Transformers Are Large Time Series Models. In Forty-first International Conference on Machine Learning.

---

> ### Author Response · Authors · 2024-12-03
> **Meta Response 2**
>
> ### **Meta Response 2. Clarifying and strengthening the technical contribution of our work**
>
> ---
>
> We would like to emphasize that the key technical contribution of ShuffleMTM lies in its pre-training approach, which is specifically designed to integrate cross-channel dependencies into the channel-independent encoder. In particular, we propose reconstructing each channel by integrating original and shuffled masked series. To achieve this integration, our approach involves generating original and shuffled masked series, encoding both views simultaneously using Siamese encoders, and utilizing a cross-attention decoder to integrate the cross-time and cross-channel dependencies encoded in the representations to reconstruct each channel. **Thus, the shuffling method, Siamese encoders, and cross-attention decoder are essential components for capturing cross-time and cross-channel dependencies within the channel-independent framework.** The architectural design of ShuffleMTM represents a significant technical contribution, as it extends the channel-independent reconstruction task to effectively capture both types of dependencies, leveraging this novel architecture.
>
>
> In response to comments from reviewers g464, riEy, and wko4, **we conducted simulation experiments to examine the cross-channel dependencies captured by ShuffleMTM.** We evaluated the forecasting performance of ShuffleMTM and PatchTST on two synthetic datasets, each meticulously designed to exhibit different channel dependency structures. On the first synthetic dataset, which exhibits a lagged structure between patches across channels, ShuffleMTM outperforms PatchTST in all settings (see Table 3). **This result demonstrates ShuffleMTM’s capability to capture patch-level dependencies between channels with lagged structures.** In contrast, on the second synthetic dataset, where all channels share the same long-term trend but have different local patterns at each time step, ShuffleMTM underperforms compared to PatchTST. This outcome indicates that ShuffleMTM is less effective in short-term forecasting when channels are independent of each other in a local context. However, ShuffleMTM becomes more effective in long-term forecasting on this dataset, as demonstrated by the reduced performance gap between the two models, leveraging the shared long-term context across channels. As reviewer wko4 mentioned, we believe that the analysis of cross-channel dependence that ShuffleMTM captures strengthens the technical contribution of our work.
>
> ---
>
> Table 3. Evaluation of ShuffleMTM and PatchTST on two synthetic datasets. All experiments are repeated three times.
>
> |||&nbsp;&nbsp;Shuffle|MTM&nbsp;|&nbsp;Patch|TST&nbsp;|
> |:---:|:---:|:---:|:---:|:---:|:---:|
> |Dataset|Pred_len|MSE|MAE|MSE|MAE|
> |Synthetic|32|**1.023**|0.896|1.030|**0.895**|
> |data1|48|**1.024**|**0.897**|1.037|0.901|
> ||96|**1.026**|**0.898**|1.033|0.901|
> ||192|**1.026**|**0.899**|1.031|0.901|
> |Synthetic|32|0.292|0.477|**0.284**|**0.470**|
> |data2|48|0.292|0.477|**0.288**|**0.474**|
> ||96|0.294|0.479|**0.292**|**0.478**|
> ||192|0.296|0.480|**0.294**|**0.479**|

---

> > ### Author Response · Authors · 2024-12-03
> > **(Cont'd) Meta Response 2**
> >
> > Through synthetic datasets, we validated our method via simulation experiments. Specifically, we evaluated the forecasting performance of ShuffleMTM and its variant without the shuffled view on two synthetic datasets, each meticulously designed to exhibit different channel dependencies. The ShuffleMTM variant without the shuffled view removes the shuffled view and uses only the original view as the query, key, and value in the decoder. Comparing this variant with ShuffleMTM highlights the effectiveness of utilizing the shuffled masked series and the cross-attention decoder for pre-training channel-independent MTM. As shown in Table 4, ShuffleMTM outperformed its variant on both datasets, with a larger performance gap observed in the first dataset than in the second. These experiments suggest that **the shuffling method enables the channel-independent encoder to capture fine-grained dependencies between channels.** However, **the shuffling method is less effective on datasets with weak local dependencies compared to those with strong local dependencies.** In addition, we further evaluated ShuffleMTM against its variant without the shuffled view on the PEMS08 dataset, a benchmark dataset exhibiting strong periodicity and spatial dependencies. ShuffleMTM consistently outperforms its variant across all prediction lengths (see Table 5). These results, combined with findings from the simulation experiments, confirm that the shuffling method and the cross-attention decoder are effective for forecasting in datasets with high channel dependencies. We have included this in-depth analysis in Section E of the Appendix.
> >
> > ---
> >
> > Table 4. Comparison between ShuffleMTM and its variant without the shuffled view on two synthetic datasets. All experiments were repeated three times.
> >
> > |Dataset||&nbsp;&nbsp;Shuffle|MTM&nbsp;| &nbsp;&nbsp;ShuffleMTM  |  w/o shuffle &nbsp; &nbsp;|
> > |:---:|:---:|:---:|:---:|:---:|:---:|
> > ||Pred_len|MSE|MAE|MSE|MAE|
> > |Synthetic|32|**1.023**|**0.896**|1.034|0.899|
> > |data 1|48|**1.024**|**0.897**|1.034|0.899|
> > ||96|**1.026**|**0.898**|1.033|0.898|
> > ||192|**1.026**|0.899|1.029|**0.897**|
> > |Synthetic|32|**0.292**|**0.477**|0.293|0.479|
> > |data 2|48|**0.292**|**0.477**|0.293|0.479|
> > ||96|**0.294**|**0.479**|0.295|**0.479**|
> > ||192|**0.296**|**0.480**|**0.296**|**0.480**|
> >
> > Table 5. Comparison between ShuffleMTM and its variant without the shuffled view on the PEMS08 datasets. All experiments were repeated three times.
> >
> > |Dataset||&nbsp;&nbsp;Shuffle|MTM&nbsp;| &nbsp;&nbsp;ShuffleMTM  |  w/o shuffle &nbsp; &nbsp;|
> > |:---:|:---:|:---:|:---:|:---:|:---:|
> > ||Pred_len|MSE|MAE|MSE|MAE|
> > |PEMS08|96|**0.380**|**0.418**|0.383|0.421|
> > ||192|**0.438**|**0.446**|0.442|0.449|
> > ||336|**0.394**|**0.407**|0.397|0.409|
> > ||720|**0.465**|**0.459**|0.468|0.461|

---

> > > ### Author Response · Authors · 2024-12-03
> > > **(Cont'd) Meta Response 2**
> > >
> > > As a result of pre-training, the learned representations inherently capture cross-channel information. Although the encoder processes channels independently during fine-tuning and inference, this approach enables the model to utilize channel dependencies without altering the channel-independent nature of the encoder during these stages. Thus, while the encoder processes each channel independently during inference, the pre-trained representations reflect the cross-channel dependencies learned during pre-training. ShuffleMTM may not encode sudden noise from other channels during inference. However, the encoder's outputs can still be influenced by the cross-channel dependencies learned during pre-training, even if the other channels are noisy during inference. Notably, this challenge is inherent to all channel-independent MTM methods, as the channel-independent encoder inherently processes each channel independently during inference. Given this, **we emphasize that pre-training to integrate cross-channel information into the channel-independent encoder is an indispensable approach for MTM.** We believe our pre-training method significantly enhances the model’s ability to capture and utilize cross-channel information during downstream tasks, even if certain edge cases may exhibit limitations.
> > >
> > > To clarify, the essential contribution of our research is the development of **a novel MTM approach that enables channel-independent models to incorporate cross-channel information during the pre-training stage.** As other MTM studies focus on enhancing the pre-training method, it is essential to validate the effectiveness of the pre-training stage for downstream tasks using the same fine-tuning process. We experimentally demonstrate that ShuffleMTM outperforms state-of-the-art MTM methods that share the same channel-independent fine-tuning process. While our model demonstrates comparable performance with state-of-the-art forecasting models that explicitly consider cross-channel dependencies on certain datasets, it is significant that the channel-independent encoder achieves competitive performance with these channel-dependent models through the proposed pre-training approach. Thus, we emphasize that ShuffleMTM achieves notable performance in forecasting tasks. Furthermore, ShuffleMTM demonstrates superior performance in time series classification, confirming the universal effectiveness of our pre-training approach across various downstream tasks. These results validate that ShuffleMTM effectively captures cross-channel dependencies in multivariate time series with strong spatial dependency structures. Additionally, the mechanisms of ShuffleMTM enable pre-training to successfully learn cross-channel dependencies.

---

> > > > ### Author Response · Authors · 2024-12-03
> > > > **Meta Response 3**
> > > >
> > > > ### **Meta Response 3. Conducting additional ablation experiments**
> > > >
> > > > ---
> > > >
> > > > Based on feedback from reviewers g464 and 5ATj, we conducted additional ablation experiments to investigate the effects of hyperparameters in ShuffleMTM. First, we varied the look-back window sizes and evaluated the forecasting performance, consistently demonstrating superior performance compared to other channel-independent MTM methods. Second, we examined the effect of the masking ratio on high-channel datasets. Our analysis revealed that the masking ratio regulates the degree of self-supervision and the diversity of patch replacements during the shuffling process, underscoring its significant impact on forecasting performance.
> > > >
> > > > These additional experiments further validate the effectiveness and technical soundness of our approach. The results have been included in Section 5 of the revised manuscript, with updates highlighted in blue for your reference.

---

> > > > > ### Author Response · Authors · 2024-12-03
> > > > > **Meta Response 4**
> > > > >
> > > > > ### **Meta Response 4. Improving the overall presentation of our work**
> > > > >
> > > > > ---
> > > > >
> > > > > In this rebuttal, we have enhanced the clarity of our proposed ShuffleMTM and improved the presentation of our work. The key improvements to the manuscript are summarized as follows:
> > > > >
> > > > > 1)	**Enhanced presentation of Figures 1 and 2**: We have added detailed explanations of the colors and notations used in Figures 1 and 2, enabling clearer positioning of our work within existing masked time-series modeling research and improving the clarity of our framework. Please refer to the captions of Figures 1 and 2 in L62–64 and L177–182.
> > > > >
> > > > > 2)	**Improved cross-channel dependence analysis**: We have clarified the cross-channel dependence analysis by detailing the procedure for computing measures and presenting analytic results. Please refer to the blue-colored text in Section 6.1.
> > > > >
> > > > > 3)	**Fine-tuning process of ShuffleMTM**: We have outlined the fine-tuning process of ShuffleMTM, illustrating how the channel-independent encoder reflects cross-channel dependencies during fine-tuning. Please refer to Section 3.4 in the revised manuscript.
> > > > >
> > > > > 4)	**Comprehensive experimental results**: We now include the full experimental results presented in Tables 1 to 4, along with the standard deviation of evaluation results across all trials. Please refer to Section F in the Appendix.
> > > > >
> > > > > Once again, we sincerely appreciate your time and effort in providing constructive comments and engaging in discussions. Your feedback has been invaluable in strengthening the academic and technical contributions of our work, conducting additional ablation studies, and improving the clarity and presentation of the manuscript.
> > > > >
> > > > > Thank you very much.

---

### Meta-Review · Area_Chair_FJH3 · 2024-12-20

**Metareview:**

This paper aims to address the limitations of existing channel-independent (CI) masked time series modeling methods by learning cross-channel dependence from shuffled patches. It trains Siamese encoders to learn two views of masked patch representations to capture the temporal dependence within a channel and spatial dependence across multiple channels.

Major strengths:
- Learning cross-channel dependence effectively is an important topic in multivariate time series modeling.

Major weaknesses:
- The effectiveness of the proposed pipeline which involves a multi-channel pre-training stage followed by a single-channel fine-tuning stage for modeling channel dependency needs to be justified more thoroughly.
- Since no theoretical justification for the proposed method is presented in the paper, it would need strong experimental validation to demonstrate the effectiveness of the method. However, for datasets with strong correlation between channels, no significant improvement can be observed.
- The effectiveness of the shuffle operation is dataset-dependent, but it may not be easy to know in advance whether it will help or not before trying it out.

Despite studying an important topic in multivariate time series modeling, the effectiveness of the somewhat counter-intuitive method needs to be justified more convincingly, either through theoretical justification or based on comprehensive experimentation. Among other things, some additional experiments can use synthetic datasets in which some channels are really correlated.

The authors are encouraged to improve their paper for future submission by considering the comments and suggestions of the reviewers.

**Additional Comments On Reviewer Discussion:**

In my batch, this is the most difficult paper to handle. Even after extensive discussions involving the authors and all reviewers, a consensus was not reached among the reviewers, with overall ratings (after adjustment) ranging from 3 (reject, not good enough) to 8 (accept, good paper).

The reviewer giving the highest rating of 8 (in fact s/he would prefer 7 if there were such an option) explicitly says that s/he wouldn’t mind if the AC decides to reject the paper. There is also strong reservation from another reviewer that such a channel-independent method would not be able to capture cross-channel dependencies effectively.

---

### Decision · Program_Chairs · 2025-01-22

Reject